# A STOCHASTIC GRADIENT METHOD FOR TRILEVEL OPTIMIZATION

## ABSTRACT

With the success that the field of bilevel optimization has seen in recent years, similar methodologies have started being applied to solving more difficult applications that arise in trilevel optimization. At the helm of these applications are new machine learning formulations that have been proposed in the trilevel context and, as a result, efficient and theoretically sound stochastic methods are required. In this work, we propose the first-ever stochastic gradient descent method for solving unconstrained trilevel optimization problems and provide a convergence theory that covers all forms of inexactness of the trilevel adjoint gradient, such as the inexact solutions of the middle-level and lower-level problems, inexact computation of the trilevel adjoint formula, and noisy estimates of the gradients, Hessians, Jacobians, and tensors of third-order derivatives involved. We also demonstrate the promise of our approach by providing numerical results on both synthetic trilevel problems and trilevel formulations for hyperparameter adversarial tuning.

## 1 INTRODUCTION

Multi-level optimization (MLO) is a general class of problems with the goal of optimizing an upper-level objective while requiring subsets of the considered variables to satisfy optimality principles for some number of nested sub-problems. Hierarchical in nature, these MLO problems have a variety of applications that appear in fields such as defense industry Arguello et al. (2023); Lai et al. (2019); Yao et al. (2007); Wu & Conejo (2017); Guo et al. (2023), signal recovery and power control Liduka (2011); Cang & Petrusel (2010), supply chain networks Xu et al. (2013); Rahdar et al. (2018); Fathollahi-Fard et al. (2018), and more recently in the field of machine learning Jiao et al. (2023); Choe et al. (2022); Guo et al. (2019); Jiao et al. (2024); Liu et al. (2019); Giovannelli et al. (2025); Jin et al. (2019). Due to the difficulty of these MLO problems, most of the algorithms have largely only been developed for solving the bilevel case. However, the trilevel case has recently seen further interest by applying similar methodologies that have been utilized in the bilevel case. With this interest comes the aim of developing efficient and theoretically sound first-order stochastic gradient methods for handling large-scale applications of trilevel optimization problems that arise in the field of machine learning. As far as we know, this is the first work that addresses the stochastic setting of a trilevel problem, both theoretically and numerically.

In this paper, we consider the general trilevel optimization (TLO) problem formulation

$$
\begin{aligned}
\min_{x\in\mathbb{R}^n,\, y\in\mathbb{R}^m,\, z\in\mathbb{R}^t} \quad & f_1(x,y,z) \\
\text{s.t.} \quad & x \in X \\
& y,z \in \operatorname*{arg\,min}_{y\in Y(x),\, z\in\mathbb{R}^t} f_2(x,y,z) \\
& \text{s.t.} \ z \in \operatorname*{arg\,min}_{z\in Z(x,y)} f_3(x,y,z).
\end{aligned}
\qquad \text{TLO}
$$

The goal of the upper-level (UL) problem is to determine the optimal value of the UL function $f_1 : \mathbb{R}^n \times \mathbb{R}^m \times \mathbb{R}^t \to \mathbb{R}$, where the UL variables $x$ are subjected to UL constraints ($x \in X$), the middle-level (ML) variables $y$ are subjected to being an optimal solution of the ML problem, and the lower-level (LL) variables $z$ are subjected to being an optimal solution of the LL problem. In the ML problem, the ML function $f_2 : \mathbb{R}^n \times \mathbb{R}^m \times \mathbb{R}^t \to \mathbb{R}$ is optimized in the ML variables $y$, subject to

the ML constraints $y \in Y(x)$. Similarly, in the LL problem, the LL function $f_3 : \mathbb{R}^n \times \mathbb{R}^m \times \mathbb{R}^t \to \mathbb{R}$ is optimized in the LL variables $z$, subject to the constraints $z \in Z(x, y)$. In this paper, we will assume that the ML and LL problems are strongly convex (see Subsection 3.1 below) and that the UL problem is possibly nonconvex (see Theorem B.6 below).

## 1.1 Trilevel optimization in the literature

Trilevel and multi-level optimization has been studied as early as the 1980s (see Blair (1992); Bard & Falk (1982); Bard (1984); Ue-Pyng & Bialas (1986); Benson (1989)), but in many of the afore-mentioned fields (e.g., defense industry, supply chain networks, etc.), problem-specific formulations typically lack general solution methodologies. We mention here a few notable exceptions that do consider general methodologies. The authors of Tilahun et al. (2012) introduced an evolutionary strategy to update each level sequentially, but without convergence guarantees. In contrast, the authors of Shafiei et al. (2024) proposed a proximal gradient method for TLO problems with convex objective functions, offering convergence guarantees but lacking numerical validation. For thorough reviews of the development of multi-level optimization, see the surveys Vicente & Calamai (1994); Lu et al. (2016); Liu et al. (2021); Chen et al. (2022a).

**Trilevel optimization for machine learning.** More recently, TLO (also referred to as *trilevel learning* when taking on applications in a machine learning context) and MLO problems have seen utilization in being applied to solving large-scale hierarchical machine learning problems with applications of hyperparameter tuning, adversarial learning, and federated learning. In Sato et al. (2021), the authors developed a gradient-based method for solving an approximate formulation of the general MLO problem, as well as presenting convergence guarantees and numeric results for their method in the deterministic case. Such a paper builds on pre-existing methods utilized in Franceschi et al. (2017) for the bilevel case that approximate the solution to each of the lower-level problems with an iterative method. Complimenting this development, the authors of Choe et al. (2022) introduced BETTY, an automatic differentiation library for general multi-level optimization, which has helped facilitate applications like neural architecture search (NAS) with adversarial robustness Guo et al. (2019). Trilevel optimization has also been further extended to decentralized learning environments in Jiao et al. (2023; 2024), where the authors aim at developing methods with convergence guarantees for federated trilevel learning problems. However, it bears mentioning that all of the aforementioned papers only consider the deterministic setting in their analysis.

## 1.2 Contributions of this paper

The field of bilevel optimization has seen a rich development of first-order descent methods for solving large-scale problems that arise in the field of machine learning (e.g., see Chen et al. (2022b; 2021); Liu et al. (2019); Giovannelli et al. (2025; 2024); Jin et al. (2019)). However, as we have seen in the existing literature, no works have yet begun extending the theory and implementation of stochastic methods to trilevel and higher-level problems. In this paper, we propose TSG, the first stochastic gradient method for solving trilevel optimization problems, along with an extensive convergence analysis with general nonlinear and nonconvex UL functions. This is done by extending the concepts and methodologies developed for first-order bilevel optimization methods that utilize the so-called adjoint gradient (or hyper-gradient) via implicit differentiation, and adapting them to the trilevel setting. To address the significant difficulties imposed by the presence of second-order and third-order derivatives in handling these problems, we also propose practical and efficient strategies for implementing our TSG method and demonstrate its performance on a series of trilevel problems. The numerical results show that the trilevel formulation we propose for hyperparameter adversarial tuning consistently yields the most robust performance across all tested datasets, outperforming the corresponding bilevel formulations for both hyperparameter tuning and adversarial learning.

## 2 Trilevel optimization

In this paper, we will only focus on the unconstrained ML and LL cases of problem TLO, i.e., $Y(x) = \mathbb{R}^m$ and $Z(x, y) = \mathbb{R}^t$. Since our goal is to propose and analyze a general optimization methodology for a stochastic TLO, the LL problem is assumed to be well-defined, in the sense of having a unique solution $z(x, y)$ for all $x \in \mathbb{R}^n$ and $y \in \mathbb{R}^m$. Thus, problem TLO is equivalent

to the following bilevel optimization (BLO) problem, which is defined solely in terms of the UL and ML variables:

$$\min_{x \in \mathbb{R}^n, \, y \in \mathbb{R}^m} \quad f_1(x, y, z(x,y))$$

$$\text{s.t.} \quad y \in \arg\min_{y \in \mathbb{R}^m} \quad \bar{f}(x,y) := f_2(x, y, z(x,y)). \tag{BLO}$$

Similarly, problem BLO can be even further reduced to a single-level optimization problem under the assumption that the lower-level problem in BLO also has a unique solution $y(x)$. In this way, since $y(x)$ is solely determined by $x$, it is clear that the unique solution $z(x, y(x))$ is solely determined by $x$ as well, which we denote simply as $z(x)$. Thus, problem TLO ultimately reduces to the single-level optimization problem given by

$$\min_{x \in \mathbb{R}^n} \quad f(x) = f_1(x, y(x), z(x, y(x))) \quad \text{s.t.} \quad x \in X. \tag{2.1}$$

We define the trilevel adjoint gradient of $f$ at $x$ as

$$\nabla f = (\nabla_x f_1 - \nabla_{xz}^2 f_3 \nabla_{zz}^2 f_3^{-1} \nabla_z f_1) - \nabla_{xy}^2 \bar{f} \nabla_{yy}^2 \bar{f}^{-1} (\nabla_y f_1 - \nabla_{yz}^2 f_3 \nabla_{zz}^2 f_3^{-1} \nabla_z f_1), \tag{2.2}$$

where all of the gradient and Hessian terms involved are evaluated at the point $(x, y(x), z(x))$. Notice that this is essentially a classical adjoint gradient calculation applied to problem BLO. The complete statement, along with all term definitions and full derivation, is given by Proposition A.1 in Appendix A.

## 2.1 THE TRILEVEL STOCHASTIC GRADIENT METHOD

The stochastic algorithm developed in this paper proceeds by iteratively updating the LL variables first, followed by the ML variables, and lastly the UL variables. The iterations corresponding to the UL, ML, and LL problems are denoted by $i$, $j$, and $k$, respectively, with the total number of iterations denoted as $I$, $J$, and $K$, respectively. Let $\{\xi^i\}$, $\{\xi^{i,j}\}$, and $\{\xi^{i,j,k}\}$ denote sequences of random variables defined in a probability space (with probability measure independent from $x$, $y$, and $z$) such that i.i.d. samples can be observed or generated. Such random variables are introduced for gradient, Jacobian, and Hessian evaluations, and their realizations can be interpreted as a single sample or a batch of samples for a mini-batch stochastic gradient (SG). For simplicity, we also adopt the following terminology throughout this paper: $z^{i,j} = z^{i,j,0}$, $z^{i,j+1} = z^{i,j+1,0} = z^{i,j,K}$, $z^i = z^{i,0,0}$, and $z^{i+1} = z^{i+1,0,0} = z^{i,J,K}$ for the LL iterations and $y^i = y^{i,0}$ and $y^{i+1} = y^{i+1,0} = y^{i,J}$ for the ML iterations. Most of this terminology is merely notation; however, by letting $z^{i+1} = z^{i,J,K}$, $z^{i,j+1} = z^{i,j,K}$, and $y^{i+1} = y^{i,J}$, we are saying that the initial iterates for new cycles are the last ones of the previous corresponding cycles.

Given the current iterate $(x^i, y^{i,j}, z^{i,j,k})$, the update direction that is used for the LL problem is simply the stochastic gradient of the LL objective function $f_3$, denoted as $g_{f_3}^{i,j,k}$ and given by $g_{f_3}^{i,j,k} = \nabla_z f_3(x^i, y^{i,j}, z^{i,j,k}; \xi^{i,j,k})$. Letting $\gamma_i \in (0, 1]$ denote the step size for the LL problem at the UL iteration $i$, the update of the LL variables is given by $z^{i,j,k+1} = z^{i,j,k} - \gamma_i g_{f_3}^{i,j,k}$. The SG algorithm used to obtain the approximate solution $z^{i,j+1} \approx z(x^i, y^{i,j})$ is stated by Algorithm 1.

The exact gradient for the ML problem is computed via the following standard adjoint gradient (by combining equations equation A.9 and equation A.4 in Appendix A):

$$\nabla_y \bar{f}(x, y) = \nabla_y f_2 - \nabla_{yz}^2 f_3 \nabla_{zz}^2 f_3^{-1} \nabla_z f_2, \tag{2.3}$$

where all gradients and Hessians are evaluated at the point $(x, y, z(x, y))$. However, since we solve the LL problem inexactly to obtain an approximate solution $z^{i,j+1} \approx z(x^i, y^{i,j})$, the ML adjoint gradient equation 2.3 now becomes "inexact". Thus, given the current iterate $(x^i, y^{i,j}, z^{i,j+1})$, the update direction that is used for the ML problem is the inexact stochastic gradient of the function $\bar{f}$, denoted as $\tilde{g}_{f_2}^{i,j}$ and given by

$$\tilde{g}_{f_2}^{i,j} = \nabla_y \bar{f}(x^i, y^{i,j}, z^{i,j+1}; \xi^{i,j}) = \nabla_y f_2 - \nabla_{yz}^2 f_3 \nabla_{zz}^2 f_3^{-1} \nabla_z f_2, \tag{2.4}$$

where all gradients and Hessians are evaluated at the point $(x^i, y^{i,j}, z^{i,j+1}; \xi^{i,j})$. We highlight this slight abuse of notation, since $\bar{f}$ is a function of $(x, y)$ and not $(x, y, z)$, as we are utilizing the

approximation $z^{i,j+1} \approx z(x, y^{i,j})$ in computing the gradient $\nabla_y \bar{f}$. It is for this reason that we adopt the notation $\tilde{g}_2$ to denote an "inexact" SG (as opposed to simply $g_2$, which would denote the "exact" SG $\nabla \bar{f}(x^i, y^{i,j})$). Letting $\beta_i \in (0,1]$ denote the step size for the ML problem at the UL iteration $i$, the update of the ML variables is given by $y^{i,j+1} = y^{i,j} - \beta_i \tilde{g}_{f_2}^{i,j}$. The bilevel SG algorithm that is used to obtain the approximate solution $y^{i+1} \approx y(x^i)$ is stated by Algorithm 2. It bears mentioning that after every ML iteration, we will perform another LL update to obtain an approximation $z^{i+1}$ to $z(x^i, y^{i+1})$.

---

**Algorithm 1** SG (LL Problem)

---

**Input:** Initial $z^{i,j,0}$, $\gamma_i \in (0,1]$.
**For** $k = 0, 1, 2, \ldots, K-1$ **do**
    **1.** Compute an SG $g_{f_3}^{i,j,k}$.
    **2.** Update $z^{i,j,k+1} = z^{i,j,k} - \gamma_i\, g_{f_3}^{i,j,k}$.
**Return** $z^{i,j+1} = z^{i,j,K}$.

---

**Algorithm 2** Bilevel SG (ML Problem)

---

**Input:** Initial $y^{i,0}$, $\beta_i \in (0,1]$, $\gamma_i \in (0,1]$.
**For** $j = 0, 1, 2, \ldots, J-1$ **do**
    **1.** Compute $z^{i,j+1}$ via Algorithm 1
    **2.** Compute an approximation $\tilde{g}_{f_2}^{i,j}$.
    **3.** Update $y^{i,j+1} = y^{i,j} - \beta_i\, \tilde{g}_{f_2}^{i,j}$.
**Return** $(y^{i+1} = y^{i,J},\ z^{i,J})$.

---

Now, recall that the exact gradient for the UL problem is computed via the trilevel adjoint gradient given by equation equation 2.2. Since we only solve the ML problem inexactly to obtain an approximate solution $y^{i+1} \approx y(x^i)$, the trilevel adjoint gradient equation 2.2 also becomes "inexact". Notice that the inexactness here comes from two sources: one related to the inexactness of the LL variables and the other to the inexactness of the ML variables. The first source of inexactness arises from the two Hessian terms of the true ML problem, i.e., $\nabla_{xy}^2 \bar{f}$ and $\nabla_{yy}^2 \bar{f}^{-1}$,

---

**Algorithm 3** Trilevel Stochastic Gradient (TSG)

---

**Input:** Initial $(x^0, y^{0,0}, z^{0,0,0})$, $\alpha_i \in (0,1]$, $\beta_i \in (0,1]$, $\gamma_i \in (0,1]$.

**For** $i = 0, 1, 2, \ldots, I-1$ **do**
    **1.** Compute $y^{i+1} = y^{i,J}$ and $z^{i,J,0}$ via Algorithm 2.
    **2.** Compute $z^{i+1} = z^{i,J,K}$ via Algorithm 1.
    **3.** Compute an approximation $\tilde{g}_{f_1}^i$.
    **4.** Update $x^{i+1} = x^i - \alpha_i\, \tilde{g}_{f_1}^i$.
**Return** $x^I$.

---

due to them being evaluated at the approximate solution $z^{i+1}$ instead of $z(x^i, y(x^i))$. The second source of inexactness comes from all of the terms involved being evaluated at the approximate solution $y^{i+1}$ instead of $y(x^i)$. Thus, given the current iterate $(x^i, y^{i+1}, z^{i+1})$, the update direction that is used for the UL problem is the inexact stochastic gradient of $f$, denoted as $\tilde{g}_{f_1}^i$ and given by

$$\tilde{g}_{f_1}^i = \nabla f(x^i, y^{i+1}, z^{i+1}; \xi^i). \tag{2.5}$$

We again highlight this slight abuse of notation, since $f$ is a function of $(x)$ and not $(x, y, z)$, as we are utilizing the approximations $y^{i+1} \approx y(x^i)$ and $z^{i+1} \approx z(x^i, y(x^i))$ in computing the gradient $\bar{f}$. It is again for this reason that we adopt the notation $\tilde{g}_1$ to denote an "inexact" SG (as opposed to simply $g_1$, which would denote the "exact" SG $\nabla f(x^i)$). Letting $\alpha_i \in (0,1]$ denote the step size for the UL problem in the UL iteration $i$, the update of the UL variables is given by $x^{i+1} = x^i - \alpha_i \tilde{g}_{f_1}^i$. Finally, the schema of the resulting trilevel stochastic gradient (TSG) algorithm developed in this paper is given by Algorithm 3.

## 3   CONVERGENCE ANALYSIS OF THE TSG METHOD

Throughout this section, to simplify notation when there are no ambiguities, we will write functions, gradients, Jacobians, and Hessians by omitting their arguments $(x, y, z)$. When dealing with stochastic estimates, we will replace the arguments $(x, y, z; \xi)$ with an $\xi$-superscript. For example, we denote $\nabla_{zz}^2 f_3^\xi = \nabla_{zz}^2 f_3(x, y, z; \xi)$. It also bears mentioning that in the following assumptions, we will omit the iterates $(i, j, k)$ for the evaluated point $(x, y, z)$ and the iterate $i$ for the step sizes $\alpha$, $\beta$, and $\gamma$, as the results are required to hold true for any iterate. For convenience throughout the analysis, we utilize the following composite step-size:

$$\theta_i := \alpha_i \beta_i \gamma_i \quad (\text{or } \theta := \alpha\beta\gamma \text{ in the general case}). \tag{3.1}$$

Further, we define the expectations to be taken over $\sigma$-algebras generated by the sets of the relevant random variables. For simplicity, we define a general $\sigma$-algebra $\mathcal{F}_\xi$ that includes all the events up to the generation of a general point $(x, y, z)$, before observing a realization of $\xi$. Further, $\mathbb{E}[\cdot|\mathcal{F}_\xi]$ denotes the expectation taken with respect to the probability distribution of $\xi$ given $\mathcal{F}_\xi$. We will also use $\mathbb{E}[\cdot]$ to denote the *total expectation*, i.e., the expected value with respect to the joint distribution of all the random variables. For a full description of all $\sigma$-algebras used in the analysis, see Section B.1 of Appendix B.

### 3.1 Assumptions on the trilevel problem

We now provide all of the assumptions that are required for the convergence analysis of Algorithm 3. It bears mentioning that throughout this paper, we use $\| \cdot \|$ to denote the $\ell_2$-Euclidean norm when dealing with vectors and the spectral norm when dealing with matrices. We begin by imposing Assumption 3.1 below which ensures that the functions of interest are differentiable and satisfy appropriate smoothness requirements on the functions, gradients, Jacobians, Hessians, and tensors of third-order derivatives involved in problem TLO.

**Assumption 3.1 (Differentiability and Lipschitz continuity)** *The function $f_1$ is once continuously differentiable, $f_2$ is twice continuously differentiable, and $f_3$ is thrice continuously differentiable. Further, the functions $f_1$, $\nabla f_1$, $f_2$, $\nabla f_2$, $\nabla^2 f_2$, $\nabla f_3$, $\nabla^2 f_3$, and $\nabla^3 f_3$ are Lipschitz continuous with constants $L_{f_1}$, $L_{\nabla f_1}$, $L_{f_2}$, $L_{\nabla f_2}$, $L_{\nabla^2 f_2}$, $L_{\nabla f_3}$, $L_{\nabla^2 f_3}$, and $L_{\nabla^3 f_3}$, respectively.*

To ensure that problem TLO is well-defined, Assumptions 3.2–3.3 below require that the LL function $f_3$ as well as the true ML function $\bar{f}$ are strongly convex. These kind of assumptions are standard in the stochastic approximation literature (e.g., see Ghadimi & Wang (2018)) and will guarantee the existence and uniqueness of the ML and LL optimal solutions $y(x)$ and $z(x)$, respectively, for any fixed value of $x$. Further, the constants $\mu_z$ and $\mu_y$ defined in these assumptions are positive.

**Assumption 3.2 (Strong convexity of $f_3$ in $z$)** *For any fixed $x$ and $y$, $f_3$ is $\mu_z$-strongly convex in $z$, i.e., $f_3(x, y, z_1) \geq f_3(x, y, z_2) + \nabla_z f_3(x, y, z_2)^\top (z_1 - z_2) + \frac{\mu_z}{2}\|z_1 - z_2\|^2$, for all $(z_1, z_2)$.*

**Assumption 3.3 (Strong convexity of $\bar{f}$ in $y$)** *For any fixed $x$, $\bar{f}$ is $\mu_y$-strongly convex in $y$, i.e., $\bar{f}(x, y_1) \geq \bar{f}(x, y_2) + \nabla_y \bar{f}(x, y_2)^\top (y_1 - y_2) + \frac{\mu_y}{2}\|y_1 - y_2\|^2$, for all $(y_1, y_2)$.*

The assumption that the second level is strongly convex is prevalent throughout the bilevel optimization literature (e.g., see Ghadimi & Wang (2018); Ji et al. (2020); Liu et al. (2021); Chen et al. (2021; 2022b;a); Giovannelli et al. (2024; 2025)). In practice, $\bar{f}$ will be strongly convex when $f_2$ is strongly convex in $(y, z)$ and $z(x, y)$ is an affine function in $(x, y)$. Hence, assuming strong convexity of $\bar{f}$ covers cases where the LL problem is a QP problem or even certain special cases of polynomial functions of even order, such as the squared norm of a quadratic function (see equation F.16 in Subsection 4.2).

Next, as is standard in the stochastic approximation literature, we require that all stochastic estimates be unbiased with bounded variances and that all random variables that are sampled are independent and identically distributed, stated in Assumption 3.4 below. This ensures that the stochastic terms that are used to approximate the gradients, Hessians, Jacobians, and third-order tensors are reliable approximations of their corresponding deterministic counter-parts. In applications of empirical risk minimization like machine learning, such an assumption can easily be satisfied in practice by taking larger sample sizes when approximating these terms.

**Assumption 3.4 (Stochastic estimates)** *The stochastic derivatives $\nabla f_1^\xi$, $\nabla f_2^\xi$, $\nabla^2 f_2^\xi$, $\nabla f_3^\xi$, $\nabla^2 f_3^\xi$, and $\nabla^3 f_3^\xi$ are unbiased estimators of $\nabla f_1$, $\nabla f_2$, $\nabla^2 f_2$, $\nabla f_3$, $\nabla^2 f_3$, and $\nabla^3 f_3$, respectively. Further, the variances of the stochastic derivatives are bounded by constants $\sigma_{\nabla f_1}^2$, $\sigma_{\nabla f_2}^2$, $\sigma_{\nabla^2 f_2}^2$, $\sigma_{\nabla f_3}^2$, $\sigma_{\nabla^2 f_3}^2$, and $\sigma_{\nabla^3 f_3}^2$, respectively. Further, all of the random variables $\xi$ that are sampled are independent and idententically distributed (i.i.d.).*

Although Assumptions 3.2–3.3 ensure that the Hessian sub-matrices $\nabla_{zz}^2 f_3$ and $\nabla_{yy}^2 \bar{f}$ are bounded away from singularity, we also require that their stochastic estimates be bounded away from singularity, stated as Assumption 3.5 below, which ensures that these estimates provide a robust measure of the curvature of the functions $f_3$ and $\bar{f}$.

**Assumption 3.5 (Uniform bound on inverted stochastic Hessians)** *The stochastic principal sub-matrices $[\nabla^2_{zz} f_3^\xi]^{-1}$ and $[\nabla^2_{yy} \bar{f}^\xi]^{-1}$ are upper-bounded in norm at all points by the positive constants $b_{zz}$ and $b_{yy}$, respectively.*

In the stochastic gradient literature concerning second-order derivatives, it is common to assume that a Hessian matrix, stochastic or not, is uniformly bounded below Bollapragada et al. (2018), implying that its inverse is uniformly bounded above. The motivation is that, if the Hessian matrix is not uniformly bounded below, a regularization term can be added to such a matrix to ensure it is non-singular.

Lastly, Assumption 3.6 below is imposed to ensure that the bias of the inverted stochastic estimates $[\nabla^2_{zz} f_3^\xi]^{-1}$ and $[\nabla^2_{yy} \bar{f}^\xi]^{-1}$ approach zero on the order $\mathcal{O}(\theta)$. It is known that such an assumption can be satisfied in practice, e.g., by utilizing a truncated-Neumann series (see Ghadimi & Wang (2018)) and incrementally increasing the number of samples used when approximating the terms $\nabla^2_{zz} f_3$ and $\nabla^2_{yy} \bar{f}$ (the authors in Chen et al. (2021) utilize such a property to establish a similar bound; though they do not state it as an assumption, but instead leave the number of samples as a parameter in their analysis that they choose to yield their desired convergence result).

**Assumption 3.6 (Bounded bias of inverted stochastic Hessians)** *The stochastic principal sub-matrices $[\nabla^2_{zz} f_3^\xi]^{-1}$ and $[\nabla^2_{yy} \bar{f}^\xi]^{-1}$ are estimators of $[\nabla^2_{zz} f_3]^{-1}$ and $[\nabla^2_{yy} \bar{f}]^{-1}$, respectively, with biases that are bounded on the order of $\mathcal{O}(\theta)$, i.e., there exist positive constants $W_{zz}$ and $W_{yy}$ such that $\|[\nabla^2_{zz} f_3]^{-1} - \mathbb{E}[[\nabla^2_{zz} f_3^\xi]^{-1}|\mathcal{F}_\xi]\| \leq W_{zz}\theta$ and $\|[\nabla^2_{yy} \bar{f}]^{-1} - \mathbb{E}[[\nabla^2_{yy} \bar{f}^\xi]^{-1}|\mathcal{F}_\xi]\| \leq W_{yy}\theta$, respectively.*

## 3.2 Convergence of the TSG method

To derive our analysis of TLO methods, we introduce the following Lyapunov-type function

$$\mathbb{V}^i := f(x^i) + \|y^i - y(x^i)\|^2 + \|z^i - z(x^i)\|^2 + \|z^i - z(x^i, y^i)\|^2, \qquad (3.2)$$

which is telescopically summed over all iterates (see Appendix B). The first two terms in (3.2) were used in the analysis of Chen et al. (2021) for bilevel optimization. While carrying out our analysis, we realized that adding the third term was not enough for TLO, and the need for the fourth term arises from the inexact LL error relative to the ML variables. We now present an overview of the primary convergence result of Algorithm 3, in which we consider the general case where the true UL function $f$ is possibly nonconvex. Further, for the full description and proof of this theorem, see Theorem B.5 and Appendix C.5, respectively.

**Convergence of TSG – Nonconvex $f$ (Theorem B.5).** Under Assumptions 3.1–3.6, when choosing the step-sizes $\alpha_i$, $\beta_i$, and $\gamma_i$ to incorporate problem-specific information, a convergence rate of $\frac{1}{I}\sum_{i=0}^{I-1}\mathbb{E}[\|\nabla f(x^i)\|^2] = \mathcal{O}(1/\sqrt{I})$ can be obtained. This result does not require lower-bounds on the UL or ML variables $I$ and $J$, but requires $K \geq \mathcal{O}(J^4 I)$.

We state this theorem as our primary convergence result since it matches the tightest known bound derived for general nonconvex bilevel optimization problems under similar assumptions Chen et al. (2021). However, we also include another less-tight convergence rate of $\mathcal{O}(J/\sqrt{I})$ (see Theorem B.6) that provides a more intuitive choice of step-sizes that directly impact algorithmic implementability, requiring $K \geq \mathcal{O}(J^3 I)$. We highlight that the $J$ present in the numerator of this alternative rate can be thought of as the extra $J$ that is present in the iteration complexity on $K$ in Theorem B.5 (i.e., $K \geq \mathcal{O}((J \times J^3)I)$.

Notice that both Theorems B.5 and B.6 share a common constraint: the LL iterations $K$ must scale linearly in $I$ and polynomially in $J$. We argue that such a requirement follows intuitively, as the accuracy of the LL solution directly impacts the inexactness of the bilevel adjoint gradient for the ML problem. Further, this constraint reveals the hierarchical interplay within trilevel problems, i.e., more LL iterations are required to obtain a higher accuracy in the ML problem than in the UL problem. This implies that the trilevel adjoint gradient $\nabla f$ tolerates more inexactness from the ML problem than the bilevel adjoint gradient $\nabla_y \bar{f}$ does from the LL problem. Such a relationship underscores how errors in the LL propagate upward through the levels: greater accuracy at any sub-upper level necessitates significantly higher precision in the LL solution. Whether this pattern

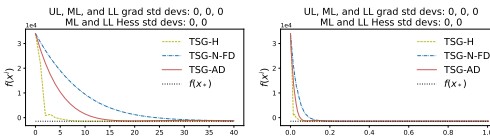
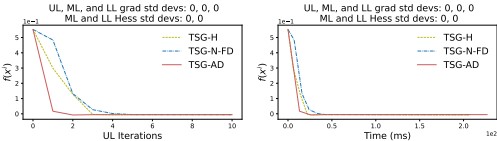

Figure 1: Quadratic problem, deterministic case.

Figure 2: Quartic problem, deterministic case.

extends to all sup-upper levels in general multi-level problems or entirely shifts the computational burden to the lowest level remains an open question for future research.

## 4 NUMERICAL EXPERIMENTS

The experimental results were obtained on a desktop workstation with 128GB of RAM, an Intel(R) Core(TM) i9-13950HX processor (24 cores, 32 threads) running at 2200 MHz under Windows 11.

### 4.1 OUR PRACTICAL TSG METHODS

A major difficulty in the adjoint gradient equation 2.2 is the need for second-order derivatives of $\bar{f}$ (a challenge that also arises in the adjoint gradient of a BLO problem), and, in particular, the presence of third-order derivative tensors in $\nabla_{yx}\bar{f}$ and $\nabla_{yy}\bar{f}$ in equation A.1 and equation A.2, due to equation A.5 and equation A.6, respectively. We consider two approaches to address this issue (see Appendix F.1), leading to two practical versions of the TSG method, referred to as TSG-N-FD and TSG-AD. In the numerical experiments, we are mainly interested in testing these two practical implementations (Algorithms 4 and 7 in Appendices F.2 and F.3, respectively) rather than the method we refer to as TSG-H, which uses the true Hessians and third-order derivative tensors (Algorithm 3 in Section 2).

The first algorithm we propose, TSG-N-FD, is based on the adjoint equation approach and involves solving any adjoint system arising in equation 2.2 and equation F.2 by using the linear CG method, where each Hessian-vector product is approximated via a finite-difference (FD) scheme. When using TSG-H, we will apply the linear CG method to solve any adjoint system arising in equation 2.2 and equation F.2 until non-positive curvature is detected. The second algorithm we propose, TSG-AD, is based on the truncated Neumann series approach and consists of approximating each Hessian-vector product by using automatic differentiation (AD). Note that TSG-H is not suited for practical optimization problems, but we include it in the experiments for completeness. For very large problems, one must use TSG-N-FD or TSG-AD.

To determine the ML and LL iterations $J$ and $K$, we used an increasing accuracy strategy inspired by Giovannelli et al. (2025): the number of ML iterations increases by one when the change in $f_1$ between two consecutive UL iterations drops below $10^{-2}$, and the number of LL iterations increases similarly when the change in $f_2$ between two consecutive ML iterations drops below $10^{-1}$.

### 4.2 NUMERICAL RESULTS FOR SYNTHETIC TRILEVEL PROBLEMS

We first report results for two synthetic trilevel problems that differ in their LL problem formulations (see Appendix F.4). In the first, all levels have quadratic objective functions, leading to a quadratic trilevel problem (with zero third-order derivatives). In the second, the UL and ML objective functions are quadratic, while the LL objective is quartic (resulting in non-zero third-order derivatives). For simplicity, we refer to the second trilevel problem as quartic.

Figures 1, 2, 3, and 4 compare the sequences of $f(x^i)$ values obtained by TSG-H, TSG-N-FD, and TSG-AD over UL iterations and running time. In the stochastic case, we computed the stochastic gradients and Hessians by adding Gaussian noise with mean zero to the corresponding deterministic quantities. We did not add noise to the third-order tensors, as these are not used in the practical algorithms TSG-N-FD and TSG-AD. All figures involving stochasticity include 95% confidence intervals computed using the t-distribution over 10 runs.

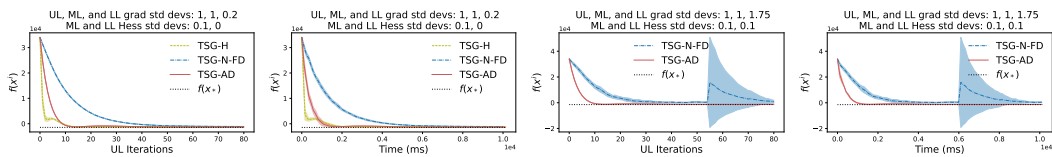

Figure 3: Quadratic problem, stochastic case (low noise: two left plots; high noise: two right plots).

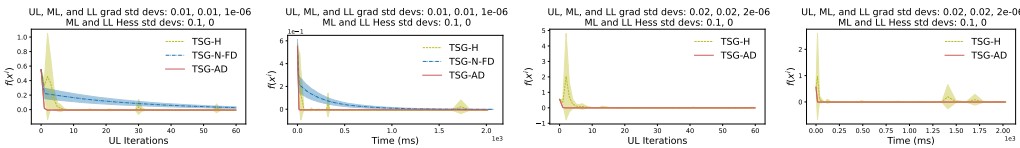

Figure 4: Quartic problem, stochastic case (low noise: two left plots; high noise: two right plots).

For the quadratic problem, Figure 1 shows that TSG-H, which uses Hessians and third-order tensors, outperforms TSG-N-FD and TSG-AD in terms of both UL iterations and time in the deterministic case. Figure 3 shows the plots for the stochastic case. Note that TSG-N-FD and TSG-AD are not affected by the noise in the Hessians of $f_2$ and $f_3$, as they rely only on first-order derivatives. TSG-H is highly sensitive to the standard deviation of the Hessian of $f_3$ (which appears in the trilevel adjoint gradient equation 2.2), and its performance deteriorates significantly when this value exceeds $0.1$. Such behavior aligns with the well-known fact that stochastic Hessians require lower noise levels (i.e., larger mini-batch sizes when noise arises from sampling finite-sum Hessians in SG contexts) than stochastic gradients to perform well (Bottou et al., 2018, Section 6.1.1). For this reason, we omit TSG-H from the two right plots. As noise levels increase, the performance of TSG-N-FD deteriorates, whereas TSG-AD remains more robust. The most critical source of noise for TSG-N-FD is that added to $\nabla f_3$, which is used to approximate the matrix-vector products involving the Hessian of $f_3$ via the FD scheme in equation F.5. Note that such an FD scheme affects the computation of both equation F.3 and equation F.4.

For the quartic problem, in the deterministic case, Figure 2 shows that TSG-H is the least competitive algorithm in terms of time, as the computation of third-order tensors slows it down. In the stochastic case, shown in Figure 4, increasing noise levels lead to performance deterioration for both TSG-N-FD and TSG-H, whereas TSG-AD remains the most robust. We can conclude that when third-order derivatives are non-zero, the FD approximations used in TSG-N-FD become less accurate.

### 4.3 NUMERICAL RESULTS FOR TRILEVEL HYPERPARAMETER ADVERSARIAL TUNING

In the TLO formulation we propose for hyperparameter adversarial tuning (see problem F.18 in Appendix F.5 for the rigorous formulation), the UL problem minimizes the validation loss over a regularization parameter used in the training loss, the ML problem minimizes the training loss over the model parameters, and the LL problem is posed on the variables that perturb the data in a worst-case fashion. In the formulation proposed in Sato et al. (2021), the ML and LL problems are swapped compared to our formulation in equation F.18. We adopt equation F.18 because it more accurately reflects the original minimax formulation for adversarial training equation F.17, and indeed leads to improved performance (see Appendix F.5.1). We will also evaluate BLO formulations obtained by removing either the UL or LL problem from equation F.18. Removing the UL problem yields a BLO problem similar in spirit to the original minimax formulation of adversarial learning, while removing the LL problem gives a BLO problem for hyperparameter tuning without adversarial learning.

The BLO problems obtained from equation F.18 are solved using corresponding bilevel algorithms (denoted as BSG-AD) derived from TSG-AD. Such algorithms are essentially equivalent to the well-known StocBiO Ji et al. (2020). In this section, we will not test TSG-H, as it requires second and third-order derivatives, which are impractical to compute in applications involving large-scale datasets. Similarly, we will not test the trilevel algorithm proposed in Sato et al. (2021), as it is designed specifically for the deterministic setting. When using equation F.18, TSG-N-FD does not perform well and is therefore excluded from further analysis (see Appendix F.5.1 for a discussion).

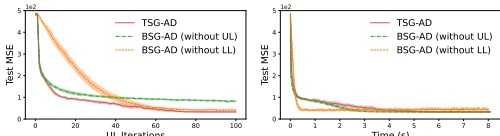
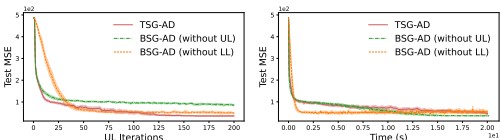

Figure 5: Adversarial learning formulation equation F.18, red wine quality dataset.

Figure 6: Adversarial learning formulation equation F.18, white wine quality dataset.

For the experiments, we consider three popular tabular datasets: the red and white wine quality datasets Cortez et al. (2009) and the California housing dataset Pace & Barry (1997). To assess the performance of the algorithms and formulations on these datasets, we compute the test MSE after adding Gaussian noise (with a standard deviation of 5) to the features of the test data, averaged over 100 realizations of the noise. The optimal solution obtained from the trilevel formulation equation F.18 is expected to yield a model robust to such noise.

The results for our TLO formulation in equation F.18, along with those for the BLO formulations obtained by removing the UL and LL problems from equation F.18, are shown in Figures 5–7. The TLO formulation in equation F.18 proves to be the most consistently effective for hyperparameter adversarial tuning, with the BLO variants demonstrating competitive runtime but greater sensitivity to the nature of the dataset, reflected in the contrasting dependencies observed across the datasets. In fact, the superior performance of BSG-AD (without LL) over BSG-AD (without UL) on the red and white wine datasets is an indication of the reliance of these datasets on hyperparameter tuning, whereas the inverted performance of the BSG algorithms on the California housing dataset is a symptom of this dataset's dependence on adversarial learning. Overall, TSG-AD, which leverages both adversarial and hyperparameter tuning components during model training, consistently yields the most robust performance across all the tested datasets and will likely deliver further performance improvements in settings where both components are jointly critical.

## 5 CONCLUSION

In this paper, we proposed the first stochastic first-order method for trilevel optimization along with a rigorous convergence theory for the nonconvex setting. The proposed theory also covers all forms of inexactness that arise within the trilevel adjoint gradient, such as the inexact solutions of the middle and lower-level problems, inexact computation of the trilevel adjoint formula, and noisy estimates of the gradients, Hessians, Jacobians, and tensors of third-order derivatives involved. Our experi-

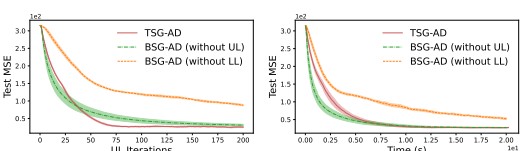

Figure 7: Adversarial learning formulation equation F.18, California housing dataset.

ments demonstrate that the proposed TLO formulation can be more robust than the BLO formulations corresponding to its UL and ML (i.e., hyperparameter tuning without adversarial learning), or its ML and LL (i.e., the original minimax adversarial training), as well as the TLO formulation in Sato et al. (2021), where the ML and LL are swapped compared to ours. A natural direction left for future research lies in thoroughly exploring how the accuracy at any given intermediate level relates to the precision required at lower levels within general multi-level optimization problems. Specifically, such an investigation would seek to clarify whether increasing the accuracy at a particular level necessitates higher precision at all subsequent lower levels, or if the computational burden entirely shifts to the lowest level. Following directions similar to those emerging in the BLO literature, an additional avenue for future work is to relax the strong convexity assumptions on the ML and LL objective functions by exploring penalization techniques that allow for non-convex objectives at lower levels.

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

## A    DERIVATION OF THE TRILEVEL ADJOINT GRADIENT

This appendix contains the formal statement and derivation of the trilevel adjoint gradient given by equation equation 2.2.

**Proposition A.1 (Trilevel adjoint gradient)** *Under assumptions that will ensure all terms are well-defined (specifically, Assumptions 3.1–3.3), we define the adjoint gradient of $f$ as (referenced as equation 2.2)*

$$\nabla f \;=\; (\nabla_x f_1 - \nabla^2_{xz} f_3 \nabla^2_{zz} f_3^{-1} \nabla_z f_1) - \nabla^2_{xy} \bar{f} \nabla^2_{yy} \bar{f}^{-1} (\nabla_y f_1 - \nabla^2_{yz} f_3 \nabla^2_{zz} f_3^{-1} \nabla_z f_1),$$

*where all of the gradient and Hessian terms of $f_1$ and $f_3$ on the right-hand side are evaluated at $(x, y(x), z(x))$. Further, the $\bar{f}$ terms are evaluated at $(x, y(x))$ where*

$$\nabla^2_{yx} \bar{f}(x, y) \;=\; \nabla^2_{yx} f_2 + \nabla^2_{yz} f_2 \nabla_x z^\top + \frac{\partial}{\partial x}\left[\nabla_y z \nabla_z f_2\right], \tag{A.1}$$

$$\nabla^2_{yy} \bar{f}(x, y) \;=\; \nabla^2_{yy} f_2 + \nabla^2_{yz} f_2 \nabla_y z^\top + \frac{\partial}{\partial y}\left[\nabla_y z \nabla_z f_2\right], \tag{A.2}$$

*with*

$$\nabla_x z(x, y)^\top \;=\; -\nabla^2_{zz} f_3^{-1} \nabla^2_{zx} f_3, \tag{A.3}$$

$$\nabla_y z(x, y)^\top \;=\; -\nabla^2_{zz} f_3^{-1} \nabla^2_{zy} f_3, \tag{A.4}$$

$$\frac{\partial}{\partial x}[\nabla_y z \nabla_z f_2] \;=\; -[\nabla^3_{yzx} f_3 + \nabla^3_{yzz} f_3 \nabla_x z^\top - \nabla^2_{yz} f_3 \nabla^2_{zz} f_3^{-1}(\nabla^3_{zzx} f_3 + \nabla^3_{zzz} f_3 \nabla_x z^\top)]\nabla^2_{zz} f_3^{-1} \nabla_z f_2$$
$$- \nabla^2_{yz} f_3 \nabla^2_{zz} f_3^{-1}(\nabla^2_{zx} f_2 + \nabla^2_{zz} f_2 \nabla_x z^\top), \tag{A.5}$$

$$\frac{\partial}{\partial y}[\nabla_y z \nabla_z f_2] \;=\; -[\nabla^3_{yzy} f_3 + \nabla^3_{yzz} f_3 \nabla_y z^\top - \nabla^2_{yz} f_3 \nabla^2_{zz} f_3^{-1}(\nabla^3_{zzy} f_3 + \nabla^3_{zzz} f_3 \nabla_y z^\top)]\nabla^2_{zz} f_3^{-1} \nabla_z f_2$$
$$- \nabla^2_{yz} f_3 \nabla^2_{zz} f_3^{-1}(\nabla^2_{zy} f_2 + \nabla^2_{zz} f_2 \nabla_y z^\top). \tag{A.6}$$

*Notice that all of the gradients and Hessians of $f_2$ and the gradients, Hessians, and tensors of third-order derivatives (which we denote by $\nabla^3$)* of $f_3$ in equation A.1–equation A.6 are evaluated at the point $(x, y, z(x, y))$ and all of the $\nabla z$ terms are evaluated at the point $(x, y)$.*

**Proof.** One arrives at the adjoint formula equation 2.2 by first applying the multivariate chain rule to $f_1(x, y(x), z(x, y(x)))$ in the following manner:

$$\nabla f \;=\; \frac{d}{dx} f_1(x, y(x), z(x, y(x))) \;=\; \frac{\partial f_1}{\partial x} + \frac{dy}{dx}\frac{\partial f_1}{\partial y} + \frac{d}{dx} z(x, y(x))\frac{\partial f_1}{\partial z},$$

---

*To clarify the notation for third-order derivatives, consider the following example: given an $m \times t \times n$ tensor $\nabla^3_{yzx} f_3$ and a $t \times t$ matrix $\nabla^2_{zz} f_3^{-1}$, the product $\nabla^3_{yzx} f_3 \nabla^2_{zz} f_3^{-1}$ yields an $m \times t \times n$ matrix. Left-multiplying a $t$-dimensional vector $\nabla_z f_2$ by $\nabla^3_{yzx} f_3 \nabla^2_{zz} f_3^{-1}$ results in an $m \times 1 \times n$ matrix (or $m \times n$, for brevity).

where

$$\frac{d}{dx}z(x, y(x)) \;=\; \frac{\partial z}{\partial x} + \frac{dy}{dx}\frac{\partial z}{\partial y}.$$

Thus, we have

$$
\begin{aligned}
\nabla f \;&=\; \frac{\partial f_1}{\partial x} + \frac{dy}{dx}\frac{\partial f_1}{\partial y} + \big(\frac{\partial z}{\partial x} + \frac{dy}{dx}\frac{\partial z}{\partial y}\big)\frac{\partial f_1}{\partial z} \\
&=\; \nabla_x f_1 + \nabla y \nabla_y f_1 + (\nabla_x z + \nabla y \nabla_y z)\,\nabla_z f_1 \\
&=\; \nabla_x f_1 + \nabla_x z \nabla_z f_1 + \nabla y\,(\nabla_y f_1 + \nabla_y z \nabla_z f_1)\,.
\end{aligned}
\tag{A.7}
$$

The Jacobian of $y(x)$, i.e., $\nabla y(x)^\top \in \mathbb{R}^{m\times n}$, can be computed from the first-order necessary optimality conditions of the ML problem, defined by $\nabla_y \bar{f}(x, y(x)) = 0$. In particular, taking the derivative of both sides with respect to $x$, utilizing the chain rule and the implicit function theorem (which ensures $y(\cdot)$ to be continuously differentiable Rudin (1953)), we obtain $\nabla^2_{yx}\bar{f} + \nabla^2_{yy}\bar{f}\nabla y(x)^\top = 0$ (where all Hessians are evaluated at $(x, y(x))$), which yields

$$\nabla y(x)^\top \;=\; -\nabla^2_{yy}\bar{f}(x, y(x))^{-1}\nabla^2_{yx}\bar{f}(x, y(x)).\tag{A.8}$$

Since

$$\nabla_y \bar{f}(x, y) = \nabla_y f_2(x, y, z(x, y)) + \nabla_y z \nabla_z f_2(x, y, z(x, y)),\tag{A.9}$$

taking the derivative of both sides with respect to $x$ and $y$ and utilizing the chain rule, we obtain the expressions for $\nabla^2_{yx}\bar{f}(x, y)$ and $\nabla^2_{yy}\bar{f}(x, y)$ in equation A.1 and equation A.2, respectively.

Similarly, we can derive expressions for both of the Jacobians of $z(x, y)$, i.e., $\nabla_x z(x, y)^\top \in \mathbb{R}^{t\times n}$ and $\nabla_y z(x, y)^\top \in \mathbb{R}^{t\times m}$, respectively, from the first-order necessary optimality conditions of the LL problem, defined by $\nabla_z f_3(x, y, z(x, y)) = 0$. In particular, taking derivatives of both sides with respect to $x$ will yield $\nabla^2_{zx}f_3(x, y, z(x, y)) + \nabla^2_{zz}f_3(x, y, z(x, y))\nabla_x z(x, y)^\top = 0$, whereas taking derivatives with respect to $y$ will yield $\nabla^2_{zy}f_3(x, y, z(x, y)) + \nabla^2_{zz}f_3(x, y, z(x, y))\nabla_y z(x, y)^\top = 0$. Solving these two equations for both $\nabla_x z(x, y)^\top$ and $\nabla_y z(x, y)^\top$, respectively, we obtain the expressions for $\nabla_x z(x, y)^\top$ and $\nabla_y z(x, y)^\top$ in equation A.3 and equation A.4, respectively. Now, substituting equation A.8, equation A.3, and equation A.4 into equation A.7, we obtain the adjoint gradient defined by equation 2.2.

It remains to derive equation A.5 and equation A.6. Using the property that the derivative of the inverse of a matrix $K\,(g(x))$ with respect to $x$, where $g$ is a vector-valued function of $x$, is given by

$$\frac{\partial}{\partial x}K\,(g(x))^{-1} = -K\,(g(x))^{-1}\left[\frac{\partial}{\partial x}K\,(g(x))\right]K\,(g(x))^{-1},$$

and applying the product rule twice, it follows that the last term in equation A.1 can be written as

$$
\begin{aligned}
\frac{\partial}{\partial x}\left[\nabla_y z \nabla_z f_2\right] \;&=\; \frac{\partial}{\partial x}\left[-\nabla^2_{yz}f_3 \nabla^2_{zz}f_3^{-1}\nabla_z f_2\right] \\
&=\; -\left(\frac{\partial}{\partial x}\nabla^2_{yz}f_3\right)\nabla^2_{zz}f_3^{-1}\nabla_z f_2 - \nabla^2_{yz}f_3\left(\frac{\partial}{\partial x}\nabla^2_{zz}f_3^{-1}\nabla_z f_2\right) \\
&=\; -\left(\frac{\partial}{\partial x}\nabla^2_{yz}f_3\right)\nabla^2_{zz}f_3^{-1}\nabla_z f_2 - \nabla^2_{yz}f_3\left[\left(\frac{\partial}{\partial x}\nabla^2_{zz}f_3^{-1}\right)\nabla_z f_2 + \nabla^2_{zz}f_3^{-1}\left(\frac{\partial}{\partial x}\nabla_z f_2\right)\right],
\end{aligned}
\tag{A.10}
$$

where

$$
\begin{aligned}
\frac{\partial}{\partial x}\nabla^2_{yz}f_3 \;&=\; \nabla^3_{yzx}f_3 + \nabla^3_{yzz}f_3 \nabla_x z(x, y)^\top, & \frac{\partial}{\partial x}\nabla_z f_2 \;&=\; \nabla^2_{zx}f_2 + \nabla^2_{zz}f_2 \nabla_x z(x, y)^\top, \\
\frac{\partial}{\partial x}\nabla^2_{zz}f_3^{-1} \;&=\; -\nabla^2_{zz}f_3^{-1}\left(\frac{\partial}{\partial x}\nabla^2_{zz}f_3\right)\nabla^2_{zz}f_3^{-1}, & \frac{\partial}{\partial x}\nabla^2_{zz}f_3 \;&=\; \nabla^3_{zzx}f_3 + \nabla^3_{zzz}f_3 \nabla_x z(x, y)^\top.
\end{aligned}
\tag{A.11}
$$

Substituting these equations into equation A.10 and simplifying, we obtain equation A.5. Through a similar process, we obtain that the right-most term of equation A.2 is given by equation A.6. $\qquad\square$

## B  Discussion on the convergence analysis of the TSG method

In this appendix, we outline the convergence analysis for the TSG method (Algorithm 3) and highlight all of the relevant results and notation involved. For simplicity of the convergence analysis, we utilize the Lyapunov function 3.2, which we restate here for ease of reference:

$$\mathbb{V}^i := f(x^i) + \|y^i - y(x^i)\|^2 + \|z^i - z(x^i)\|^2 + \|z^i - z(x^i, y^i)\|^2.$$

There is no particular property that is required from Lyapunov functions for our analysis. Rather, equation 3.2 is defined to allow for appropriate telescoping cancellations in the proofs of Theorems B.5–B.6 (which is an extension of the methodology utilized in Chen et al. (2021) for bilevel problems). Further, the difference between two consecutive Lyapunov evaluations can by quantified as

$$
\mathbb{V}^{i+1} - \mathbb{V}^i
$$
$$
= \underbrace{f(x^{i+1}) - f(x^i)}_{\text{Lemma B.1}} + \underbrace{\|y^{i+1} - y(x^{i+1})\|^2 - \|y^i - y(x^i)\|^2}_{\text{Lemma B.4}}
$$
$$
+ \underbrace{\|z^{i+1} - z(x^{i+1})\|^2 - \|z^i - z(x^i)\|^2}_{\text{Lemma B.2}} + \underbrace{\|z^{i+1} - z(x^{i+1}, y^{i+1})\|^2 - \|z^i - z(x^i, y^i)\|^2}_{\text{Lemma B.3}}.
$$
$$(B.1)$$

Notice that this consists of four differences: the first difference measures the amount of descent that is achieved in the UL problem, the second and third differences correspond to the error present in the ML and LL problems, respectively, and the fourth difference is an auxiliary term that corresponds to the inexact LL error relative to the ML variables. Further, it bears mentioning that Appendix C contains the proofs of Lemmas B.1–B.4 and Theorems B.5–B.6, and Appendix D contains the statements and proofs of intermediary results that are required for the arguments used in Appendix C. Lastly, Appendix E includes auxiliary lemmas proving Lipschitz continuity properties for the following functions, gradients, and Jacobians: $z(x)$, $z(x,y)$, $y(x)$, $\nabla_y \bar{f}$, $\nabla^2_{xy} \bar{f}$, $\nabla^2_{yy} \bar{f}$, $\nabla f$, $\nabla z$, and $\nabla y$. For ease of reference, Table 1 below compiles all the relevant constants utilized throughout the theory which are not defined in Lemmas B.1–B.4.

Table 1: Reference table of constants associated with derived bounds.

| Descriptions | Constants | References |
|---|---|---|
| Bounds on bias & variance | $U_x, U_y, U_{xy}, U_{yy}, V_{xy}, V_{yy}$ | Lemmas D.1 – D.2 |
| Bounds on UL inexactness | $\omega, \tau, \zeta$ | Lemmas D.3 – D.4 |
| Bounds on ML inexactness | $\hat{\omega}, \hat{\tau}, \Upsilon$ | Lemmas D.5 – D.6 |
| Derived Lipschitz properties | $L_z, L_{z_{xy}}, L_{z_y}, L_y, L_{\nabla_z}, L_{\bar{F}}, L_{\bar{F}_y}$ $L_{\bar{F}_z}, L_{\nabla^2_{yx}\bar{f}}, L_{\nabla^2_{yy}\bar{f}}, L_F, L_{F_{yz}}, L_{\nabla y}$ | Equations E.1 – E.13 |

### B.1  Descriptions of $\sigma$-algebras

We denote three auxiliary sets $\Sigma_i$, $\Sigma_{i,j}$, and $\Sigma_{i,j,k}$, each corresponding to the set of iterates generated by Algorithm 3 for the UL update, ML update, and LL update, respectively. We define these sets explicitly in the following way:

$$\Sigma_i := \{x^{\hat{i}}, y^{\hat{i}}, z^{\hat{i}} \mid \forall \hat{i} \in \{0, 1, ..., i\}\},$$

$$\Sigma_{i,j} := \{x^{\hat{i}}, y^{\hat{i},\hat{j}}, z^{\hat{i},\hat{j}} \mid \forall \hat{i} \in \{0, 1, ..., i\} \text{ and } \forall \hat{j} \in \{0, 1, ..., j\}\},$$

$$\Sigma_{i,j,k} := \{x^{\hat{i}}, y^{\hat{i},\hat{j}}, z^{\hat{i},\hat{j},\hat{k}} \mid \forall \hat{i} \in \{0, 1, ..., i\} \text{ and } \forall \hat{j} \in \{0, 1, ..., j\} \text{ and } \forall \hat{k} \in \{0, 1, ..., k\}\}.$$

Now, we define the corresponding $\sigma$-algebras generated as $\mathcal{F}_i := \sigma\left(\Sigma_i \cup \{y^{i+1}, z^{i+1}\}\right)$, $\mathcal{F}_{i,j} := \sigma\left(\Sigma_{i,j} \cup \{z^{i,j+1}\}\right)$, and $\mathcal{F}_{i,j,k} := \sigma\left(\Sigma_{i,j,k}\right)$, respectively. Further, we will use the expressions $\mathbb{E}\left[\cdot | \mathcal{F}_i\right]$, $\mathbb{E}\left[\cdot | \mathcal{F}_{i,j}\right]$, and $\mathbb{E}\left[\cdot | \mathcal{F}_{i,j,k}\right]$ to denote the conditional expectations taken with respect to the probability distributions of $\xi^i$, $\xi^{i,j}$, and $\xi^{i,j,k}$ given $\mathcal{F}_i$, $\mathcal{F}_{i,j}$, and $\mathcal{F}_{i,j,k}$, respectively. Recalling from the beginning of Section 3, we also define a general sigma-algebra $\mathcal{F}_\xi$ that includes all the

events up to the generation of a general point $(x, y, z)$, before observing a realization of $\xi$; similarly, $\mathbb{E}[\cdot|\mathcal{F}_\xi]$ denotes the expectation taken with respect to the probability distribution of $\xi$ given $\mathcal{F}_\xi$. We also use $\mathbb{E}[\cdot]$ to denote the *total expectation*, i.e., the expected value with respect to the joint distribution of all the random variables.

## B.2 STATEMENTS OF DESCENT AND ERROR BOUND RESULTS

We now provide the statements of Lemmas B.1–B.4 and Theorems B.5–B.6 below, that bound the terms in the Lyapunov difference given by equation B.1, and which are ultimately required to prove the main convergence results of Algorithm 3, presented in Theorems B.5–B.6. The proofs of such lemmas and theorems are provided in Appendix C. They required a non-trivial adaptation of the proofs in Chen et al. (2021), which were specific for bilevel problems.

**Lemma B.1 (Descent of the true trilevel UL problem)** *Recalling* $\bar{g}_{f_1}^i = \mathbb{E}[\tilde{g}_{f_1}^i|\mathcal{F}_i]$, *under Assumptions 3.1–3.6, the sequence of iterates* $\{x^i\}_{i\geq 0}$ *generated by Algorithm 3 satisfies*

$$
\mathbb{E}[f(x^{i+1})] - \mathbb{E}[f(x^i)] \leq -\frac{\alpha_i}{2}\mathbb{E}[\|\nabla f(x^i)\|^2] - \left(\frac{\alpha_i}{2} - \frac{L_F \alpha_i^2}{2}\right)\mathbb{E}[\|\bar{g}_{f_1}^i\|^2] + \tilde{\omega}\alpha_i^2
$$
$$
+ \alpha_i L_{F_{yz}}^2 (\mathbb{E}[\|y(x^i) - y^{i+1}\|^2] + \mathbb{E}[\|z(x^i) - z^{i+1}\|^2]), \quad \text{(B.2)}
$$

*where* $\tilde{\omega}$ *is given by equation C.1 in Appendix C.1.*

**Lemma B.2 (Error bounds of the trilevel LL problem)** *Suppose that Assumptions 3.1–3.6 hold. Then, choosing the LL step-size* $\gamma_i$ *such that* $\gamma_i \leq \frac{1}{\mu_z + L_{\nabla f_3}}$, *there exists the positive constant* $\rho_{f_3}$, *given by equation C.3 in Appendix C.2, and a positive quantity* $\kappa_i$, *such that*

$$
\mathbb{E}[\|z^{i,j+1} - z(x^i)\|^2] \leq (1 - \gamma_i \rho_{f_3})^K \mathbb{E}[\|z^{i,j} - z(x^i)\|^2] + K\gamma_i^2 \sigma_{\nabla f_3}^2, \quad \text{(B.3)}
$$
$$
\mathbb{E}[\|z^{i+1} - z(x^i)\|^2] \leq (1 - \gamma_i \rho_{f_3})^{JK} \mathbb{E}[\|z^i - z(x^i)\|^2] + JK\gamma_i^2 \sigma_{\nabla f_3}^2, \quad \text{(B.4)}
$$
$$
\mathbb{E}[\|z^{i+1} - z(x^{i+1})\|^2] \leq \left(1 + 2\kappa_i + \frac{L_{\nabla z}\alpha_i^2 \zeta}{2}\right)\mathbb{E}[\|z^{i+1} - z(x^i)\|^2]
$$
$$
+ \left(2L_z^2 + \frac{L_z^2}{2\kappa_i} + \frac{L_{\nabla z}}{2}\right)\alpha_i^2 \mathbb{E}[\|\bar{g}_{f_1}^i\|^2] + \left(2L_z^2 + \frac{L_{\nabla z}}{2}\right)\tau\alpha_i^2. \quad \text{(B.5)}
$$

**Lemma B.3 (Auxiliary error bounds of the trilevel LL problem)** *Suppose that Assumptions 3.1–3.6 hold. Then, choosing the LL step-size* $\gamma_i$ *such that* $\gamma_i \leq \frac{1}{\mu_z + L_{\nabla f_3}}$, *there exist positive quantities* $\eta_i$ *and* $\hat{\eta}_i$ *such that*

$$
\mathbb{E}[\|z^{i,j+1} - z(x^i, y^{i,j+1})\|^2] \leq ((1 - \gamma_i \rho_{f_3})^K + \eta_i)\mathbb{E}[\|z^{i,j} - z(x^i, y^{i,j})\|^2] + \hat{\eta}_i L_{z_y}^2 \Upsilon \beta_i^2 + K\gamma_i^2 \sigma_{\nabla f_3}^2,
$$
$$
\text{(B.6)}
$$
$$
\mathbb{E}[\|z^{i,j+1} - z(x^i, y^i)\|^2] \leq (1 - \gamma_i \rho_{f_3})^K \mathbb{E}[\|z^i - z(x^i, y^i)\|^2] + K\gamma_i^2 \sigma_{\nabla f_3}^2, \quad \text{(B.7)}
$$
$$
\mathbb{E}[\|z^{i+1} - z(x^i, y^i)\|^2] \leq (1 - \gamma_i \rho_{f_3})^{JK} \mathbb{E}[\|z^i - z(x^i, y^i)\|^2] + JK\gamma_i^2 \sigma_{\nabla f_3}^2, \quad \text{(B.8)}
$$
$$
\mathbb{E}[\|z^{i+1} - z(x^{i+1}, y^{i+1})\|^2] \leq 2\mathbb{E}[\|z^{i+1} - z(x^i, y^i)\|^2] + 4L_{z_{xy}}^2 \alpha_i^2 (\mathbb{E}[\|\bar{g}_{f_1}^i\|^2] + \tau) + 2J^2 \Upsilon L_{z_{xy}}^2 \beta_i^2,
$$
$$
\text{(B.9)}
$$
$$
\mathbb{E}[\|z^{i,j+1} - z(x^i, y^{i,j})\|^2] \leq (1 - \gamma_i \rho_{f_3})^K \mathbb{E}[\|z^{i,j} - z(x^i, y^{i,j})\|^2] + K\gamma_i^2 \sigma_{\nabla f_3}^2. \quad \text{(B.10)}
$$

**Lemma B.4 (Error bounds of the trilevel ML problem)** *Suppose that Assumptions 3.1–3.6 hold. Then, choosing the ML step-size* $\beta_i$ *such that* $\beta_i \leq \frac{1}{\mu_y + L_{\nabla f}}$ *and* $\beta_i \leq \frac{\rho}{2\tilde{\omega}^2+1}$ *as well as choosing*

the LL step-size $\gamma_i$ such that $\gamma_i \leq \frac{1}{\mu_z + L_{\nabla f_3}}$, there are positive quantities $\psi_i$ and $\phi_i$ such that

$$
\begin{aligned}
\mathbb{E}[\|y^{i+1} - y(x^i)\|^2] \quad &\leq \quad (1 - \psi_i \beta_i)^J \mathbb{E}[\|y^i - y(x^i)\|^2] + \left(1 + \frac{1}{2}(J-1)\hat{\eta}_i L_{z_y}^2\right) J \Upsilon \beta_i^2 \\
&\quad + (1 - \gamma_i \rho_{f_3})^K J \mathbb{E}[\|z^i - z(x^i, y^i)\|^2] + \frac{J+1}{2} JK \gamma_i^2 \sigma_{\nabla f_3}^2, \quad \text{(B.11)}
\end{aligned}
$$

$$
\begin{aligned}
\mathbb{E}[\|y^{i+1} - y(x^{i+1})\|^2] &\leq \left(1 + 2\phi_i + \frac{L_{\nabla y} \alpha_i^2 \zeta}{2}\right) \mathbb{E}[\|y^{i+1} - y(x^i)\|^2] \\
&\quad + \left(2L_y^2 + \frac{L_y^2}{2\phi_i} + \frac{L_{\nabla y}}{2}\right) \alpha_i^2 \mathbb{E}[\|\bar{g}_{f_1}^i\|^2] + \left(2L_y^2 + \frac{L_{\nabla y}}{2}\right)\tau \alpha_i^2, \quad \text{(B.12)}
\end{aligned}
$$

where $\rho$ is given by equation C.7 in Appendix C.4. Specifically, $\psi_i$ is a function of $\theta_i$ given by equation C.7 in Appendix C.4.

**Theorem B.5 (V1. Convergence of TSG – Nonconvex $f$)** *Under Assumptions 3.1–3.6, define the constants*

$$
\bar{\alpha}_1 = \min \left\{1, \frac{1}{2(L_F + 4L_y^2 + L_{\nabla y} + 4L_z^2 + L_{\nabla z} + 8L_{z_{xy}}^2)}\right\},
$$

$$
\bar{\alpha}_2 = \frac{2J\Gamma \bar{\beta}_1}{2L_{F_{yz}}^2 + 16L_y^2 + L_{\nabla y}\bar{\alpha}_1 \zeta}, \qquad \bar{\alpha}_3 = J\sqrt{K}\bar{\gamma}_1,
$$

$$
\text{with} \qquad \bar{\beta}_1 = \min\left\{1, \frac{1}{\mu_y + L_{\nabla f}}, \frac{\rho}{2\hat{\omega}^2 + 1}\right\} \qquad \text{and} \qquad \bar{\gamma}_1 := \min\left\{1, \frac{1}{\mu_z + L_{\nabla f_3}}\right\},
$$

*where $\Gamma$ is a positive constant given by equation C.39 in Appendix C.5 and $\rho_{f_3}$ was introduced in Lemma B.2. Choose the step-sizes*

$$
\alpha_i = \min\left\{\bar{\alpha}_1, \bar{\alpha}_2, \bar{\alpha}_3, \frac{\alpha_0}{\sqrt{I}}\right\}, \qquad \beta_i = \frac{2L_{F_{yz}}^2 + 16L_y^2 + L_{\nabla y}\bar{\alpha}_1 \zeta}{2J\Gamma}\alpha_i, \qquad \gamma_i = \frac{\varrho(J, K)}{J\sqrt{K}}\alpha_i,
$$

*where $\varrho : (J, K) \to \mathbb{R}_+$ is defined by equation C.46 in Appendix C.5. Then, for any $I \in \{1, 2, ...\}$, $J \in \{1, 2, ...\}$, and $K \geq \mathcal{O}\left(J^4 I\right)$ as defined by equation C.59, the iterates $\{x^i\}_{i \geq 0}$ generated by Algorithm 3 satisfy*

$$
\frac{1}{I}\sum_{i=0}^{I-1} \mathbb{E}[\|\nabla f(x^i)\|^2] = \mathcal{O}\left(\frac{1}{\sqrt{I}}\right).
$$

**Theorem B.6 (V2. Convergence of TSG – Nonconvex $f$)** *Under Assumptions 3.1–3.6, choose the step-sizes*

$$
\alpha_i = \frac{1}{\sqrt{I}}, \qquad \beta_i = \frac{1}{\sqrt{J}}\alpha_i, \qquad \gamma_i = \frac{1}{\sqrt{J}\sqrt{K}}\alpha_i.
$$

*Then, the iterates $\{x^i\}_{i \geq 0}$ generated by Algorithm 3 satisfy*

$$
\frac{1}{I}\sum_{i=0}^{I-1} \mathbb{E}[\|\nabla f(x^i)\|^2] = \mathcal{O}\left(\frac{J}{\sqrt{I}}\right),
$$

*when choosing any $I \in \mathbb{N}_+$, $J \in \mathbb{N}_+$, and $K \in \mathbb{N}_+$ such that*

$$
\varsigma \leq J, \qquad \varpi \leq I, \qquad \Xi(I, J) = \mathcal{O}(J^3 I) \leq K,
$$

*where $\varsigma \in \mathbb{R}_+$ is defined by equation C.97, $\varpi \in \mathbb{R}_+$ is defined by equation C.99, and $\Xi(I, J) : \mathbb{N}_+ \times \mathbb{N}_+ \to \mathbb{R}_+$ is defined by equation C.101, all in Appendix C.6.*

## C CONVERGENCE THEORY PROOFS

This appendix contains the proofs of Lemmas B.1–B.4 (which are utilized to bound the terms in the Lyapunov function given by equation B.1) as well as the proofs of Theorems B.5–B.6.

## C.1 PROOF OF LEMMA B.1

**Proof.** From the Lipschitz property of $\nabla f$ (equation equation E.11), taking expectation conditioned on $\mathcal{F}_i$, and letting $\bar{g}_{f_1}^i = \mathbb{E}[\tilde{g}_{f_1}^i | \mathcal{F}_i]$, we have

$$\mathbb{E}[f(x^{i+1})|\mathcal{F}_i] - \mathbb{E}[f(x^i)|\mathcal{F}_i] \leq \mathbb{E}[\nabla f(x^i)^\top (x^{i+1} - x^i)|\mathcal{F}_i] + \frac{L_F}{2}\mathbb{E}[\|x^{i+1} - x^i\|^2|\mathcal{F}_i]$$

$$= \mathbb{E}[\nabla f(x^i)^\top (x^i - \alpha_i \tilde{g}_{f_1}^i - x^i)|\mathcal{F}_i] + \frac{L_F}{2}\mathbb{E}[\|x^i - \alpha_i \tilde{g}_{f_1}^i - x^i\|^2|\mathcal{F}_i]$$

$$= -\alpha_i \nabla f(x^i)^\top \bar{g}_{f_1}^i + \frac{L_F}{2}\alpha_i^2 \mathbb{E}[\|\tilde{g}_{f_1}^i\|^2|\mathcal{F}_i],$$

Using the fact that $2a^\top b = \|a\|^2 + \|b\|^2 - \|a - b\|^2$ twice, with $a$ and $b$ real-valued vectors, yields

$$\mathbb{E}[f(x^{i+1})|\mathcal{F}_i] - \mathbb{E}[f(x^i)|\mathcal{F}_i] \leq -\frac{\alpha_i}{2}\|\nabla f(x^i)\|^2 - \frac{\alpha_i}{2}\|\bar{g}_{f_1}^i\|^2 + \frac{\alpha_i}{2}\|\nabla f(x^i) - \bar{g}_{f_1}^i\|^2 + \frac{L_F}{2}\alpha_i^2 \mathbb{E}[\|\tilde{g}_{f_1}^i\|^2|\mathcal{F}_i]$$

$$= -\frac{\alpha_i}{2}\|\nabla f(x^i)\|^2 - \frac{\alpha_i}{2}\|\bar{g}_{f_1}^i\|^2 + \frac{\alpha_i}{2}\|\nabla f(x^i) - \bar{g}_{f_1}^i\|^2$$

$$+ \frac{L_F \alpha_i^2}{2}\mathbb{E}[2\left(\tilde{g}_{f_1}^i\right)^\top \bar{g}_{f_1}^i - \|\bar{g}_{f_1}^i\|^2 + \|\tilde{g}_{f_1}^i - \bar{g}_{f_1}^i\|^2|\mathcal{F}_i]$$

$$= -\frac{\alpha_i}{2}\|\nabla f(x^i)\|^2 - \frac{\alpha_i}{2}\|\bar{g}_{f_1}^i\|^2 + \frac{\alpha_i}{2}\|\nabla f(x^i) - \bar{g}_{f_1}^i\|^2$$

$$+ \frac{L_F \alpha_i^2}{2}\mathbb{E}[\|\bar{g}_{f_1}^i\|^2|\mathcal{F}_i] + \frac{L_F \alpha_i^2}{2}\mathbb{E}[\|\tilde{g}_{f_1}^i - \bar{g}_{f_1}^i\|^2|\mathcal{F}_i].$$

Utilizing Lemma D.3 and realizing that $\mathbb{E}[\|\bar{g}_{f_1}^i\|^2|\mathcal{F}_i] = \|\bar{g}_{f_1}^i\|^2$, we have

$$\mathbb{E}[f(x^{i+1})|\mathcal{F}_i] - \mathbb{E}[f(x^i)|\mathcal{F}_i] \leq -\frac{\alpha_i}{2}\|\nabla f(x^i)\|^2 - \left(\frac{\alpha_i}{2} - \frac{L_F \alpha_i^2}{2}\right)\|\bar{g}_{f_1}^i\|^2 + \frac{\alpha_i}{2}\|\nabla f(x^i) - \bar{g}_{f_1}^i\|^2 + \frac{\tau L_F \alpha_i^2}{2}.$$

Further, we decompose the gradient bias term by adding and subtracting $\nabla f(x^i, y^{i+1}, z^{i+1})$, using the fact that $\|a + b\|^2 \leq 2\|a\|^2 + 2\|b\|^2$, with $a$ and $b$ real-valued vectors, yielding

$$\|\nabla f(x^i) - \bar{g}_{f_1}^i\|^2 \leq 2\|\nabla f(x^i, y(x^i), z(x^i)) - \nabla f(x^i, y^{i+1}, z^{i+1})\|^2 + 2\|\nabla f(x^i, y^{i+1}, z^{i+1}) - \bar{g}_{f_1}^i\|^2$$

$$\leq 2L_{F_{yz}}^2 \|(y(x^i), z(x^i)) - (y^{i+1}, z^{i+1})\|^2 + 2\omega^2 \theta_i^2$$

$$\leq 2L_{F_{yz}}^2 (\|y(x^i) - y^{i+1}\|^2 + \|z(x^i) - z^{i+1}\|^2) + 2\omega^2 \alpha_i,$$

where the second inequality follows from equation E.12 and Lemma D.3, and the last inequality follows from the fact that $\theta_i = \alpha_i \beta_i \gamma_i \leq \alpha_i$ and $0 < \alpha_i^2 \leq \alpha_i \leq 1$. Putting this all together, we have

$$\mathbb{E}[f(x^{i+1})|\mathcal{F}_i] - \mathbb{E}[f(x^i)|\mathcal{F}_i]$$

$$\leq -\frac{\alpha_i}{2}\|\nabla f(x^i)\|^2 - \left(\frac{\alpha_i}{2} - \frac{L_F \alpha_i^2}{2}\right)\|\bar{g}_{f_1}^i\|^2 + \alpha_i L_{F_{yz}}^2 (\|y(x^i) - y^{i+1}\|^2 + \|z(x^i) - z^{i+1}\|^2) + \tilde{\omega}\alpha_i^2,$$

$$\text{where} \quad \tilde{\omega} := \left(\omega^2 + \frac{\tau L_F}{2}\right). \tag{C.1}$$

Taking total expectation, we obtain the final bound, completing the proof. $\qquad\square$

## C.2 PROOF OF LEMMA B.2

**Proof.** To derive the error bound defined by equation B.5, we start by decomposing the error of the LL variables by adding and subtracting $z(x^i)$ in the following way:

$$\mathbb{E}[\|z^{i+1} - z(x^{i+1})\|^2] = \underbrace{\mathbb{E}[\|z^{i+1} - z(x^i)\|^2]}_{A_1^{(1)}} + \underbrace{\mathbb{E}[\|z(x^i) - z(x^{i+1})\|^2]}_{A_2^{(1)}}$$

$$+ 2\underbrace{\mathbb{E}[(z^{i+1} - z(x^i))^\top (z(x^i) - z(x^{i+1}))]}_{A_3^{(1)}}. \tag{C.2}$$

(**Analysis of $A_1^{(1)}$**)**:** To derive an upper-bound on $A_1^{(1)}$ in equation C.2, recall that $z^{i+1} = z^{i+1,0,0} = z^{i,J,K}$ and $g_{f_3}^{i,j,k} = \nabla_z f_3(x^i, y^{i,j}, z^{i,j,k}; \xi^{i,j,k})$. Further, notice that there will be a total of $JK$ updates to the LL variables starting from $z^i$ to obtain $z^{i+1}$. Thus, in general, taking expectation conditioned on $\mathcal{F}_{i,j,k}$, we have

$$\mathbb{E}[\|z^{i,j,k+1} - z(x^i)\|^2 | \mathcal{F}_{i,j,k}] = \mathbb{E}[\|z^{i,j,k} - \gamma_i g_{f_3}^{i,j,k} - z(x^i)\|^2 | \mathcal{F}_{i,j,k}]$$
$$= \|z^{i,j,k} - z(x^i)\|^2 - 2\gamma_i (z^{i,j,k} - z(x^i))^\top \nabla_z f_3^{i,j,k} + \gamma_i^2 \mathbb{E}[\|g_{f_3}^{i,j,k}\|^2 | \mathcal{F}_{i,j,k}],$$

where the last equality follows from the unbiasedness of the stochastic estimates (Assumption 3.4). Using the fact that $\mathrm{Var}[X|Y] = \mathbb{E}[X^2|Y] - \mathbb{E}[X|Y]^2$, where $X$ and $Y$ are random variables, along with Assumption 3.4, we have

$$\mathbb{E}[\|z^{i,j,k+1} - z(x^i)\|^2 | \mathcal{F}_{i,j,k}] \leq \|z^{i,j,k} - z(x^i)\|^2 - 2\gamma_i (z^{i,j,k} - z(x^i))^\top \nabla_z f_3^{i,j,k} + \gamma_i^2 \|\nabla_z f_3^{i,j,k}\|^2 + \gamma_i^2 \sigma_{\nabla f_3}^2.$$

Now, utilizing (Nesterov, 2018, Theorem 2.1.12), which follows from the strong convexity and Lipschitz continuity of $f_3$ (Assumptions 3.1 and 3.2, respectively), we have

$$\mathbb{E}[\|z^{i,j,k+1} - z(x^i)\|^2 | \mathcal{F}_{i,j,k}]$$
$$\leq \|z^{i,j,k} - z(x^i)\|^2 - 2\gamma_i \left( \frac{\mu_z L_{\nabla f_3}}{\mu_z + L_{\nabla f_3}} \|z^{i,j,k} - z(x^i)\|^2 + \frac{1}{\mu_z + L_{\nabla f_3}} \|\nabla_z f_3^{i,j,k}\|^2 \right) + \gamma_i^2 \|\nabla_z f_3^{i,j,k}\|^2 + \gamma_i^2 \sigma_{\nabla f_3}^2$$
$$= \left( 1 - \frac{2\gamma_i \mu_z L_{\nabla f_3}}{\mu_z + L_{\nabla f_3}} \right) \|z^{i,j,k} - z(x^i)\|^2 + \gamma_i \left( \gamma_i - \frac{2}{\mu_z + L_{\nabla f_3}} \right) \|\nabla_z f_3^{i,j,k}\|^2 + \gamma_i^2 \sigma_{\nabla f_3}^2$$
$$\leq (1 - \gamma_i \rho_{f_3}) \|z^{i,j,k} - z(x^i)\|^2 + \gamma_i^2 \sigma_{\nabla f_3}^2,$$

where the last inequality follows from the assumption that $\gamma_i \leq \frac{1}{\mu_z + L_{\nabla f_3}}$ and by letting

$$\rho_{f_3} := \frac{2\mu_z L_{\nabla f_3}}{\mu_z + L_{\nabla f_3}}. \tag{C.3}$$

Using induction over $K$ and taking total expectation, we obtain the bound equation B.3.

At this point, there would be an update in the ML variables $y$, i.e., $(x^i, y^{i,j}, z^{i,j,K}) \rightarrow (x^i, y^{i,j+1}, z^{i,j,K})$. However, since this upper-bound is not dependent on $y$, we can use induction over all $J$ iterations (each consisting of $K$ iterations), which yields the bound equation B.4. These results follow by ensuring that $0 \leq 1 - \gamma_i \rho_{f_3} \leq 1$, which is satisfied by the assumption $\gamma_i \leq \frac{1}{\mu_z + L_{\nabla f_3}}$ and recalling that $\gamma_i$ and $\rho_{f_3}$ are positive.

(**Analysis of $A_2^{(1)}$**)**:** Taking expectation conditioned on $\mathcal{F}_i$ and applying equation E.1 yields

$$\mathbb{E}[\|z(x^i) - z(x^{i+1})\|^2 | \mathcal{F}_i] \leq L_z^2 \mathbb{E}[\|x^i - x^{i+1}\|^2 | \mathcal{F}_i] = L_z^2 \alpha_i^2 \mathbb{E}[\|\tilde{g}_{f_1}^i\|^2 | \mathcal{F}_i].$$

Adding and subtracting $\bar{g}_{f_1}^i = \mathbb{E}[\tilde{g}_{f_1}^i | \mathcal{F}_i]$ followed by using the fact that $\|a+b\|^2 \leq 2 \left( \|a\|^2 + \|b\|^2 \right)$, with $a$ and $b$ real-valued vectors, along with Lemma D.3, we have

$$\mathbb{E}[\|z(x^i) - z(x^{i+1})\|^2 | \mathcal{F}_i] \leq L_z^2 \alpha_i^2 \mathbb{E}[\|\tilde{g}_{f_1}^i - \bar{g}_{f_1}^i + \bar{g}_{f_1}^i\|^2 | \mathcal{F}_i] \leq 2L_z^2 \alpha_i^2 \left( \mathbb{E}[\|\bar{g}_{f_1}^i\|^2 | \mathcal{F}_i] + \tau \right).$$

Lastly, taking total expectation, we obtain the bound

$$\mathbb{E}[\|z(x^i) - z(x^{i+1})\|^2] \leq 2L_z^2 \alpha_i^2 (\mathbb{E}[\|\bar{g}_{f_1}^i\|^2] + \tau).$$

(**Analysis of $A_3^{(1)}$**)**:** Taking expectation conditioned on $\mathcal{F}_i$ followed by adding and subtracting $\nabla_x z (x^i)^\top (x^{i+1} - x^i)$ in the following way:

$$\mathbb{E}[(z^{i+1} - z(x^i))^\top (z(x^i) - z(x^{i+1}))|\mathcal{F}_i]$$
$$= -\mathbb{E}[(z^{i+1} - z(x^i))^\top (\nabla_x z(x^i)^\top (x^{i+1} - x^i) + z(x^{i+1}) - z(x^i) - \nabla_x z(x^i)^\top (x^{i+1} - x^i))|\mathcal{F}_i]$$
$$= \underbrace{-\mathbb{E}[(z^{i+1} - z(x^i))^\top (\nabla_x z(x^i)^\top (x^{i+1} - x^i))|\mathcal{F}_i]}_{B_1^{(1)}}$$
$$\underbrace{-\mathbb{E}[(z^{i+1} - z(x^i))^\top (z(x^{i+1}) - z(x^i) - \nabla_x z(x^i)^\top (x^{i+1} - x^i))|\mathcal{F}_i]}_{B_2^{(1)}}. \tag{C.4}$$

(**Analysis of $B_1^{(1)}$**): Utilizing the update $x^{i+1} = x^i - \alpha_i g_{f_1}^i$, the fact that $\mathbb{E}[X] = \mathbb{E}[\mathbb{E}[X|Y]]$, along with the Cauchy-Schwarz inequality, yields

$$
\begin{aligned}
B_1^{(1)} &\leq \alpha_i \mathbb{E}[\|z^{i+1} - z(x^i)\|\|\nabla_x z(x^i)^\top \bar{g}_{f_1}^i\|\,|\mathcal{F}_i] \\
&\leq \alpha_i L_z \mathbb{E}[\|z^{i+1} - z(x^i)\|\|\bar{g}_{f_1}^i\|\,|\mathcal{F}_i] \\
&\leq \kappa_i \mathbb{E}[\|z^{i+1} - z(x^i)\|^2|\mathcal{F}_i] + \frac{\alpha_i^2 L_z^2}{4\kappa_i} \mathbb{E}[\|\bar{g}_{f_1}^i\|^2|\mathcal{F}_i],
\end{aligned}
$$

where the second inequality comes from equation E.1, and the last inequality comes from using Young's inequality (i.e., $ab \leq \frac{\epsilon a^2}{2} + \frac{b^2}{2\epsilon}$ for $\epsilon > 0$), where $\epsilon = 2\kappa_i$ for some $\kappa_i > 0$, and where $a = \|z^{i+1} - z(x^i)\|$ and $b = \alpha_i L_z \|\bar{g}_{f_1}^i\|$.

(**Analysis of $B_2^{(1)}$**): Now, we can bound the term $B_2^{(1)}$ in equation C.4 by using the Cauchy-Schwarz inequality and applying the Lipschitz property of equation E.5 (i.e, $z(x^{i+1}) - z(x^i) - \nabla z(x^i)(x^{i+1} - x^i) \leq \frac{L_{\nabla z}}{2}\|x^{i+1} - x^i\|^2$) to obtain:

$$
\begin{aligned}
&-\mathbb{E}[(z^{i+1} - z(x^i))^\top(z(x^{i+1}) - z(x^i) - \nabla_x z(x^i)^\top(x^{i+1} - x^i))|\mathcal{F}_i] \\
&\leq \frac{L_{\nabla z}}{2}\mathbb{E}[\|z^{i+1} - z(x^i)\|\|x^{i+1} - x^i\|\|x^{i+1} - x^i\|\,|\mathcal{F}_i].
\end{aligned}
$$

Further, using Young's inequality with $a = \|z^{i+1} - z(x^i)\|\|x^{i+1} - x^i\|$ and $b = \|x^{i+1} - x^i\|$ such that $ab \leq \frac{a^2}{2} + \frac{b^2}{2}$, along with the update $x^{i+1} = x^i - \alpha_i \tilde{g}_{f_1}^i$ and the fact that $\mathbb{E}[X] = \mathbb{E}[\mathbb{E}[X|Y]]$, we have

$$
\begin{aligned}
&-\mathbb{E}[(z^{i+1} - z(x^i))^\top(z(x^{i+1}) - z(x^i) - \nabla_x z(x^i)^\top(x^{i+1} - x^i))|\mathcal{F}_i] \\
&\leq \frac{L_{\nabla z}}{2}\left(\frac{1}{2}\mathbb{E}[\|z^{i+1} - z(x^i)\|^2\|x^{i+1} - x^i\|^2|\mathcal{F}_i] + \frac{1}{2}\mathbb{E}[\|x^{i+1} - x^i\|^2|\mathcal{F}_i]\right) \\
&\leq \frac{L_{\nabla z}\alpha_i^2\zeta}{4}\mathbb{E}[\|z^{i+1} - z(x^i)\|^2|\mathcal{F}_i] + \frac{L_{\nabla z}\alpha_i^2}{4}\mathbb{E}[\|\tilde{g}_{f_1}^i\|^2|\mathcal{F}_i] \\
&\leq \frac{L_{\nabla z}\alpha_i^2\zeta}{4}\mathbb{E}[\|z^{i+1} - z(x^i)\|^2|\mathcal{F}_i] + \frac{L_{\nabla z}\alpha_i^2}{4}(\mathbb{E}[\|\bar{g}_{f_1}^i\|^2|\mathcal{F}_i] + \tau).
\end{aligned}
$$

where the second inequality follows by applying Lemma D.4 and the last follows by applying the definition of variance along with Lemma D.3.

Substituting these bounds for $B_1^{(1)}$ and $B_2^{(1)}$ back into equation C.4 and taking total expectation, we obtain the bound on the term $A_3^{(1)}$ as

$$
\begin{aligned}
&\mathbb{E}[(z^{i+1} - z(x^i))^\top(z(x^i) - z(x^{i+1}))] \\
&\leq \left(\kappa_i + \frac{L_{\nabla z}\alpha_i^2\zeta}{4}\right)\mathbb{E}[\|z^{i+1} - z(x^i)\|^2] + \left(\frac{\alpha_i^2 L_z^2}{4\kappa_i} + \frac{L_{\nabla z}\alpha_i^2}{4}\right)\mathbb{E}[\|\bar{g}_{f_1}^i\|^2] + \frac{\tau L_{\nabla z}}{4}\alpha_i^2.
\end{aligned}
$$

Finally, substituting these bounds for $A_1^{(1)}$, $A_2^{(1)}$, and $A_3^{(1)}$ back into equation C.2, we obtain the desired upper-bound on $\mathbb{E}[\|z^{i+1} - z(x^{i+1})\|^2]$, completing the proof. $\qquad\square$

## C.3  PROOF OF LEMMA B.3

**Proof.** To derive the error bound defined by equation B.6, recall that $z^{i+1} = z^{i+1,0,0} = z^{i,J,K}$ and $g_{f_3}^{i,j,k} = \nabla_z f_3(x^i, y^{i,j}, z^{i,j,k}; \xi^{i,j,k})$ and notice that there will be a total of $K$ updates to the LL variables starting from $z^{i,j}$ to obtain $z^{i,j+1}$. Further, following the exact same steps utilized in Lemma B.2 to derive bound equation B.3 (only with $z(x^i)$ replaced with $z(x^i, y^{i,j+1})$), we have

$$
\mathbb{E}[\|z^{i,j+1} - z(x^i, y^{i,j+1})\|^2] \leq (1 - \gamma_i \rho_{f_3})^K \|z^{i,j} - z(x^i, y^{i,j+1})\|^2 + K\gamma_i^2\sigma_{\nabla f_3}^2.
$$

Now, adding and subtracting $z(x^i, y^{i,j})$ in the norm, followed by using the fact that $\|a + b\|^2 = \|a\|^2 + \|b\|^2 + 2a^\top b$, with $a$ and $b$ real-valued vectors, the Cauchy-Schwarz inequality, and the fact

that $(1 - \gamma_i \rho_{f_3})^K \leq 1$ which is satisfied by our choice of $\gamma_i \leq \frac{1}{\mu_z + L_{\nabla f_3}}$, we have

$$\mathbb{E}[\|z^{i,j+1} - z(x^i, y^{i,j+1})\|^2]$$

$$\leq (1 - \gamma_i \rho_{f_3})^K \|z^{i,j} - z(x^i, y^{i,j})\|^2 + (1 - \gamma_i \rho_{f_3})^K \|z(x^i, y^{i,j}) - z(x^i, y^{i,j+1})\|^2 + K\gamma_i^2 \sigma_{\nabla f_3}^2$$

$$+ 2\|z^{i,j} - z(x^i, y^{i,j})\|\|z(x^i, y^{i,j}) - z(x^i, y^{i,j+1})\|$$

$$\leq (1 - \gamma_i \rho_{f_3})^K \|z^{i,j} - z(x^i, y^{i,j})\|^2 + (1 - \gamma_i \rho_{f_3})^K \|z(x^i, y^{i,j}) - z(x^i, y^{i,j+1})\|^2 + K\gamma_i^2 \sigma_{\nabla f_3}^2$$

$$+ \eta_i \|z^{i,j} - z(x^i, y^{i,j})\|^2 + \frac{1}{\eta_i} \|z(x^i, y^{i,j}) - z(x^i, y^{i,j+1})\|^2$$

$$\leq ((1 - \gamma_i \rho_{f_3})^K + \eta_i)\|z^{i,j} - z(x^i, y^{i,j})\|^2 + \left((1 - \gamma_i \rho_{f_3})^K + \frac{1}{\eta_i}\right) L_{z_y}^2 \|y^{i,j+1} - y^{i,j}\|^2 + K\gamma_i^2 \sigma_{\nabla f_3}^2$$

$$= ((1 - \gamma_i \rho_{f_3})^K + \eta_i)\|z^{i,j} - z(x^i, y^{i,j})\|^2 + \left((1 - \gamma_i \rho_{f_3})^K + \frac{1}{\eta_i}\right) L_{z_y}^2 \|y^{i,j} - \beta_i \tilde{g}_{f_2}^{i,j} - y^{i,j}\|^2$$

$$+ K\gamma_i^2 \sigma_{\nabla f_3}^2$$

$$\leq ((1 - \gamma_i \rho_{f_3})^K + \eta_i)\|z^{i,j} - z(x^i, y^{i,j})\|^2 + \hat{\eta}_i L_{z_y}^2 \beta_i^2 \|\tilde{g}_{f_2}^{i,j}\|^2 + K\gamma_i^2 \sigma_{\nabla f_3}^2,$$

where the second inequality follows from applying Young's inequality (i.e., $ab \leq \frac{\epsilon a^2}{2} + \frac{b^2}{2\epsilon}$ for $\epsilon > 0$) with $\epsilon = \eta_i$ for some $\eta_i > 0$ (notice that $a = \|z^{i,j} - z(x^i, y^{i,j})\|$ and $b = \|z(x^i, y^{i,j}) - z(x^i, y^{i,j+1})\|$ here), the third inequality follows from applying equation E.3, and the last inequality follows from the fact that $0 \leq 1 - \gamma_i \rho_{f_3} \leq 1$ (where we define $\hat{\eta}_i := 1 + \frac{1}{\eta_i}$). Lastly, taking total expectation and using the fact that $\mathbb{E}[X] = \mathbb{E}[\mathbb{E}[X|Y]]$, we will obtain the bound equation B.6

$$\mathbb{E}[\|z^{i,j+1} - z(x^i, y^{i,j+1})\|^2]$$

$$\leq ((1 - \gamma_i \rho_{f_3})^K + \eta_i)\mathbb{E}[\|z^{i,j} - z(x^i, y^{i,j})\|^2] + \hat{\eta}_i L_{z_y}^2 \beta_i^2 \mathbb{E}[\mathbb{E}[\|\tilde{g}_{f_2}^{i,j}\|^2|\mathcal{F}_{i,j}]] + K\gamma_i^2 \sigma_{\nabla f_3}^2$$

$$\leq ((1 - \gamma_i \rho_{f_3})^K + \eta_i)\mathbb{E}[\|z^{i,j} - z(x^i, y^{i,j})\|^2] + \hat{\eta}_i L_{z_y}^2 \beta_i^2 \Upsilon + K\gamma_i^2 \sigma_{\nabla f_3}^2,$$

where the last inequality follows by applying Lemma D.6.

Now, to derive results equation B.7, equation B.8, and equation B.9, we start by decomposing the expected error of the LL variables by adding and subtracting $z(x^i, y^i)$ followed by using the fact that $\|a + b\|^2 \leq 2(\|a\|^2 + \|b\|^2)$ with $a$ and $b$ real-valued vectors:

$$\mathbb{E}[\|z^{i+1} - z(x^{i+1}, y^{i+1})\|^2] \leq 2\underbrace{\mathbb{E}[\|z^{i+1} - z(x^i, y^i)\|^2]}_{A_1^{(2)}} + 2\underbrace{\mathbb{E}[\|z(x^i, y^i) - z(x^{i+1}, y^{i+1})\|^2]}_{A_2^{(2)}}.$$

$$\text{(C.5)}$$

(**Analysis of $A_1^{(2)}$**): To derive an upper-bound on $A_1^{(2)}$ in equation C.5, we can follow the exact same steps that were utilized in Lemma B.2 to derive bound equation B.3 (only with $z(x^i)$ replaced with $z(x^i, y^i)$), which will yield the bound equation B.7. Further, using induction over $J$ (each consisting of $K$ iterations) will yield the following bound on $A_1^{(2)}$ in equation C.5 (which is the bound equation B.8). Notice that this induction result again follows by ensuring that $0 \leq 1 - \gamma_i \rho_{f_3} \leq 1$, which is satisfied by the assumption $\gamma_i \leq \frac{1}{\mu_z + L_{\nabla f_3}}$ and recalling that $\gamma_i$ and $\rho_{f_3}$ are positive.

(**Analysis of $A_2^{(2)}$**): Now, the upper-bound on $A_2^{(2)}$ in equation C.5 can be derived by taking total expectation, using the fact that $\mathbb{E}[X] = \mathbb{E}[\mathbb{E}[X|Y]]$, applying equation E.2, and recursively using the fact that $y^{i,j+1} = y^{i,j} - \beta_i \tilde{g}_{f_2}^{i,j}$ (while recalling that $y^{i+1} = y^{i,J}$ and $x^{i+1} = x^i - \alpha_i \tilde{g}_{f_1}^i$):

$$\mathbb{E}[\|z(x^i, y^i) - z(x^{i+1}, y^{i+1})\|^2] \leq L_{z_{xy}}^2 \mathbb{E}[\|x^i - x^{i+1}\|^2] + L_{z_{xy}}^2 \mathbb{E}[\mathbb{E}[\|y^i - y^{i+1}\|^2|\mathcal{F}_{i,j}]]$$

$$= L_{z_{xy}}^2 \alpha_i^2 \mathbb{E}[\|\tilde{g}_{f_1}^i\|^2] + L_{z_{xy}}^2 \mathbb{E}[\mathbb{E}[\|y^i - \sum_{j=0}^{J-1} \beta_i \tilde{g}_{f_2}^{i,j} - y^i\|^2|\mathcal{F}_{i,j}]]$$

$$\leq L_{z_{xy}}^2 \alpha_i^2 \mathbb{E}[\|\tilde{g}_{f_1}^i\|^2] + J L_{z_{xy}}^2 \beta_i^2 \sum_{j=0}^{J-1} \mathbb{E}[\mathbb{E}[\|\tilde{g}_{f_2}^{i,j}\|^2|\mathcal{F}_{i,j}]]$$

$$\leq L_{z_{xy}}^2 \alpha_i^2 \mathbb{E}[\|\tilde{g}_{f_1}^i\|^2] + J^2 \Upsilon L_{z_{xy}}^2 \beta_i^2,$$

where the second inequality follows from using the fact that $\|\sum_{i=1}^{N} a_i\|^2 \leq N \sum_{i=1}^{N} \|a_i\|^2$ (for some $a \in \mathbb{R}^N$) and the last inequality follows from applying Lemma D.6. Now, using the fact that $\mathbb{E}[X] = \mathbb{E}[\mathbb{E}[X|Y]]$, adding and subtracting $\bar{g}_{f_1}^i$ in the norm, followed by using the fact that $\|a + b\|^2 \leq 2\left(\|a\|^2 + \|b\|^2\right)$, and applying Lemma D.3, we have

$$\mathbb{E}[\|z(x^i, y^i) - z(x^{i+1}, y^{i+1})\|^2] = L_{z_{xy}}^2 \alpha_i^2 \mathbb{E}[\mathbb{E}[\|\tilde{g}_{f_1}^i\|^2|\mathcal{F}_i]] + J^2 \Upsilon L_{z_{xy}}^2 \beta_i^2$$
$$\leq 2L_{z_{xy}}^2 \alpha_i^2 (\mathbb{E}[\|\bar{g}_{f_1}^i\|^2] + \tau) + J^2 \Upsilon L_{z_{xy}}^2 \beta_i^2.$$

Notice in the inequality that $\mathbb{E}[\|\bar{g}_{f_1}^i\|^2] = \mathbb{E}[\mathbb{E}[\|\bar{g}_{f_1}^i\|^2|\mathcal{F}_i]] = \|\bar{g}_{f_1}^i\|^2$ since $\bar{g}_{f_1}^i$ is deterministic. Finally, substituting these bounds for $A_1^{(2)}$ and $A_2^{(2)}$ back into equation C.5, we can obtain the desired upper-bound on $\mathbb{E}[\|z^{i+1} - z\left(x^{i+1}, y^{i+1}\right)\|^2]$ defined by equation B.9.

Lastly, to derive the upper-bound equation B.10, we can follow the exact same steps that were utilized in Lemma B.2 to derive bound equation B.3 (only with $z(x^i)$ replaced with $z(x^i, y^{i,j})$). Notice that this induction result again follows by ensuring that $0 \leq 1 - \gamma_i \rho_{f_3} \leq 1$, which is satisfied by the assumption of $\gamma_i \leq \frac{1}{\mu_z + L_{\nabla f_3}}$ and recalling that $\gamma_i$ and $\rho_{f_3}$ are positive. $\qquad\square$

### C.4  Proof of Lemma B.4

**Proof.** To derive the error bound defined by equation B.12, we start by decomposing the expected error of the LL variables by adding and subtracting $y(x^i)$ in the following way:

$$\mathbb{E}[\|y^{i+1} - y(x^{i+1})\|^2] = \underbrace{\mathbb{E}[\|y^{i+1} - y(x^i)\|^2]}_{A_1^{(3)}} + \underbrace{\mathbb{E}[\|y(x^i) - y(x^{i+1})\|^2]}_{A_2^{(3)}}$$
$$+ 2\underbrace{\mathbb{E}[(y^{i+1} - y(x^i))^\top (y(x^i) - y(x^{i+1}))]}_{A_3^{(3)}}. \tag{C.6}$$

(**Analysis of $A_1^{(3)}$**): To derive an upper-bound on $A_1^{(3)}$ in equation C.6, recall that $y^{i+1} = y^{i,J}$ and $\tilde{g}_{f_2}^{i,j} = \nabla_y \bar{f}(x^i, y^{i,j}, z^{i,j+1}; \xi^{i,j})$. Further, notice that there will be a total of $J$ updates to the ML variables starting from $y^i$ to obtain $y^{i+1}$. Thus, in general, taking expectation conditioned on $\mathcal{F}_{i,j}$ and applying Lemma D.6, we have

$$\mathbb{E}[\|y^{i,j+1} - y(x^i)\|^2|\mathcal{F}_{i,j}] = \mathbb{E}[\|y^{i,j} - \beta_i \tilde{g}_{f_2}^{i,j} - y(x^i)\|^2|\mathcal{F}_{i,j}]$$
$$\leq \|y^{i,j} - y(x^i)\|^2 - 2\beta_i (y^{i,j} - y(x^i))^\top \bar{g}_{f_2}^{i,j} + \Upsilon \beta_i^2$$
$$= \|y^{i,j} - y(x^i)\|^2 + \Upsilon \beta_i^2 - 2\beta_i (y^{i,j} - y(x^i))^\top \nabla_y \bar{f}(x^i, y^{i,j})$$
$$- 2\beta_i (y^{i,j} - y(x^i))^\top (\bar{g}_{f_2}^{i,j} - \nabla_y \bar{f}(x^i, y^{i,j})),$$

where the last equality follows from adding and subtracting $\nabla_y \bar{f}(x^i, y^{i,j})$ to the $\bar{g}_{f_2}^{i,j}$ term in the cross-product. Now, under the strong convexity of $\bar{f}$ (Assumption 3.3) and the Lipschitz continuity of $\nabla_y \bar{f}$ in $y$ (equation equation E.7), we can utilize (Nesterov, 2018, Theorem 2.1.12), yielding

$$\mathbb{E}[\|y^{i,j+1} - y(x^i)\|^2|\mathcal{F}_{i,j}] \leq \|y^{i,j} - y(x^i)\|^2 + \Upsilon \beta_i^2$$
$$- 2\beta_i \left( \frac{\mu_y L_{\nabla \bar{f}}}{\mu_y + L_{\nabla \bar{f}}} \|y^{i,j} - y(x^i)\|^2 + \frac{1}{\mu_y + L_{\nabla \bar{f}}} \|\nabla_y \bar{f}(x^i, y^{i,j})\|^2 \right)$$
$$+ 2\beta_i \|y^{i,j} - y(x^i)\|^2 \|\bar{g}_{f_2}^{i,j} - \nabla_y \bar{f}(x^i, y^{i,j}, z^{i,j+1})\|^2$$
$$+ 2\beta_i \|y^{i,j} - y(x^i)\| \|\nabla_y \bar{f}(x^i, y^{i,j}, z^{i,j+1}) - \nabla_y \bar{f}(x^i, y^{i,j})\|,$$

where the last two added terms come from adding and subtracting $\nabla_y \bar{f}(x^i, y^{i,j}, z^{i,j+1})$ to the $\bar{g}_{f_2}^{i,j} - \nabla_y \bar{f}(x^i, y^{i,j})$ term in the cross product followed by applying the Cauchy Schwarz inequality. Now, utilizing the Lipschitz continuity of $\nabla_y \bar{f}$ in $z$ (equation equation E.8), the bound on the biasedness

of $\tilde{g}_{f_2}$ (Lemma D.5), and the fact that $\frac{2\beta_i}{\mu_y+L_{\nabla\bar{f}}}\|\nabla_y\bar{f}(x^i,y^{i,j})\|^2$ is non-negative, we have

$$\mathbb{E}[\|y^{i,j+1}-y(x^i)\|^2|\mathcal{F}_{i,j}]$$

$$\leq \left(1-\beta_i\left(\frac{2\mu_y L_{\nabla\bar{f}}}{\mu_y+L_{\nabla\bar{f}}}-2\hat{\omega}^2\theta_i^2\right)\right)\|y^{i,j}-y(x^i)\|^2 + 2\beta_i\|y^{i,j}-y(x^i)\|\|z^{i,j+1}-z(x^i,y^{i,j})\| + \Upsilon\beta_i^2$$

$$\leq \left(1-\beta_i\left(\frac{2\mu_y L_{\nabla\bar{f}}}{\mu_y+L_{\nabla\bar{f}}}-2\hat{\omega}^2\theta_i^2-\beta_i\right)\right)\|y^{i,j}-y(x^i)\|^2 + \|z^{i,j+1}-z(x^i,y^{i,j})\|^2 + \Upsilon\beta_i^2$$

$$= (1-\psi_i\beta_i)\|y^{i,j}-y(x^i)\|^2 + \|z^{i,j+1}-z(x^i,y^{i,j})\|^2 + \Upsilon\beta_i^2,$$

where the last inequality follows from the fact that $2ab \leq a^2+b^2$ ($a$ and $b$ positive scalars) where

$$\psi_i := \rho - 2\hat{\omega}^2\theta_i^2 - \beta_i \quad\text{and}\quad \rho := \frac{2\mu_y L_{\nabla\bar{f}}}{\mu_y+L_{\nabla\bar{f}}}. \tag{C.7}$$

Taking total expectation and using bound equation B.10 from Lemma B.3, we have

$$\mathbb{E}[\|y^{i,j+1}-y(x^i)\|^2] \tag{C.8}$$

$$\leq (1-\psi_i\beta_i)\mathbb{E}[\|y^{i,j}-y(x^i)\|^2] + \Upsilon\beta_i^2 + (1-\gamma_i\rho_{f_3})^K\mathbb{E}[\|z^{i,j}-z(x^i,y^{i,j})\|^2] + K\gamma_i^2\sigma_{\nabla f_3}^2$$

$$\leq (1-\psi_i\beta_i)^J\mathbb{E}[\|y^i-y(x^i)\|^2] + (1-\gamma_i\rho_{f_3})^K\sum_{j=0}^{J-1}\mathbb{E}[\|z^{i,j}-z(x^i,y^{i,j})\|^2] + J\Upsilon\beta_i^2 + JK\gamma_i^2\sigma_{\nabla f_3}^2,$$

$$\tag{C.9}$$

where the last inequality follows by using induction over $J$. Notice that this result follows by ensuring that $0 \leq 1-\psi_i\beta_i \leq 1$, which holds when choosing $\beta_i$ such that $\beta_i \leq \frac{1}{\mu_y+L_{\nabla\bar{f}}}$ and $\beta_i \leq \frac{\rho}{2\hat{\omega}^2+1}$. In other words, to show that $0 \leq 1-\psi_i\beta_i$, we have

$$\psi_i\beta_i = \beta_i(\rho - 2\hat{\omega}^2\theta_i^2 - \beta_i) < \beta_i\rho \leq \frac{2\mu_y L_{\nabla\bar{f}}}{(\mu_y+L_{\nabla\bar{f}})^2} \leq 1,$$

where the first inequality follows by observing that $-2\hat{\omega}^2\theta_i^2\beta_i - \beta_i^2 < 0$, the second inequality follows by choosing $\beta_i \leq \frac{1}{\mu_y+L_{\nabla\bar{f}}}$ along with the definition of $\rho$, and the third inequality follows from the fact that $2ab \leq (a+b)^2$, with $a$ and $b$ positive scalars. Notice that showing that $1-\psi_i\beta_i \leq 1$ is equivalent to showing that $\psi_i \geq 0$, i.e., using the fact that $0 < \theta_i^2 \leq \theta_i \leq 1$ along with $\theta_i = \alpha_i\beta_i\gamma_i \leq \beta_i$, we have

$$\rho - 2\hat{\omega}^2\theta_i^2 - \beta_i \geq 0 \quad\Rightarrow\quad 2\hat{\omega}^2\beta_i + \beta_i \leq \rho \quad\Rightarrow\quad \beta_i \leq \frac{\rho}{2\hat{\omega}^2+1}.$$

Now, looking at the $\sum_{j=0}^{J-1}\mathbb{E}[\|z^{i,j}-z(x^i,y^i)\|^2]$ term in equation C.9 and defining $\Theta_i := \hat{\eta}_i L_{z_y}^2\Upsilon\beta_i^2 + K\gamma_i^2\sigma_{\nabla f_3}^2$, we have

$$\sum_{j=0}^{J-1}\mathbb{E}[\|z^{i,j}-z(x^i,y^{i,j})\|^2]$$

$$= \mathbb{E}[\|z^{i,0}-z(x^i,y^{i,0})\|^2] + \mathbb{E}[\|z^{i,1}-z(x^i,y^{i,1})\|^2] + \cdots + \mathbb{E}[\|z^{i,J-1}-z(x^i,y^{i,J-1})\|^2]$$

$$= \mathbb{E}[\|z^i-z(x^i,y^i)\|^2]$$

$$\quad + \mathbb{E}[\|z^{i,1}-z(x^i,y^{i,1})\|^2] \qquad\longrightarrow\quad \left(\leq ((1-\gamma_i\rho_{f_3})^K+\eta_i)\mathbb{E}[\|z^i-z(x^i,y^i)\|^2] + \Theta_i\right)$$

$$\quad + \mathbb{E}[\|z^{i,2}-z(x^i,y^{i,2})\|^2] \qquad\longrightarrow\quad \left(\leq ((1-\gamma_i\rho_{f_3})^K+\eta_i)^2\mathbb{E}[\|z^i-z(x^i,y^i)\|^2] + 2\Theta_i\right)$$

$$\quad\vdots$$

$$\quad + \mathbb{E}[\|z^{i,J-1}-z(x^i,y^{i,J-1})\|^2] \quad\longrightarrow\quad \left(\leq ((1-\gamma_i\rho_{f_3})^K+\eta_i)^{J-1}\mathbb{E}[\|z^i-z(x^i,y^i)\|^2] + (J-1)\Theta_i\right)$$

$$\leq \mathbb{E}[\|z^i-z(x^i,y^i)\|^2] + \sum_{j=1}^{J-1}((1-\gamma_i\rho_{f_3})^K+\eta_i)^j\mathbb{E}[\|z^i-z(x^i,y^i)\|^2] + \Theta_i\sum_{j=1}^{J-1}j, \tag{C.10}$$

where the intermediate inequalities follow from applying equation equation B.6 from Lemma B.3 repeatedly while choosing $\eta_i$ such that $\eta_i \leq 1 - (1 - \gamma_i \rho_{f_3})^K$ (which will ensure that $0 \leq (1 - \gamma_i \rho_{f_3})^K + \eta_i \leq 1$ when considering the fact that $0 \leq (1 - \gamma_i \rho_{f_3})^K \leq 1$ which is satisfied by our choice of $\gamma_i \leq \frac{1}{\mu_z + L_{\nabla f_3}}$ and recalling that $\gamma_i$, $\rho_{f_3}$, and $\eta_i$ are positive). Now, looking at the $\sum_{j=1}^{J-1}((1 - \gamma_i \rho_{f_3})^K + \eta_i)^j$ term in equation C.10, we have

$$\sum_{j=1}^{J-1}((1 - \gamma_i \rho_{f_3})^K + \eta_i)^j = \left( \frac{((1 - \gamma_i \rho_{f_3})^K + \eta_i) - ((1 - \gamma_i \rho_{f_3})^K + \eta_i)^J}{1 - ((1 - \gamma_i \rho_{f_3})^K + \eta_i)} \right) = \left( \frac{\vartheta_i - \vartheta_i^J}{1 - \vartheta_i} \right),$$

where the last equality follows by using the geometric series $\sum_{j=1}^{J-1} a^j = \frac{a - a^J}{1-a}$ when $a \in [0, 1]$ and defining $\vartheta_i := (1 - \gamma_i \rho_{f_3})^K + \eta_i$ for ease of notation. Now, using the partial sum $\sum_{j=1}^{J-1} j = \frac{J(J-1)}{2}$, we can see that the bound equation C.10 on the expression $\sum_{j=0}^{J-1} \mathbb{E}[\|z^{i,j} - z(x^i, y^{i,j})\|^2]$ is given by

$$\sum_{j=0}^{J-1} \mathbb{E}[\|z^{i,j} - z(x^i, y^{i,j})\|^2] \leq \left( 1 + \left( \frac{\vartheta_i - \vartheta_i^J}{1 - \vartheta_i} \right) \right) \mathbb{E}[\|z^i - z(x^i, y^i)\|^2] + \frac{J(J-1)}{2}\Theta_i. \quad \text{(C.11)}$$

Now, we wish to analyze the limiting behavior of the term $\frac{\vartheta_i - \vartheta_i^J}{1 - \vartheta_i}$ as $\vartheta \to 0$ and $\vartheta \to 1$ in order to obtain an upper-bound. Starting by analyzing the limiting behavior as $\vartheta \to 0$, we have

$$\lim_{\vartheta_i \to 0} \frac{\vartheta_i(1 - \vartheta_i^{J-1})}{1 - \vartheta_i} = \frac{0 \cdot 1}{1} = 0.$$

Further, when $\vartheta_i \to 1$, we can analyze the limiting behavior via L'Hopital's rule to obtain

$$\lim_{\vartheta_i \to 1} \frac{\vartheta_i - \vartheta_i^J}{1 - \vartheta_i} = \lim_{\vartheta_i \to 1} \frac{\frac{d}{d\vartheta_i}(\vartheta_i - \vartheta_i^J)}{\frac{d}{d\vartheta_i}(1 - \vartheta_i)} = \lim_{\vartheta_i \to 1} -(1 - J\vartheta_i^{J-1}) = J - 1.$$

Therefore, we can see that (since $1 \leq J \in \mathbb{N}$)

$$0 \leq \frac{\vartheta_i - \vartheta_i^J}{1 - \vartheta_i} \leq J - 1. \quad \text{(C.12)}$$

Utilizing the upper-bound of equation C.12 in equation C.11 yields

$$\sum_{j=0}^{J-1} \mathbb{E}[\|z^{i,j} - z(x^i, y^{i,j})\|^2] \leq J\mathbb{E}[\|z^i - z(x^i, y^i)\|^2] + \frac{J(J-1)}{2}\Theta_i. \quad \text{(C.13)}$$

Now, substituting equation C.13 back into equation equation C.9 and using the fact that $0 \leq 1 - \gamma_i \rho_{f_3} \leq 1$, which is satisfied by our choice of $\gamma_i \leq \frac{1}{\mu_z + L_{\nabla f_3}}$ and recalling that $\gamma_i$ and $\rho_{f_3}$ are positive, yields

$$\mathbb{E}[\|y^{i+1} - y(x^i)\|^2] \leq (1 - \psi_i \beta_i)^J \mathbb{E}[\|y^i - y(x^i)\|^2] + J\Upsilon\beta_i^2 + JK\gamma_i^2\sigma_{\nabla f_3}^2 + \frac{J(J-1)}{2}\Theta_i$$
$$+ (1 - \gamma_i \rho_{f_3})^K J\mathbb{E}[\|z^i - z(x^i, y^i)\|^2],$$

Further simplifying this expression, we obtain the bound equation B.11.

(**Analysis of $A_2^{(3)}$**): The derivation of the upper-bound on $A_2^{(3)}$ in equation C.6 follows the exact same steps that were used to derive the upper-bound on the term $A_2^{(1)}$ in Lemma B.2 (only with using equation E.4 instead of equation E.1), from which we have

$$\mathbb{E}[\|y(x^i) - y(x^{i+1})\|^2] \leq 2L_y^2\alpha_i^2(\mathbb{E}[\|\bar{g}_{f_1}^i\|^2] + \tau).$$

(**Analysis of $A_3^{(3)}$**)**:** The term $A_3^{(3)}$ in equation C.6 can be bounded by taking expectation conditioned on $\mathcal{F}_i$ followed by adding and subtracting $\nabla y(x^i)(x^{i+1} - x^i)$ in the following way:

$$
\begin{aligned}
&\mathbb{E}[(y^{i+1} - y(x^i))^\top (y(x^i) - y(x^{i+1}))|\mathcal{F}_i] \\
&= -\mathbb{E}[(y^{i+1} - y(x^i))^\top (\nabla y(x^i)(x^{i+1} - x^i) + y(x^{i+1}) - y(x^i) - \nabla y(x^i)(x^{i+1} - x^i))|\mathcal{F}_i] \\
&= \underbrace{-\mathbb{E}[(y^{i+1} - y(x^i))^\top (\nabla y(x^i)(x^{i+1} - x^i))|\mathcal{F}_i]}_{B_1^{(3)}} \\
&\quad \underbrace{-\mathbb{E}[(y^{i+1} - y(x^i))^\top (y(x^{i+1}) - y(x^i) - \nabla y(x^i)(x^{i+1} - x^i))|\mathcal{F}_i]}_{B_2^{(3)}}. \tag{C.14}
\end{aligned}
$$

(**Analysis of $B_1^{(3)}$**)**:** The derivation of the upper-bound on $B_1^{(3)}$ in equation C.14 follows the exact same steps that were used to derive the upper-bound on the term $B_1^{(1)}$ in Lemma B.2 (only with using equation E.4 instead of equation E.1), from which, for some $\phi_i > 0$, we have

$$
-\mathbb{E}[(y^{i+1} - y(x^i))^\top (\nabla y(x^i)(x^{i+1} - x^i))|\mathcal{F}_i] \le \phi_i \mathbb{E}[\|y^{i+1} - y(x^i)\|^2|\mathcal{F}_i] + \frac{\alpha_i^2 L_y^2}{4\phi_i} \mathbb{E}[\|\bar{g}_{f_1}^i\|^2|\mathcal{F}_i].
$$

(**Analysis of $B_2^{(3)}$**)**:** The derivation of the upper-bound on $B_2^{(3)}$ in equation C.14 follows the exact same steps that were used to derive the upper-bound on the term $B_2^{(1)}$ in Lemma B.2 (only with using equation E.13 instead of equation E.5), from which we have

$$
\begin{aligned}
&-\mathbb{E}[(y^{i+1} - y(x^i))^\top (y(x^{i+1}) - y(x^i) - \nabla y(x^i)(x^{i+1} - x^i))|\mathcal{F}_i] \\
&\le \frac{L_{\nabla y}\alpha_i^2 \zeta}{4} \mathbb{E}[\|y^{i+1} - y(x^i)\|^2|\mathcal{F}_i] + \frac{L_{\nabla y}\alpha_i^2}{4} (\mathbb{E}[\|\bar{g}_{f_1}^i\|^2|\mathcal{F}_i] + \tau).
\end{aligned}
$$

Finally, substituting these bounds for $B_1^{(3)}$ and $B_2^{(3)}$ back into equation C.14 and taking total expectation, we obtain the bound on the term $A_3^{(3)}$ as

$$
\begin{aligned}
\mathbb{E}[(y^{i+1} - y(x^i))^\top (y(x^i) - y(x^{i+1}))] &\le \left(\phi_i + \frac{L_{\nabla y}\alpha_i^2 \zeta}{4}\right) \mathbb{E}[\|y^{i+1} - y(x^i)\|^2] \\
&\quad + \left(\frac{\alpha_i^2 L_y^2}{4\phi_i} + \frac{L_{\nabla y}\alpha_i^2}{4}\right) \mathbb{E}[\|\bar{g}_{f_1}^i\|^2] + \frac{\tau L_{\nabla y}}{4} \alpha_i^2.
\end{aligned}
$$

Finally, substituting these bounds for $A_1^{(3)}$, $A_2^{(3)}$, and $A_3^{(3)}$ back into equation C.6, we obtain the desired upper-bound on $\mathbb{E}[\|y^{i+1} - y(x^{i+1})\|^2]$, completing the proof. $\qquad\square$

## C.5 PROOF OF THEOREM B.5

**Proof.** To begin, using Lemmas B.1, B.2, B.3, and B.4, we can begin by bounding the two Lyapunov difference terms (defined in equation B.1) by taking total expectation in the following way:

$$\mathbb{E}[\mathbb{V}^{i+1}] - \mathbb{E}[\mathbb{V}^i]$$

$$= \underbrace{\mathbb{E}[f(x^{i+1})] - \mathbb{E}[f(x^i)]}_{\text{Lemma B.1}} + \underbrace{\mathbb{E}[\|y^{i+1} - y(x^{i+1})\|^2]}_{\text{Lemma B.4}} - \mathbb{E}[\|y^i - y(x^i)\|^2]$$

$$+ \underbrace{\mathbb{E}[\|z^{i+1} - z(x^{i+1})\|^2]}_{\text{Lemma B.2}} - \mathbb{E}[\|z^i - z(x^i)\|^2] + \underbrace{\mathbb{E}[\|z^{i+1} - z(x^{i+1}, y^{i+1})\|^2]}_{\text{Lemma B.3}} - \mathbb{E}[\|z^i - z(x^i, y^i)\|^2]$$

$$\leq -\frac{\alpha_i}{2}\mathbb{E}[\|\nabla f(x^i)\|^2] - \left(\frac{\alpha_i}{2} - \frac{L_F\alpha_i^2}{2}\right)\mathbb{E}[\|\bar{g}_{f_1}^i\|^2] + \tilde{\omega}\alpha_i^2$$

$$+ \alpha_i L_{F_{yz}}^2 \mathbb{E}[\|y(x^i) - y^{i+1}\|^2] + \alpha_i L_{F_{yz}}^2 \mathbb{E}[\|z(x^i) - z^{i+1}\|^2]$$

$$+ \left(1 + 2\phi_i + \frac{L_{\nabla y}\alpha_i^2\zeta}{2}\right)\mathbb{E}[\|y^{i+1} - y(x^i)\|^2]$$

$$+ \left(2L_y^2 + \frac{L_y^2}{2\phi_i} + \frac{L_{\nabla y}}{2}\right)\alpha_i^2\mathbb{E}[\|\bar{g}_{f_1}^i\|^2] + \left(2L_y^2 + \frac{L_{\nabla y}}{2}\right)\tau\alpha_i^2$$

$$+ \left(1 + 2\kappa_i + \frac{L_{\nabla z}\alpha_i^2\zeta}{2}\right)\mathbb{E}[\|z^{i+1} - z(x^i)\|^2]$$

$$+ \left(2L_z^2 + \frac{L_z^2}{2\kappa_i} + \frac{L_{\nabla z}}{2}\right)\alpha_i^2\mathbb{E}[\|\bar{g}_{f_1}^i\|^2] + \left(2L_z^2 + \frac{L_{\nabla z}}{2}\right)\tau\alpha_i^2$$

$$+ \left(1 + 2(\eta_i + \upsilon_i) + \frac{L_{z_{xy}}\alpha_i^2\zeta}{2} + \frac{J^2\Upsilon L_{z_{xy}}\beta_i^2}{2}\right)\mathbb{E}[\|z^{i+1} - z(x^i, y^i)\|^2]$$

$$+ \left(2L_{z_{xy}}^2 + \frac{L_{z_{xy}}^2}{2\eta_i} + \frac{L_{z_{xy}}}{2}\right)\alpha_i^2\mathbb{E}[\|\bar{g}_{f_1}^i\|^2] + \left(2L_{z_{xy}}^2 + \frac{L_{z_{xy}}}{2}\right)\alpha_i^2\tau + \left(1 + 2L_{z_{xy}} + \frac{L_{z_{xy}}}{\upsilon_i}\right)\frac{J^2\Upsilon L_{z_{xy}}\beta_i^2}{2}$$

$$- \mathbb{E}[\|y^i - y(x^i)\|^2] - \mathbb{E}[\|z^i - z(x^i)\|^2] - \mathbb{E}[\|z^i - z(x^i, y^i)\|^2].$$

Simplifying, we have

$$\mathbb{E}[\mathbb{V}^{i+1}] - \mathbb{E}[\mathbb{V}^i] \leq -\frac{\alpha_i}{2}\mathbb{E}[\|\nabla f(x^i)\|^2] - \left(\frac{\alpha_i}{2} - \frac{L_F\alpha_i^2}{2}\right)\mathbb{E}[\|\bar{g}_{f_1}^i\|^2]$$

$$+ \left(1 + \alpha_i L_{F_{yz}}^2 + 2\phi_i + \frac{L_{\nabla y}\alpha_i^2\zeta}{2}\right)\underbrace{\mathbb{E}[\|y^{i+1} - y(x^i)\|^2]}_{\text{Lemma B.4}} \tag{C.15}$$

$$+ \left(1 + \alpha_i L_{F_{yz}}^2 + 2\kappa_i + \frac{L_{\nabla z}\alpha_i^2\zeta}{2}\right)\underbrace{\mathbb{E}[\|z^{i+1} - z(x^i)\|^2]}_{\text{Lemma B.2}} \tag{C.16}$$

$$+ \left(1 + 2(\eta_i + \upsilon_i) + \frac{L_{z_{xy}}\alpha_i^2\zeta}{2} + \frac{J^2\Upsilon L_{z_{xy}}\beta_i^2}{2}\right)\underbrace{\mathbb{E}[\|z^{i+1} - z(x^i, y^i)\|^2]}_{\text{Lemma B.3}}$$
$$\tag{C.17}$$

$$+ \left(2L_y^2 + \frac{L_y^2}{2\phi_i} + \frac{L_{\nabla y}}{2} + 2L_z^2 + \frac{L_z^2}{2\kappa_i} + \frac{L_{\nabla z}}{2} + 2L_{z_{xy}}^2 + \frac{L_{z_{xy}}^2}{2\eta_i} + \frac{L_{z_{xy}}}{2}\right)\alpha_i^2\mathbb{E}[\|\bar{g}_{f_1}^i\|^2]$$
$$\tag{C.18}$$

$$+ \left(\left(2L_y^2 + \frac{L_{\nabla y}}{2} + 2L_z^2 + \frac{L_{\nabla z}}{2} + 2L_{z_{xy}}^2 + \frac{L_{z_{xy}}}{2}\right)\tau + \tilde{\omega}\right)\alpha_i^2 \tag{C.19}$$

$$+ \left(1 + 2L_{z_{xy}} + \frac{L_{z_{xy}}}{\upsilon_i}\right)\frac{J^2\Upsilon L_{z_{xy}}\beta_i^2}{2}$$

$$- \mathbb{E}[\|y^i - y(x^i)\|^2] - \mathbb{E}[\|z^i - z(x^i)\|^2] - \mathbb{E}[\|z^i - z(x^i, y^i)\|^2].$$

Now, for ease of notation, we denote the coefficients in equation C.15–equation C.19 as follows:

$$G_1^i := \left(1 + \alpha_i L_{F_{yz}}^2 + 2\phi_i + \frac{L_{\nabla y}\alpha_i^2\zeta}{2}\right), \tag{C.20}$$

$$G_2^i := \left(1 + \alpha_i L_{F_{yz}}^2 + 2\kappa_i + \frac{L_{\nabla z}\alpha_i^2\zeta}{2}\right), \tag{C.21}$$

$$G_3^i := \left(1 + 2\left(\eta_i + \upsilon_i\right) + \frac{L_{z_{xy}}\alpha_i^2\zeta}{2} + \frac{J^2\Upsilon L_{z_{xy}}\beta_i^2}{2}\right), \tag{C.22}$$

$$G_4^i := \left(2L_y^2 + \frac{L_y^2}{2\phi_i} + \frac{L_{\nabla y}}{2} + 2L_z^2 + \frac{L_z^2}{2\kappa_i} + \frac{L_{\nabla z}}{2} + 2L_{z_{xy}}^2 + \frac{L_{z_{xy}}^2}{2\eta_i} + \frac{L_{z_{xy}}}{2}\right), \tag{C.23}$$

$$\Phi := \left(\left(2L_y^2 + \frac{L_{\nabla y}}{2} + 2L_z^2 + \frac{L_{\nabla z}}{2} + 2L_{z_{xy}}^2 + \frac{L_{z_{xy}}}{2}\right)\tau + \tilde{\omega}\right). \tag{C.24}$$

Then, using these definitions and applying Lemmas B.2, B.3, and B.4, we have

$$\mathbb{E}[\mathbb{V}^{i+1}] - \mathbb{E}[\mathbb{V}^i] \leq -\frac{\alpha_i}{2}\mathbb{E}[\|\nabla f(x^i)\|^2] - \left(\frac{\alpha_i}{2} - \frac{L_F\alpha_i^2}{2} - G_4^i\alpha_i^2\right)\mathbb{E}[\|\bar{g}_{f_1}^i\|^2] + \Phi\alpha_i^2$$

$$+ G_1^i\left(1 - \psi_i\beta_i\right)^J \mathbb{E}[\|y^i - y(x^i)\|^2] + G_1^i\left(1 + \frac{1}{2}(J-1)\hat{\hat{\eta}}_i L_{z_y}^2\right)J\Upsilon\beta_i^2 + G_1^i\frac{J+1}{2}JK\gamma_i^2\sigma_{\nabla f_3}^2$$

$$+ G_1^i\left(1 - \gamma_i\rho_{f_3}\right)^K\left(1 + \left(\frac{\vartheta_i - \vartheta_i^J}{1 - \vartheta_i}\right)\right)\mathbb{E}[\|z^i - z(x^i, y^i)\|^2]$$

$$+ G_2^i\left(1 - \gamma_i\rho_{f_3}\right)^{JK}\mathbb{E}[\|z^i - z(x^i)\|^2] + G_2^i JK\gamma_i^2\sigma_{\nabla f_3}^2$$

$$+ G_3^i\left(1 - \gamma_i\rho_{f_3}\right)^{JK}\mathbb{E}[\|z^i - z(x^i, y^i)\|^2] + G_3^i JK\gamma_i^2\sigma_{\nabla f_3}^2$$

$$+ \left(1 + 2L_{z_{xy}} + \frac{L_{z_{xy}}}{\upsilon_i}\right)\frac{J^2\Upsilon L_{z_{xy}}\beta_i^2}{2}$$

$$- \mathbb{E}[\|y^i - y\left(x^i\right)\|^2] - \mathbb{E}[\|z^i - z\left(x^i\right)\|^2] - \mathbb{E}[\|z^i - z\left(x^i, y^i\right)\|^2].$$

Simplifying once again while using the fact that $(1 - \gamma_i\rho_{f_3})^{JK} \leq (1 - \gamma_i\rho_{f_3})^K$ (recalling that $\rho_{f_3}$ from Lemma B.2 and $\gamma_i$ are positive) as well as $J - 1 \leq J$, we have

$$\mathbb{E}[\mathbb{V}^{i+1}] - \mathbb{E}[\mathbb{V}^i]$$

$$\leq -\frac{\alpha_i}{2}\mathbb{E}[\|\nabla f(x^i)\|^2] + \Phi\alpha_i^2 - \underbrace{\left(\frac{\alpha_i}{2} - \frac{L_F\alpha_i^2}{2} - G_3^i\alpha_i^2\right)}_{A_1}\mathbb{E}[\|\bar{g}_{f_1}^i\|^2]$$

$$+ \underbrace{\left(\left(G_1^i J + 2\right)\left(1 - \gamma_i\rho_{f_3}\right)^K - 1\right)}_{A_2}\mathbb{E}[\|z^i - z(x^i, y^i)\|^2]$$

$$+ \underbrace{\left(G_1^i\left(1 - \psi_i\beta_i\right)^J - 1\right)}_{A_3}\mathbb{E}[\|y^i - y(x^i)\|^2] + \underbrace{\left(G_2^i\left(1 - \gamma_i\rho_{f_3}\right)^{JK} - 1\right)}_{A_4}\mathbb{E}[\|z^i - z(x^i)\|^2]$$

$$+ \left(2JL_{z_{xy}}^2 + \left(1 + \frac{1}{2}J\hat{\eta}_i L_{z_y}^2\right)G_1^i\right)J\Upsilon\beta_i^2 + \left(G_1^i\frac{J+1}{2} + G_2^i + 2\right)JK\gamma_i^2\sigma_{\nabla f_3}^2, \tag{C.25}$$

(**Analysis of $A_1$**): Now, consider the coefficient $A_1$ of the $\mathbb{E}[\|\bar{g}_{f_1}^i\|^2]$ term in equation C.25. We wish to determine an appropriate bound on $\alpha_i$ such that this term in non-negative. To that end, we wish to ensure that $A_1 \geq 0$, which is true if

$$\frac{1}{2} - \frac{L_y^2\alpha_i}{2\phi_i} - \frac{L_z^2\alpha_i}{2\kappa_i} - \left(\frac{L_F}{2} + 2L_y^2 + \frac{L_{\nabla y}}{2} + 2L_z^2 + \frac{L_{\nabla z}}{2} + 4L_{z_{xy}}^2\right)\alpha_i \geq 0.$$

Now, choosing

$$\phi_i = 4L_y^2\alpha_i \quad \text{and} \quad \kappa_i = 4L_z^2\alpha_i, \tag{C.26}$$

we have

$$\alpha_i \le \frac{\frac{1}{4}}{\frac{L_F}{2} + 2L_y^2 + \frac{L_{\nabla y}}{2} + 2L_z^2 + \frac{L_{\nabla z}}{2} + 4L_{z_{xy}}^2} = \frac{1}{2(L_F + 4L_y^2 + L_{\nabla y} + 4L_z^2 + L_{\nabla z} + 8L_{z_{xy}}^2)}. \tag{C.27}$$

(**Defining** $\bar{\alpha}_1$): Now, we will define $\bar{\alpha}_1$ as the largest value that $\alpha_i$ can take, which is defined by the upper-bound equation C.27 and the fact that $\alpha_i \le 1$, i.e.,

$$\bar{\alpha}_1 := \min\left\{1, \frac{1}{2(L_F + 4L_y^2 + L_{\nabla y} + 4L_z^2 + L_{\nabla z} + 8L_{z_{xy}}^2)}\right\}. \tag{C.28}$$

(**Analysis of** $A_2$): Now, consider the coefficient $A_2$ of the $\mathbb{E}[\|z^i - z(x^i, y^i)\|^2]$ term in equation C.25. We wish to determine an appropriate bound on $\gamma_i$ such that this term is non-positive. Now, recall from Lemma B.2 that $\rho_{f_3} = \frac{2\mu_z L_{\nabla f_3}}{\mu_z + L_{\nabla f_3}}$ (see equation C.3 in Appendix C.2) as well as the assumed bound (imposed in Lemmas B.2, B.3, and B.4)

$$\gamma_i \le \frac{1}{\mu_z + L_{\nabla f_3}}. \tag{C.29}$$

Recall the fact that $\gamma_i$ and $\rho_{f_3}$ are positive, along with equation C.29, which ensures that $0 \le 1 - \gamma_i \rho_{f_3} \le 1$. With this, to guarantee that $A_2$ is non-positive, we wish to ensure that

$$\left(G_1^i J + 2\right)\left(1 - \gamma_i \rho_{f_3}\right)^K \le 1, \tag{C.30}$$

Now, recall the fact that $1 + a \le e^a$ for any $a \in \mathbb{R}$. Multiplying both sides of this equation by the quantity $\left(1 - \frac{a}{K}\right)^K$, we can see that

$$(1 + a)\left(1 - \frac{a}{K}\right)^K \le e^a\left(1 - \frac{a}{K}\right)^K \le e^a\left(e^{-\frac{a}{K}}\right)^K = e^a e^{-a} = 1. \tag{C.31}$$

Now, to ensure that equation C.30 holds, applying equation C.31 with $a = K\gamma_i \rho_{f_3}$, yields the new inequality we wish to satisfy given by

$$G_1^i J + 2 \le 1 + K\gamma_i \rho_{f_3}. \tag{C.32}$$

Further, using the fact that $\alpha_i \le 1$ along with the choice $\phi_i = 4L_y^2 \alpha_i$, we have

$$G_1^i = 1 + \left(L_{F_{yz}}^2 + 8L_y^2 + \frac{L_{\nabla y}\zeta}{2}\right)\alpha_i. \tag{C.33}$$

Thus, utilizing equation C.33, we see that equation C.32 is satisfied by

$$\left(1 + \left(L_{F_{yz}}^2 + 8L_y^2 + \frac{L_{\nabla y}\zeta}{2}\right)\alpha_i\right)J + 2 \le 1 + K\gamma_i \rho_{f_3},$$

$$1 + J + \tilde{g}_1 J\alpha_i \le K\gamma_i \rho_{f_3}, \tag{C.34}$$

with $\tilde{g}_1 := L_{F_{yz}}^2 + 8L_y^2 + \frac{L_{\nabla y}\zeta}{2}$. Therefore, when choosing $\gamma_i$, and $K$ such that the inequality equation C.34 is satisfied, the coefficient $A_2$ of the $\mathbb{E}[\|z^i - z(x^i, y^i)\|^2]$ term in equation C.25 will be non-positive.

(**Analysis of** $A_3$): Now, consider the coefficient $A_3$ of the $\mathbb{E}[\|y^i - y(x^i)\|^2]$ term in equation C.25. We wish to choose a $\beta_i$ such that this term is non-positive. Recall from the proof of Lemma B.4 that $\rho = \frac{2\mu_y L_{\nabla \bar{f}}}{\mu_y + L_{\nabla \bar{f}}}$ (see equation C.7 in Appendix C.4) and that

$$\beta_i \le \frac{1}{\mu_y + L_{\nabla \bar{f}}}, \qquad \beta_i \le \frac{\rho}{2\hat{\omega}^2 + 1}. \tag{C.35}$$

Further, recall from Lemma B.4 that these two upper-bounds ensure that $0 \le 1 - \psi_i \beta_i \le 1$, where $\psi_i = \rho - 2\hat{\omega}^2 \theta_i^2 - \beta_i$. With this, we wish to ensure that $G_1^i\left(1 - \psi_i \beta_i\right)^J \le 1$. Now, once again using

the fact that $(1 + a)\left(1 - \frac{a}{J}\right)^J \leq 1$ as discussed for the analysis of $A_2$, we need to choose an $a$ such that $0 \leq \frac{a}{J} \leq 1$. Choosing $a = J\psi_i\beta_i$, we have $\frac{a}{J} = \frac{J\psi_i\beta_i}{J} = \psi_i\beta_i$, which from Lemma B.4, we know that $0 \leq \psi_i\beta_i \leq 1$, and by extension that $0 \leq \frac{a}{J} \leq 1$. Thus, we choose $\beta_i$ such that

$$G_1^i \leq 1 + J\psi_i\beta_i \quad \Rightarrow \quad \alpha_i L_{F_{yz}}^2 + 2\phi_i + \frac{L_{\nabla y}\alpha_i^2\zeta}{2} \leq J\psi_i\beta_i. \tag{C.36}$$

(**Analysis of $A_4$**): Now, consider the coefficient $A_4$ of the $\mathbb{E}[\|z^i - z(x^i)\|^2]$ term in equation C.25. We wish to choose a $\gamma_i$ such that this term is non-positive. With this, we have $G_2^i\left(1 - \gamma_i\rho_{f_3}\right)^{JK} \leq 1$. Now, recall that equation equation C.29 ensures $0 \leq (1 - \gamma_i\rho_{f_3})^{JK} \leq 1$. With this, and using the same reasoning that was used earlier, we need to choose $\gamma_i$ such that $G_2^i \leq 1 + JK\gamma_i\rho_{f_3}$, which, when utilizing the fact that $\alpha_i \leq 1$ and the choice $\kappa_i = 4L_z^2\alpha_i$, is satisfied if

$$\tilde{g}_2\alpha_i \leq JK\gamma_i\rho_{f_3}, \quad \text{where} \quad \tilde{g}_2 := L_{F_{yz}}^2 + 8L_z^2 + \frac{L_{\nabla z}\zeta}{2}. \tag{C.37}$$

(**Defining $\bar{\beta}_1$**): Now, from equation equation C.35, define the constant $\bar{\beta}_1$ as the largest value that $\beta_i$ can take, i.e.,

$$\bar{\beta}_1 := \min\left\{1, \frac{1}{\mu_y + L_{\nabla f}}, \frac{\rho}{2\hat{\omega}^2 + 1}\right\}. \tag{C.38}$$

(**Choosing the step-size $\beta_i$**): We need to choose $\beta_i$ to ensure that equation C.36 is satisfied. To that end, using the fact that $\theta_i = \alpha_i\beta_i\gamma_i \leq \beta_i$ (by $\alpha_i \leq 1$ and $\gamma_i \leq 1$) and equation C.38 in the definition of $\psi_i = \rho - 2\hat{\omega}^2\theta_i^2 - \beta_i$, we define the lower-bounding constant $\Gamma$ as

$$\Gamma := \rho - 2\hat{\omega}^2\bar{\beta}_1^2 - \bar{\beta}_1. \tag{C.39}$$

Notice that $0 \leq \Gamma \leq \psi_i$ for all feasible values of $\theta_i$ and $\beta_i$ in $\psi_i$. Now, using this definition of $\Gamma$, along with equation equation C.28 and the fact that $\phi_i = 4L_y^2\alpha_i$, from equation equation C.36, we can choose the ML step-size as

$$\beta_i := \frac{2L_{F_{yz}}^2 + 16L_y^2 + L_{\nabla y}\bar{\alpha}_1\zeta}{2J\Gamma}\alpha_i, \tag{C.40}$$

which follows by keeping one $\alpha_i$ while upper-bounding the other with $\bar{\alpha}_1$ and solving for $\beta_i$ in equation C.36. It bears mentioning that this choice of $\beta_i$ still needs to satisfy the bound $\beta_i \leq \bar{\beta}_1$, which can be satisfied by choosing a sufficiently small $\alpha_i$, which will be defined as the upper-bound $\bar{\alpha}_2$ below in equation equation C.55.

(**Defining $\bar{\gamma}_1$**): Now, from equation equation C.29, we can define the constant $\bar{\gamma}_1$ as the largest value that $\gamma_i$ can take as

$$\bar{\gamma}_1 := \min\left\{1, \frac{1}{\mu_z + L_{\nabla f_3}}\right\}. \tag{C.41}$$

(**Choosing the step-size $\gamma_i$**): From equations equation C.34 and equation C.37, we need to satisfy the lower-bound

$$\max\left\{\frac{1 + J}{K\rho_{f_3}} + \frac{\tilde{g}_1J\alpha_i}{K\rho_{f_3}}, \frac{\tilde{g}_2\alpha_i}{JK\rho_{f_3}}\right\} \leq \gamma_i. \tag{C.42}$$

Now, we can rearrange the left-hand side of this inequality by multiplying by $1 = \frac{J\sqrt{K}}{J\sqrt{K}}$, yielding

$$\Pi := \frac{1}{J\sqrt{K}}\max\left\{\frac{J(1 + J)}{\sqrt{K}\rho_{f_3}} + \frac{\tilde{g}_1J^2\alpha_i}{\sqrt{K}\rho_{f_3}}, \frac{\tilde{g}_2\alpha_i}{\sqrt{K}\rho_{f_3}}\right\} \leq \gamma_i. \tag{C.43}$$

Further, we wish to define $\gamma_i$ as some constant multiple of $\alpha_i$. This can be accomplished by imposing the following appropriate bound on $\alpha_i$, which can be enforced by a sufficiently large enough choice of $K$:

$$\frac{J(1 + J)}{\sqrt{K}\rho_{f_3}} \leq \alpha_i. \tag{C.44}$$

Thus, under this bound on $\alpha_i$, it is clear that we can replace the lower-bound equation C.43 with a more restrictive bound on $\gamma_i$ (but which is defined as a constant multiple of $\alpha_i$) in the following way:

$$\Pi \leq \frac{\max\left\{1 + \frac{\tilde{g}_1 J^2}{\sqrt{K}\rho_{f_3}}, \frac{\tilde{g}_2}{\sqrt{K}\rho_{f_3}}\right\}}{J\sqrt{K}}\alpha_i \leq \gamma_i.$$

Therefore, we choose the LL step-size as this new lower-bound, i.e.,

$$\gamma_i := \frac{\varrho(J, K)}{J\sqrt{K}}\alpha_i, \tag{C.45}$$

where the function $\varrho : \mathbb{N}_+ \times \mathbb{N}_+ \to \mathbb{R}_+$ is defined as

$$\varrho(J, K) := \max\left\{1 + \frac{\tilde{g}_1 J^2}{\sqrt{K}\rho_{f_3}}, \frac{\tilde{g}_2}{\sqrt{K}\rho_{f_3}}\right\}. \tag{C.46}$$

In order to ensure that $\varrho(J, K)$ does not grow to infinity as $J$ increases, we can impose the lower-bound on $K$ of

$$K \geq \frac{\tilde{g}_1^2 J^4}{\rho_{f_3}^2}. \tag{C.47}$$

Now, it bears mentioning that this choice of $\gamma_i$ defined by equation C.45 still needs to satisfy the bound $\gamma_i \leq \bar{\gamma}_1$, which can be ensured by choosing a sufficiently small $\alpha_i$, which will be defined as the upper-bound $\bar{\alpha}_3$ below in equation equation C.56.

(**Upper-bounding $\hat{\eta}_i$**): We need an upper-bound on the positive quantity $\hat{\eta}_i$ in the second to last term of equation C.25. Specifically, we wish to upper bound the term given by

$$\hat{\eta}_i = 1 + \frac{1}{\eta_i}. \tag{C.48}$$

Further, recall that $0 \leq (1 - \gamma_i \rho_{f_3})^K + \eta_i \leq 1$ from the assumed bound (imposed in Lemma B.4)

$$\eta_i \leq 1 - (1 - \gamma_i \rho_{f_3})^K, \tag{C.49}$$

on the positive quantity $\eta_i > 0$. To ensure that bound equation C.49 is always satisfied, we can start by choosing $\eta_i$ to be

$$\eta_i := \mathcal{E}(1 - (1 - \gamma_i \rho_{f_3})^K), \tag{C.50}$$

for some constant $0 < \mathcal{E} < 1$. Thus, we want to derive an upper-bound on the term $1/\eta_i$. Recall that $1 + a \leq e^a$ for all $a \in \mathbb{R}$. Recalling equation C.45 and letting $\hat{a} := \frac{\varrho(J,K)\rho_{f_3}}{J\sqrt{K}}\alpha_i$, we have that $(1 - \hat{a})^K \leq e^{-K\hat{a}}$. For simplification, let $\bar{a} = K\hat{a} = \frac{\sqrt{K}\varrho(J,K)\rho_{f_3}}{J}\alpha_i$. Further, multiplying both sides of the inequality by $-1$ and adding 1 to both sides, we obtain $1 - (1 - \hat{a})^K \geq 1 - e^{-\bar{a}}$. Lastly, multiplying by $\mathcal{E}$ and inverting, we obtain the inequality

$$\frac{1}{\eta_i} \leq \frac{1}{\mathcal{E}(1 - e^{-\bar{a}})}. \tag{C.51}$$

Notice that the right-hand side of this inequality decreases when $\bar{a} \to \infty$ and increases toward infinity as $\bar{a} \to 0$. Therefore, we can derive an upper-bound on $1/\eta_i$ by analyzing the limiting behavior of $\bar{a} \to 0$. To that end, we can begin by analyzing the behavior of $\varrho(J, K)$ in terms of the combinations of scenarios when $J = 1$ or $J = \infty$ and $K = \frac{\tilde{g}_1^2 J^4}{\rho_{f_3}^2}$ (lower-bound defined by equation C.47) or $K = \infty$, yielding the bounds

$$1 \leq \varrho(J, K) \leq \max\left\{2, \frac{\tilde{g}_2}{\tilde{g}_1}\right\}. \tag{C.52}$$

Therefore, we can see that when lower-bounding $\varrho(J, K)$ with 1, we have

$$\bar{a} = \frac{\sqrt{K}\varrho(J, K)\rho_{f_3}}{J}\alpha_i \geq \sqrt{K}\rho_{f_3}\frac{\alpha_i}{J}.$$

Therefore, $\bar{a}$ will approach 0 when $J\alpha_i^{-1}$ approaches infinity faster than $K$. Therefore, in order to prevent $\bar{a}$ from reaching 0, we can impose another lower-bound on $K$ in the form of

$$K \geq \frac{J^2}{\alpha_i^2}. \tag{C.53}$$

When imposing this bound, we can see that $\bar{a} \geq \rho_{f_3}$. Therefore, when imposing the bound equation C.53, we will obtain the upper-bound we desire of

$$\frac{1}{\eta_i} \leq \frac{1}{\mathcal{E}(1 - e^{-\rho_{f_3}})}. \tag{C.54}$$

(**Defining $\bar{\alpha}_2$**): Now, in order to ensure that the bound $\beta_i \leq \bar{\beta}_1$ in equation C.38 is satisfied, we can use the step-size choice of $\beta_i$ in equation C.40 along with the upper-bounds on $\beta_i$ defined in equation C.38 to define the upper-bound constant $\bar{\alpha}_2$ as

$$\bar{\alpha}_2 := \frac{2J\Gamma\bar{\beta}_1}{2L_{F_{yz}}^2 + 16L_y^2 + L_{\nabla y}\bar{\alpha}_1\zeta}. \tag{C.55}$$

(**Defining $\bar{\alpha}_3$**): Further, in order to ensure that the bound $\gamma_i \leq \bar{\gamma}_1$ in equation C.41 is satisfied, with the step-size choice of $\gamma_i$ in equation C.45 along with the upper-bounds on $\gamma_i$ defined in equation C.41 and the lower-bound of $1 \leq \varrho(J, K)$ in equation C.52 we can define the upper-bound constant $\bar{\alpha}_3$ as

$$\bar{\alpha}_3 := J\sqrt{K}\bar{\gamma}_1. \tag{C.56}$$

(**Choosing the step-size $\alpha_i$**): Therefore, in order to satisfy conditions equation C.28, equation C.55, and equation C.56 we choose $\alpha_i$ to be

$$\alpha_i := \min\left\{\bar{\alpha}_1, \bar{\alpha}_2, \bar{\alpha}_3, \frac{\alpha_0}{\sqrt{I}}\right\}, \tag{C.57}$$

where $\alpha_0 \in (0, 1]$ is some constant.

(**Identifying the lower-bounds on $K$**): Now, it bears mentioning that with the choice of step-size for $\alpha_i$ given by equation C.57, from bounds equation C.44, equation C.47, and equation C.53, we can write the consolidated lower-bound that we require $K$ to satisfy as

$$K \geq \max\left\{\frac{J^2(1 + J)^2}{\rho_{f_3}^2 \min\left\{\bar{\alpha}_1, \bar{\alpha}_2, \bar{\alpha}_3, \frac{\alpha_0}{\sqrt{I}}\right\}^2}, \frac{\tilde{g}_1^2 J^4}{\rho_{f_3}^2}, \frac{J^2}{\min\left\{\bar{\alpha}_1, \bar{\alpha}_2, \bar{\alpha}_3, \frac{\alpha_0}{\sqrt{I}}\right\}^2}\right\}. \tag{C.58}$$

Notice that only $\bar{\alpha}_3$ on the right-hand side of equation C.58 is the only term with a $K$ in it. However, since $\lim_{K\to\infty} \bar{\alpha}_3 = \infty$, it will either have no impact on the right-hand side or it will act to potentially decrease the right-hand side, making the bound on $K$ less restrictive. Thus, we can lower-bound $\bar{\alpha}_3$ by the positive constant $\hat{\alpha}_3 := \bar{\alpha}_3|_{K=1,J=1} \leq \bar{\alpha}_3$. For the sake of clarity, we can also lower-bound $\bar{\alpha}_2$ by the positive constant $\hat{\alpha}_2 := \bar{\alpha}_2|_{J=1} \leq \bar{\alpha}_2$. Lastly, by using the lower-bound $I \geq 1$, notice that

$$\frac{1}{\min\left\{\bar{\alpha}_1, \hat{\alpha}_2, \hat{\alpha}_3, \frac{\alpha_0}{\sqrt{I}}\right\}^2} = \max\left\{\frac{1}{\bar{\alpha}_1^2}, \frac{1}{\hat{\alpha}_2^2}, \frac{1}{\hat{\alpha}_3^2}, \frac{I}{\alpha_0^2}\right\} = I\max\left\{\frac{1}{I\bar{\alpha}_1^2}, \frac{1}{I\hat{\alpha}_2^2}, \frac{1}{I\hat{\alpha}_3^2}, \frac{1}{\alpha_0^2}\right\} \leq I\Lambda,$$

where $\Lambda := \max\left\{\frac{1}{\bar{\alpha}_1^2}, \frac{1}{\hat{\alpha}_2^2}, \frac{1}{\hat{\alpha}_3^2}, \frac{1}{\alpha_0^2}\right\}$ is a positive constant independent of $I$, $J$, and $K$. Now, putting this together, we can write equation C.58 alternatively as the bound

$$K \geq \max\left\{\frac{J^2(1 + J)^2 I\Lambda}{\rho_{f_3}^2}, \frac{\tilde{g}_1^2 J^4}{\rho_{f_3}^2}, J^2 I\Lambda\right\}, \tag{C.59}$$

from which it is immediately clear that $K \geq \mathcal{O}(J^4 I)$.

(**Upper-bounding the remaining terms in** *equation C.25*)**:** We wish to upper-bound $G_1^i$ and $G_2^i$. We have seen that $G_1^i$ can be given by equation C.33. Using the fact that $\alpha_i \leq 1$, we have

$$G_1^i \leq 1 + L_{F_{yz}}^2 + 8L_y^2 + \frac{L_{\nabla y}\zeta}{2} := g_1. \tag{C.60}$$

We can bound $G_2^i$ from equation C.21 similarly by using the choice $\kappa_i = 4L_z^2\alpha_i$ along with the fact that $\alpha_i \leq 1$, yielding

$$G_2^i \leq 1 + L_{F_{yz}}^2 + 8L_z^2 + \frac{L_{\nabla z}\zeta}{2} := g_2. \tag{C.61}$$

Therefore, by choosing the step-sizes $\alpha_i$, $\beta_i$, and $\gamma_i$ according to equation C.57, equation C.40, and equation C.45, respectively, it follows that $A_1$ is non-negative while $A_2$, $A_3$, and $A_4$ are non-positive in equation C.25. Therefore, when utilizing the bounds equation C.54, equation C.60, and equation C.61, we can simplify inequality equation C.25 to

$$\mathbb{E}[\mathbb{V}^{i+1}] - \mathbb{E}[\mathbb{V}^i] \leq -\frac{\alpha_i}{2}\mathbb{E}[\|\nabla f(x^i)\|^2] + \Phi\alpha_i^2 + \left(g_1\frac{J+1}{2} + g_2 + 2\right)JK\gamma_i^2\sigma_{\nabla f_3}^2$$

$$+ \left(2JL_{z_{xy}}^2 + \left(1 + J\left(\frac{1}{\mathcal{E}(1 - e^{-\rho_{f_3}})}\right)\right)\frac{L_{z_y}^2}{2}\right)g_1\right)J\Upsilon\beta_i^2$$

$$\leq -\frac{\alpha_i}{2}\mathbb{E}[\|\nabla f(x^i)\|^2] + (\Phi + c_1 + c_2)\alpha_i^2, \tag{C.62}$$

where the last inequality follows from utilizing the step-sizes $\beta_i$ and $\gamma_i$ according to equation C.40 and equation C.45, respectively, along with using the upper-bound defined in equation C.52, where the constants $c_1$ and $c_2$ are defined as the following:

$$c_1 := \left(\frac{g_1}{2} + \left(\frac{g_1}{2} + g_2 + 2\right)\frac{1}{J}\right)\sigma_{\nabla f_3}^2 \max\left\{2, \frac{\tilde{g}_2}{\tilde{g}_1}\right\}^2,$$

$$c_2 := \left(2L_{z_{xy}}^2 + \left(\frac{1}{J} + \left(\frac{1}{\mathcal{E}(1 - e^{-\rho_{f_3}})}\right)\frac{L_{z_y}^2}{2}\right)g_1\right)\frac{\Upsilon(2L_{F_{yz}}^2 + 16L_y^2 + L_{\nabla y}\bar\alpha_1\zeta)^2}{4\Gamma^2}.$$

(**Telescoping**)**:** Now, rearranging equation C.62 and telescoping over $i = 0, 1, ..., I - 1$ leads to

$$\frac{1}{2}\sum_{i=0}^{I-1}\alpha_i\mathbb{E}[\|\nabla f(x^i)\|^2] \leq \mathbb{V}^0 - \mathbb{V}^I + \sum_{i=0}^{I-1}(\Phi + c_1 + c_2)\alpha_i^2. \tag{C.63}$$

Note that $\alpha_i$ is a constant that does not depend on $i$ given by equation C.57. Thus, dividing both sides of equation C.63 by $\frac{1}{2}I\alpha_i$, while noting that $\sum_{i=0}^{I-1}\alpha_i = I\alpha_i$, and considering that $0 \leq \mathbb{V}^i$ for all $i \in \{0, 1, ..., I - 1\}$, we have

$$\frac{1}{I}\sum_{i=0}^{I-1}\mathbb{E}[\|\nabla f(x^i)\|^2] \leq \frac{\mathbb{V}^0 + (\Phi + c_1 + c_2)\sum_{i=0}^{I-1}\alpha_i^2}{\frac{1}{2}I\alpha_i}$$

$$\leq \frac{2\mathbb{V}^0}{I\alpha_i} + \frac{2(\Phi + c_1 + c_2)\alpha_0}{\sqrt{I}}$$

$$\leq \frac{2\mathbb{V}^0}{I\min\{\bar\alpha_1, \bar\alpha_2, \bar\alpha_3\}} + \frac{2\mathbb{V}^0}{\alpha_0\sqrt{I}} + \frac{2(\Phi + c_1 + c_2)\alpha_0}{\sqrt{I}},$$

where the second inequality follows from $\alpha_i \leq \frac{\alpha_0}{\sqrt{I}}$ and the last inequality follows from

$$\frac{2\mathbb{V}^0}{I\alpha_i} = \frac{2\mathbb{V}^0}{I}\left(\frac{1}{\min\left\{\bar\alpha_1, \bar\alpha_2, \bar\alpha_3, \frac{\alpha_0}{\sqrt{I}}\right\}}\right) \leq \frac{2\mathbb{V}^0}{I}\left(\frac{1}{\min\{\bar\alpha_1, \bar\alpha_2, \bar\alpha_3\}} + \frac{1}{\frac{\alpha_0}{\sqrt{I}}}\right) = \frac{2\mathbb{V}^0}{I\min\{\bar\alpha_1, \bar\alpha_2, \bar\alpha_3\}} + \frac{2\mathbb{V}^0}{\alpha_0\sqrt{I}}.$$

Therefore, we have obtained the desired convergence result, completing the proof. $\qquad\square$

## C.6   PROOF OF THEOREM B.6

**Proof.** The start of the proof of this theorem follows the same reasoning that was utilized to obtain equation C.25 in the proof of Theorem B.5 (Appendix C.5), which we restate here as equation C.64:

$$\mathbb{E}[\mathbb{V}^{i+1}] - \mathbb{E}[\mathbb{V}^i]$$

$$\leq -\frac{\alpha_i}{2}\mathbb{E}[\|\nabla f(x^i)\|^2] + \Phi\alpha_i^2 - \underbrace{\left(\frac{\alpha_i}{2} - \frac{L_F\alpha_i^2}{2} - G_3^i\alpha_i^2\right)}_{A_1}\mathbb{E}[\|\bar{g}_{f_1}^i\|^2]$$

$$+ \underbrace{\left((G_1^i J + 2)\left(1 - \gamma_i\rho_{f_3}\right)^K - 1\right)}_{A_2}\mathbb{E}[\|z^i - z(x^i, y^i)\|^2]$$

$$+ \underbrace{\left(G_1^i\left(1 - \psi_i\beta_i\right)^J - 1\right)}_{A_3}\mathbb{E}[\|y^i - y(x^i)\|^2] + \underbrace{\left(G_2^i\left(1 - \gamma_i\rho_{f_3}\right)^{JK} - 1\right)}_{A_4}\mathbb{E}[\|z^i - z(x^i)\|^2]$$

$$+ \left(2JL_{z_{xy}}^2 + \left(1 + \frac{1}{2}J\hat{\eta}_iL_{z_y}^2\right)G_1^i\right)J\Upsilon\beta_i^2 + \left(G_1^i\frac{J+1}{2} + G_2^i + 2\right)JK\gamma_i^2\sigma_{\nabla f_3}^2. \quad \text{(C.64)}$$

The definitions of $G_1^i$, $G_2^i$, $G_3^i$, and $\Phi$ are restated here as

$$G_1^i := \left(1 + \alpha_i L_{F_{yz}}^2 + 2\phi_i + \frac{L_{\nabla y}\alpha_i^2\zeta}{2}\right), \quad \text{(C.65)}$$

$$G_2^i := \left(1 + \alpha_i L_{F_{yz}}^2 + 2\kappa_i + \frac{L_{\nabla z}\alpha_i^2\zeta}{2}\right), \quad \text{(C.66)}$$

$$G_3^i := \left(2L_y^2 + \frac{L_y^2}{2\phi_i} + \frac{L_{\nabla y}}{2} + 2L_z^2 + \frac{L_z^2}{2\kappa_i} + \frac{L_{\nabla z}}{2} + 4L_{z_{xy}}^2\right), \quad \text{(C.67)}$$

$$\Phi := \left(\left(2L_y^2 + \frac{L_{\nabla y}}{2} + 2L_z^2 + \frac{L_{\nabla z}}{2} + 4L_{z_{xy}}^2\right)\tau + \tilde{\omega}\right). \quad \text{(C.68)}$$

(**Choice of step sizes**): In the proof of this theorem, we choose the UL, ML, and LL step sizes to be the following:

$$\alpha_i := \frac{1}{\sqrt{I}}, \quad \text{(C.69)}$$

$$\beta_i := \frac{1}{\sqrt{J}}\alpha_i = \frac{1}{\sqrt{I}\sqrt{J}}, \quad \text{(C.70)}$$

$$\gamma_i := \frac{1}{\sqrt{J}\sqrt{K}}\alpha_i = \frac{1}{\sqrt{I}\sqrt{J}\sqrt{K}}. \quad \text{(C.71)}$$

(**Analysis of $A_1$**): Now, consider the coefficient $A_1$ of the $\mathbb{E}[\|\bar{g}_{f_1}^i\|^2]$ term in equation C.64. We wish to determine an appropriate bound on $\alpha_i$ (in terms of $I$) such that this term in non-negative. To that end, we wish to ensure that $A_1 \geq 0$, which is true if

$$\frac{1}{2} - \frac{L_y^2\alpha_i}{2\phi_i} - \frac{L_z^2\alpha_i}{2\kappa_i} - \left(\frac{L_F}{2} + 2L_y^2 + \frac{L_{\nabla y}}{2} + 2L_z^2 + \frac{L_{\nabla z}}{2} + 4L_{z_{xy}}^2\right)\alpha_i \geq 0.$$

Now, choosing

$$\phi_i = 4L_y^2\alpha_i \quad \text{and} \quad \kappa_i = 4L_z^2\alpha_i, \quad \text{(C.72)}$$

we have

$$\alpha_i \leq \frac{\frac{1}{4}}{\frac{L_F}{2} + 2L_y^2 + \frac{L_{\nabla y}}{2} + 2L_z^2 + \frac{L_{\nabla z}}{2} + 4L_{z_{xy}}^2} = \frac{1}{2(L_F + 4L_y^2 + L_{\nabla y} + 4L_z^2 + L_{\nabla z} + 8L_{z_{xy}}^2)}. \quad \text{(C.73)}$$

Now, recalling our choice for $\alpha_i$ given by equation C.69, then from equation C.73 we see that we must choose $I \in \mathbb{N}$ such that

$$4(L_F + 4L_y^2 + L_{\nabla y} + 4L_z^2 + L_{\nabla z} + 8L_{z_{xy}}^2)^2 \le I. \tag{C.74}$$

Therefore, when choosing $I$ such that the inequality equation C.74 is satisfied, the coefficient $A_1$ of the $\mathbb{E}[\|\bar{g}_{f_1}^i\|^2]$ term in equation C.64 will be non-negative.

(**Analysis of $A_2$**): Now, consider the coefficient $A_2$ of the $\mathbb{E}[\|z^i - z(x^i, y^i)\|^2]$ term in equation C.64. We wish to determine an appropriate bound on $\gamma_i$ (in terms of $I$, $J$, and $K$) such that this term is non-positive. Now, recall from Lemma B.2 that $\rho_{f_3} = \frac{2\mu_z L_{\nabla f_3}}{\mu_z + L_{\nabla f_3}}$ (see equation C.3 in Appendix C.2) as well as the assumed bound (imposed in Lemmas B.2, B.3, and B.4)

$$\gamma_i \le \frac{1}{\mu_z + L_{\nabla f_3}}. \tag{C.75}$$

Utilizing our choice of $\gamma_i$ given by equation C.71, this can be satisfied by choosing $I$, $J$, and $K$ such that

$$(\mu_z + L_{\nabla f_3})^2 \le IJK, \tag{C.76}$$

Recall the fact that $\gamma_i$ and $\rho_{f_3}$ are positive, along with equation C.75, which ensures that $0 \le 1 - \gamma_i \rho_{f_3} \le 1$. With this, to guarantee that $A_2$ is non-positive, we wish to ensure that

$$(G_1^i J + 2)(1 - \gamma_i \rho_{f_3})^K \le 1, \tag{C.77}$$

Now, recall the fact that $1 + a \le e^a$ for any $a \in \mathbb{R}$. Multiplying both sides of this equation by the quantity $\left(1 - \frac{a}{K}\right)^K$, we can see that

$$(1 + a)\left(1 - \frac{a}{K}\right)^K \le e^a \left(1 - \frac{a}{K}\right)^K \le e^a \left(e^{-\frac{a}{K}}\right)^K = e^a e^{-a} = 1. \tag{C.78}$$

Now, to ensure that equation C.77 holds, applying equation C.78 with $a = K\gamma_i \rho_{f_3}$, yields the new inequality we wish to satisfy given by

$$G_1^i J + 2 \le 1 + K\gamma_i \rho_{f_3}. \tag{C.79}$$

Now, using the fact that $\alpha_i \le 1$ along with the choice $\phi_i = 4L_y^2 \alpha_i$, we can upper-bound $G_1^i$ as

$$G_1^i = 1 + \alpha_i L_{F_{yz}}^2 + 2\phi_i + \frac{L_{\nabla y}\alpha_i^2 \zeta}{2} \le 1 + L_{F_{yz}}^2 + 8L_y^2 + \frac{L_{\nabla y}\zeta}{2} := g_1. \tag{C.80}$$

Thus, utilizing equation C.80, we can guarantee equation C.79 if $Jg_1 + 1 \le K\gamma_i \rho_{f_3}$ is satisfied. Now, utilizing the choice of $\gamma_i$ given by equation C.71, we have

$$\frac{Jg_1 + 1}{\rho_{f_3}} \le \frac{K}{\sqrt{I}\sqrt{J}\sqrt{K}} \quad \Rightarrow \quad \frac{IJ(Jg_1 + 1)^2}{\rho_{f_3}^2} \le K. \tag{C.81}$$

Therefore, when choosing $I$, $J$, and $K$ such that the inequality equation C.81 is satisfied, the coefficient $A_2$ of the $\mathbb{E}[\|z^i - z(x^i, y^i)\|^2]$ term in equation C.64 will be non-positive.

(**Analysis of $A_3$**): Now, consider the coefficient $A_3$ of the $\mathbb{E}[\|y^i - y(x^i)\|^2]$ term in equation C.64. We wish to determine an appropriate bound on $\beta_i$ (in terms of $I$ and $J$) such that this term is non-positive. Recall from the proof of Lemma B.4 that $\rho = \frac{2\mu_y L_{\nabla \bar{f}}}{\mu_y + L_{\nabla \bar{f}}}$ (see equation C.7 in Appendix C.4) and that

$$\beta_i \le \frac{1}{\mu_y + L_{\nabla \bar{f}}}, \qquad \beta_i \le \frac{\rho}{2\hat{\omega}^2 + 1}. \tag{C.82}$$

Utilizing our choice of $\beta_i$ given by equation C.70, this can be satisfied by choosing $I$ and $J$ such that

$$\max\left\{\mu_y + L_{\nabla \bar{f}}, \frac{2\hat{\omega}^2 + 1}{\rho}\right\}^2 \le IJ. \tag{C.83}$$

Further, recall from Lemma B.4 that these two upper-bounds ensure that $0 \le 1 - \psi_i \beta_i \le 1$, where $\psi_i = \rho - 2\hat{\omega}^2 \theta_i^2 - \beta_i$. With this, we wish to ensure that $G_1^i (1 - \psi_i \beta_i)^J \le 1$. Now, once again using

the fact that $(1 + a)\left(1 - \frac{a}{J}\right)^J \leq 1$ as discussed for the analysis of $A_2$, we need to choose an $a$ such that $0 \leq \frac{a}{J} \leq 1$. Choosing $a = J\psi_i\beta_i$, we have $\frac{a}{J} = \frac{J\psi_i\beta_i}{J} = \psi_i\beta_i$, which from Lemma B.4, we know that $0 \leq \psi_i\beta_i \leq 1$, and by extension that $0 \leq \frac{a}{J} \leq 1$. Thus, we have

$$G_1^i \leq 1 + J\psi_i\beta_i \quad \Rightarrow \quad \alpha_i L_{F_{yz}}^2 + 2\phi_i + \frac{L_{\nabla y}\alpha_i^2\zeta}{2} \leq J\psi_i\beta_i. \tag{C.84}$$

Notice that from equation equation C.82, $\beta_i$ is upper-bounded by the constant $\bar{\beta}_1$ (defined as the largest value that $\beta_i$ can take) given by

$$\bar{\beta}_1 := \min\left\{1, \frac{1}{\mu_y + L_{\nabla\bar{f}}}, \frac{\rho}{2\hat{\omega}^2 + 1}\right\}. \tag{C.85}$$

Using the fact that $\theta_i = \alpha_i\beta_i\gamma_i \leq \beta_i$ (by $\alpha_i \leq 1$ and $\gamma_i \leq 1$) and equation C.85 in the definition of $\psi_i = \rho - 2\hat{\omega}^2\theta_i^2 - \beta_i$, we can define the new lower-bounding constant $\Gamma$ as

$$\Gamma := \rho - 2\hat{\omega}^2\bar{\beta}_1^2 - \bar{\beta}_1. \tag{C.86}$$

Notice that $0 \leq \Gamma \leq \psi_i$ for all feasible values of $\theta_i$ and $\beta_i$ in $\psi_i$. Now, using this definition of $\Gamma$, the fact that $\alpha_i \leq 1$, and the choice $\phi_i = 4L_y^2\alpha_i$, we have that the following implies equation C.84:

$$\alpha_i\left(L_{F_{yz}}^2 + 8L_y^2 + \frac{L_{\nabla y}\zeta}{2}\right) \leq J\Gamma\beta_i.$$

Utilizing the choices for $\alpha_i$ and $\beta_i$ given by equation C.69 and equation C.70, respectively, it follows that the bound

$$\frac{\left(L_{F_{yz}}^2 + 8L_y^2 + \frac{L_{\nabla y}\zeta}{2}\right)^2}{\Gamma^2} \leq J, \tag{C.87}$$

implies that equation C.84 is satisfied. Therefore, when choosing $J$ such that the inequality equation C.87 is satisfied, the coefficient $A_3$ of the $\mathbb{E}[\|y^i - y(x^i)\|^2]$ term in equation C.64 will be non-positive.

(**Analysis of $A_4$**): Now, consider the coefficient $A_4$ of the $\mathbb{E}[\|z^i - z(x^i)\|^2]$ term in equation C.64. We wish to determine an appropriate bound on $\gamma_i$ (in terms of $I$, $J$, and $K$) such that this term is non-positive. That is, we wish to show $G_2^i\left(1 - \gamma_i\rho_{f_3}\right)^{JK} \leq 1$. Now, recall that equation equation C.75 ensures $0 \leq \left(1 - \gamma_i\rho_{f_3}\right)^{JK} \leq 1$. With this, and using the same reasoning that was used earlier, we need to show that

$$G_2^i \leq 1 + JK\gamma_i\rho_{f_3} \quad \Rightarrow \quad \alpha_i L_{F_{yz}}^2 + 2\kappa_i + \frac{L_{\nabla z}\alpha_i^2\zeta}{2} \leq JK\gamma_i\rho_{f_3}. \tag{C.88}$$

Now, using the fact that $\alpha_i \leq 1$ along with the choice $\kappa_i = 4L_z^2\alpha_i$, we can see that equation C.88 is satisfied if

$$\alpha_i\left(L_{F_{yz}}^2 + 8L_z^2 + \frac{L_{\nabla z}\zeta}{2}\right) \leq JK\gamma_i\rho_{f_3}.$$

Utilizing the choices for $\alpha_i$ and $\gamma_i$ given by equation C.69 and equation C.71, respectively, it follows that the bound

$$\frac{\left(L_{F_{yz}}^2 + 8L_z^2 + \frac{L_{\nabla z}\zeta}{2}\right)^2}{\rho_{f_3}^2} \leq JK, \tag{C.89}$$

implies equation C.88. Therefore, when choosing $J$ and $K$ such that the inequality equation C.89 is satisfied, the coefficient $A_4$ of the $\mathbb{E}[\|z^i - z(x^i)\|^2]$ term in equation C.64 will be non-positive.

(**Upper-bounding $\hat{\eta}_i$**): We need an upper-bound on the positive quantity $\hat{\eta}_i$ in the second to last term of equation C.64. Specifically, we wish to upper bound the term given by

$$\hat{\eta}_i = 1 + \frac{1}{\eta_i}. \tag{C.90}$$

Further, recall that $0 \leq \left(1 - \gamma_i\rho_{f_3}\right)^K + \eta_i \leq 1$ from the assumed bound (imposed in Lemma B.4)

$$\eta_i \leq 1 - \left(1 - \gamma_i\rho_{f_3}\right)^K, \tag{C.91}$$

on the positive quantity $\eta_i > 0$. To ensure that bound equation C.91 is always satisfied, we can start by choosing $\eta_i$ to be

$$\eta_i := \mathcal{E}(1 - (1 - \gamma_i \rho_{f_3})^K), \tag{C.92}$$

for some constant $0 < \mathcal{E} < 1$. When utilizing the choice of $\gamma_i$ given by equation C.71, we have

$$\eta_i := \mathcal{E}\left(1 - \left(1 - \frac{\rho_{f_3}}{\sqrt{I}\sqrt{J}\sqrt{K}}\right)^K\right).$$

Thus, we want to derive an upper-bound on the term $1/\eta_i$. Recall that $1 + a \leq e^a$ for all $a \in \mathbb{R}$. Letting $\hat{a} := \frac{\rho_{f_3}}{\sqrt{I}\sqrt{J}\sqrt{K}}$, we have that $(1 - \hat{a})^K \leq e^{-K\hat{a}}$. For simplification, let $\bar{a} = K\hat{a} = \frac{\sqrt{K}}{\sqrt{I}\sqrt{J}}\rho_{f_3}$. Further, multiplying both sides of the inequality by $-1$ and adding 1 to both sides, we obtain $1 - (1 - \hat{a})^K \geq 1 - e^{-\bar{a}}$. Lastly, multiplying by $\mathcal{E}$ and inverting, we obtain the inequality

$$\eta_i = \mathcal{E}(1 - (1 - \hat{a})^K) \geq \mathcal{E}(1 - e^{-\bar{a}}) \qquad \Longrightarrow \qquad \frac{1}{\eta_i} \leq \frac{1}{\mathcal{E}(1 - e^{-\bar{a}})}. \tag{C.93}$$

It is clear that as $\bar{a} \to \infty$ (i.e., $\sqrt{K}$ approaches infinity faster than $\sqrt{I}\sqrt{J}$) then $\lim_{\bar{a} \to \infty} e^{-\bar{a}} = 0$, leading to the lower-bounding limit of

$$\lim_{K \to \infty} \frac{1}{\mathcal{E}(1 - e^{-\bar{a}})} = \frac{1}{\mathcal{E}} \qquad \Longrightarrow \qquad \frac{1}{\mathcal{E}} \leq \frac{1}{\eta_i} \leq \frac{1}{\mathcal{E}(1 - e^{-\bar{a}})}.$$

Now, notice that the expression $\frac{1}{\mathcal{E}(1-e^{-\bar{a}})}$ grows toward infinity as $\bar{a} \to 0^+$ (which will occur when $\sqrt{I}\sqrt{J}$ approaches infinity faster than $\sqrt{K}$), since $\lim_{\bar{a} \to 0^+} e^{-\bar{a}} = 1$. Therefore, to prevent the term $\bar{a}$ from approaching 0, we can impose the bound

$$IJ \leq K. \tag{C.94}$$

Thus, when imposing bound equation C.94 and considering that $I \geq 1$, $J \geq 1$, and $K \geq 1$, we can see that $\bar{a} = \frac{\sqrt{K}}{\sqrt{I}\sqrt{J}}\rho_{f_3}$ is bounded by

$$\rho_{f_3} \leq \bar{a}. \tag{C.95}$$

Therefore, utilizing the lower-bound in equation C.95 will yield the desired upper-bound on $1/\eta_i$ of

$$\frac{1}{\eta_i} \leq \frac{1}{\mathcal{E}(1 - e^{-\rho_{f_3}})}. \tag{C.96}$$

**(Consolidation of bounds):** To summarize, we choose the step-sizes $\alpha_i$, $\beta_i$, and $\gamma_i$ according to equation C.69, equation C.70, and equation C.71, respectively, as well as impose the following bounds on $I$, $J$, and $K$ (defined by equation C.74, equation C.76, equation C.81, equation C.83, equation C.87, equation C.89, and lastly equation C.94, respectively), restated here for convenience:

$$4(L_F + 4L_y^2 + L_{\nabla y} + 4L_z^2 + L_{\nabla z} + 8L_{z_{xy}}^2)^2 \leq I,$$

$$(\mu_z + L_{\nabla f_3})^2 \leq IJK, \qquad \frac{IJ(Jg_1 + 1)^2}{\rho_{f_3}^2} \leq K, \qquad \max\left\{\mu_y + L_{\nabla \bar{f}}, \frac{2\hat{\omega}^2 + 1}{\rho}\right\}^2 \leq IJ,$$

$$\frac{\left(L_{F_{yz}}^2 + 8L_y^2 + \frac{L_{\nabla y}\zeta}{2}\right)^2}{\Gamma^2} \leq J, \qquad \frac{\left(L_{F_{yz}}^2 + 8L_z^2 + \frac{L_{\nabla z}\zeta}{2}\right)^2}{\rho_{f_3}^2} \leq JK, \qquad IJ \leq K.$$

We can denote the constant lower-bound on $J$ given by equation C.87 as

$$J \geq \varsigma := \frac{\left(L_{F_{yz}}^2 + 8L_y^2 + \frac{L_{\nabla y}\zeta}{2}\right)^2}{\Gamma^2}. \tag{C.97}$$

Using equation C.97, the bounds equation C.74 and equation C.83 are implied by the consolidated bound

$$\varpi \leq I, \tag{C.98}$$

where the constant $\varpi$ is defined as

$$\varpi := \max\left\{ 4(L_F + 4L_y^2 + L_{\nabla y} + 4L_z^2 + L_{\nabla z} + 8L_{z_{xy}}^2)^2, \frac{\max\left\{\mu_y + L_{\nabla \bar{f}}, \frac{2\hat{\omega}^2+1}{\rho}\right\}^2}{\varsigma} \right\}.$$
(C.99)

Similarly, using equation C.97 and equation C.99, we can see that the bounds equation C.76, equation C.81, equation C.89, and equation C.94 are implied by the following consolidated bound

$$\Xi(I, J) \leq K,$$
(C.100)

where the function $\Xi : \mathbb{N}_+ \times \mathbb{N}_+ \to \mathbb{R}_+$ is defined as

$$\Xi(I, J) := \max\left\{ \frac{(\mu_z + L_{\nabla f_3})^2}{\varpi\varsigma}, \frac{IJ(Jg_1 + 1)^2}{\rho_{f_3}^2}, \frac{\left(L_{F_{yz}}^2 + 8L_z^2 + \frac{L_{\nabla z}\varsigma}{2}\right)^2}{\varsigma\rho_{f_3}^2}, IJ \right\},$$
(C.101)

from which it is immediately clear that $K \geq \mathcal{O}(J^3 I)$.

(**Upper-bounding the remaining terms in** *equation C.64*)**:** When choosing the step-sizes $\alpha_i$, $\beta_i$, and $\gamma_i$ according to equation C.69, equation C.70, and equation C.71, respectively, as well as choosing $I$, $J$, and $K$ according to equation C.98, equation C.97, and equation C.100, respectively, it follows that $A_1$ is non-negative while $A_2$, $A_3$, and $A_4$ are non-positive in equation C.64. Thus, we can simplify inequality equation C.64 to

$$\mathbb{E}[\mathbb{V}^{i+1}] - \mathbb{E}[\mathbb{V}^i] \leq -\frac{\alpha_i}{2}\mathbb{E}[\|\nabla f(x^i)\|^2] + \Phi\alpha_i^2 + \left(G_1^i \frac{J+1}{2} + G_2^i + 2\right) JK\gamma_i^2\sigma_{\nabla f_3}^2$$

$$+ \left(2JL_{z_{xy}}^2 + \left(1 + J\left(\frac{1}{\mathcal{E}(1 - e^{-\rho_{f_3}})}\right)\frac{L_{z_y}^2}{2}\right)G_1^i\right) J\Upsilon\beta_i^2$$

$$\leq -\frac{\alpha_i}{2}\mathbb{E}[\|\nabla f(x^i)\|^2] + (\Phi + c_1 + c_2 J)\alpha_i^2,$$
(C.102)

where the last inequality follows from utilizing the step-sizes $\alpha_i$, $\beta_i$, and $\gamma_i$ according to equation C.69, equation C.70, and equation C.71, respectively, as well as the inequality equation C.80, recalling that $g_1 = 1 + L_{F_{yz}}^2 + 8L_y^2 + \frac{L_{\nabla y}\varsigma}{2}$, and defining the upper-bound on $G_2^i$ of $G_2^i \leq 1 + L_{F_{yz}}^2 + 8L_z^2 + \frac{L_{\nabla z}\varsigma}{2} := g_2$ (obtained from equation C.66 by using $\alpha_i \leq 1$ and $\kappa_i = 4L_z^2\alpha_i$). Further, the constants $c_1$ and $c_2$ are defined as

$$c_1 := \sigma_{\nabla f_3}^2\left(\frac{g_1}{2} + g_2 + 2\right) + g_1\Upsilon, \qquad c_2 := 2L_{z_{xy}}^2\Upsilon + \frac{g_1\sigma_{\nabla f_3}^2}{2} + \frac{g_1 L_{z_y}^2 \Upsilon}{2}\left(\frac{1}{\mathcal{E}(1 - e^{-\rho_{f_3}})}\right).$$

(**Telescoping**)**:** Now, rearranging equation C.102 and telescoping over $i = 0, 1, ..., I-1$ leads to

$$\frac{1}{2}\sum_{i=0}^{I-1}\alpha_i\mathbb{E}[\|\nabla f(x^i)\|^2] \leq \mathbb{V}^0 - \mathbb{V}^I + \sum_{i=0}^{I-1}(\Phi + c_1 + c_2 J)\alpha_i^2.$$
(C.103)

Note that $\alpha_i$ is a constant that does not depend on $i$ given by equation C.69. Thus, dividing both sides of equation C.103 by $\frac{1}{2}I\alpha_i$, while noting that $\sum_{i=0}^{I-1}\alpha_i = I\alpha_i$, and considering that $0 \leq \mathbb{V}^i$ for all $i \in \{0, 1, ..., I-1\}$, we have

$$\frac{1}{I}\sum_{i=0}^{I-1}\mathbb{E}[\|\nabla f(x^i)\|^2] \leq \frac{\mathbb{V}^0 + (\Phi + c_1 + c_2 J)\sum_{i=0}^{I-1}\alpha_i^2}{\frac{1}{2}I\alpha_i} = \frac{2\mathbb{V}^0 + 2(\Phi + c_1 + c_2 J)}{\sqrt{I}}.$$

Therefore, we have obtained the desired convergence result, completing the proof. □

# D  BOUNDS ON BIAS, VARIANCE, AND INEXACTNESS

This appendix contains derivations of results that yield bounds on the biasedness and variance of stochastic terms as well as bounds on the sizes, inexactness, and variances of the UL and ML search directions. For ease of notation, since all expectations that are present in the proofs of Lemmas D.1, D.2, and D.3 are conditioned on $\mathcal{F}_\xi$, we utilize the short-hand notation of $\mathbb{E}[\cdot] := \mathbb{E}[\cdot|\mathcal{F}_\xi]$, unless stated otherwise.

**Lemma D.1 (Bounds on bias of $\nabla z$ and $\nabla^2 \bar{f}$)** *Under Assumptions 3.1, 3.2, 3.4, 3.5, and 3.6, the stochastic terms $\nabla_x z^\xi$, $\nabla_y z^\xi$, $\nabla^2_{xy} \bar{f}^\xi$, and $\nabla^2_{yy} \bar{f}^\xi$ estimate $\nabla_x z$, $\nabla_y z$, $\nabla^2_{xy} \bar{f}$, and $\nabla^2_{yy} \bar{f}$, respectively, with biases that are bounded on the order of $\mathcal{O}(\theta)$, i.e., there exist positive constants $U_x$, $U_y$, $U_{xy}$, and $U_{yy}$ such that*

$$\|\nabla_x z(x,y)^\top - \mathbb{E}[\nabla_x z(x,y;\xi)^\top|\mathcal{F}_\xi]\| \leq U_x \theta, \tag{D.1}$$

$$\|\nabla_y z(x,y)^\top - \mathbb{E}[\nabla_y z(x,y;\xi)^\top|\mathcal{F}_\xi]\| \leq U_y \theta, \tag{D.2}$$

$$\|\nabla^2_{xy} \bar{f}(x,y,z) - \mathbb{E}[\nabla^2_{xy} \bar{f}(x,y,z;\xi)|\mathcal{F}_\xi]\| \leq U_{xy} \theta, \tag{D.3}$$

$$\|\nabla^2_{yy} \bar{f}(x,y,z) - \mathbb{E}[\nabla^2_{yy} \bar{f}(x,y,z;\xi)|\mathcal{F}_\xi]\| \leq U_{yy} \theta. \tag{D.4}$$

**Proof.** For this proof, we will omit the point $(x,y,z)$ that the terms are evaluated at; we will simply use a $\xi$-superscript as short-hand to indicate any random terms. We can obtain the bound on the biasedness of the estimator $\nabla_x z(x,y;\xi)$ in equation equation D.1 by utilizing the consistency of norms along with equation A.3 and Assumption 3.4 to obtain

$$\begin{aligned}
\|\nabla_x z(x,y)^\top - \mathbb{E}[\nabla_x z(x,y;\xi)^\top]\| &= \|[\nabla^2_{zz} f_3]^{-1} \nabla^2_{zx} f_3 - \mathbb{E}[[\nabla^2_{zz} f_3^\xi]^{-1}] \nabla^2_{zx} f_3\| \\
&\leq \|\nabla^2_{zx} f_3\| \|[\nabla^2_{zz} f_3]^{-1} - \mathbb{E}[[\nabla^2_{zz} f_3^\xi]^{-1}]\| \\
&\leq L_{\nabla f_3} W_{zz} \theta := U_x \theta, \tag{D.5}
\end{aligned}$$

where the last inequality follows from applying Assumptions 3.1 and 3.6. The proof of biasedness for the estimator $\nabla_y z(x,y;\xi)$ in equation equation D.2 can be established following identical arguments.

Now, to prove the biasedness of the estimator $\nabla^2_{xy} \bar{f}(x,y,z;\xi)$, referencing equations equation A.1 and equation A.5, utilizing Assumption 3.4, applying the triangle inequality along with the consistency of matrix norms, we have

$$\|\nabla^2_{xy} \bar{f} - \mathbb{E}[\nabla^2_{xy} \bar{f}^\xi]\|$$

$$\leq \|\nabla^3_{yzx} f_3\| \|\nabla_z f_2\| \|[\nabla^2_{zz} f_3]^{-1} - \mathbb{E}[[\nabla^2_{zz} f_3^\xi]^{-1}]\| \tag{D.6}$$

$$+ \|\nabla^3_{yzz} f_3\| \|\nabla_z f_2\| \|\nabla_x z^\top [\nabla^2_{zz} f_3]^{-1} - \mathbb{E}[\nabla_x z^{\xi\top} [\nabla^2_{zz} f_3^\xi]^{-1}]\| \tag{D.7}$$

$$+ \|\nabla^2_{yz} f_3\| \|\nabla_z f_2\| \|[\nabla^2_{zz} f_3]^{-1} \nabla^3_{zzx} f_3 [\nabla^2_{zz} f_3]^{-1} - \mathbb{E}[[\nabla^2_{zz} f_3^\xi]^{-1} \nabla^3_{zzx} f_3^\xi [\nabla^2_{zz} f_3^\xi]^{-1}]\| \tag{D.8}$$

$$+ \|\nabla^2_{yz} f_3\| \|\nabla_z f_2\| \|[\nabla^2_{zz} f_3]^{-1} \nabla^3_{zzz} f_3 \nabla_x z^\top [\nabla^2_{zz} f_3]^{-1} - \mathbb{E}[[\nabla^2_{zz} f_3^\xi]^{-1} \nabla^3_{zzz} f_3^\xi \nabla_x z^{\xi\top} [\nabla^2_{zz} f_3^\xi]^{-1}]\|$$
$$\tag{D.9}$$

$$+ \|\nabla^2_{yz} f_3\| \|\nabla^2_{zx} f_2\| \|[\nabla^2_{zz} f_3]^{-1} - \mathbb{E}[[\nabla^2_{zz} f_3^\xi]^{-1}]\| \tag{D.10}$$

$$+ \|\nabla^2_{yz} f_3\| \|[\nabla^2_{zz} f_3]^{-1} \nabla^2_{zz} f_2 \nabla_x z^\top - \mathbb{E}[[\nabla^2_{zz} f_3^\xi]^{-1} \nabla^2_{zz} f_2^\xi \nabla_x z^{\xi\top}]\|. \tag{D.11}$$

Notice that there are six difference terms here. Applying Assumption 3.1 and 3.6, we can bound equations equation D.6 and equation D.10 in the following way:

$$\|\nabla^3_{yzx} f_3\| \|\nabla_z f_2\| \|[\nabla^2_{zz} f_3]^{-1} - \mathbb{E}[[\nabla^2_{zz} f_3^\xi]^{-1}]\| \leq L_{\nabla^2 f_3} L_{f_2} W_{zz} \theta, \tag{D.12}$$

$$\|\nabla^2_{yz} f_3\| \|\nabla^2_{zx} f_2\| \|[\nabla^2_{zz} f_3]^{-1} - \mathbb{E}[[\nabla^2_{zz} f_3^\xi]^{-1}]\| \leq L_{\nabla f_3} L_{\nabla f_2} W_{zz} \theta. \tag{D.13}$$

Now, looking at equation equation D.7, applying Assumption 3.1, adding and subtracting $\nabla_x z^\top \mathbb{E}[[\nabla^2_{zz} f^\xi_3]^{-1}]$, and applying the triangle inequality, we have

$$\|\nabla^3_{yzz} f_3\|\|\nabla_z f_2\|\|\nabla_x z^\top [\nabla^2_{zz} f_3]^{-1} - \mathbb{E}[\nabla_x z^{\xi\top} [\nabla^2_{zz} f^\xi_3]^{-1}]\|$$

$$\leq L_{\nabla^2 f_3} L_{f_2} \|\nabla_x z^\top [\nabla^2_{zz} f_3]^{-1} - \mathbb{E}[\nabla_x z^{\xi\top}] \mathbb{E}[[\nabla^2_{zz} f^\xi_3]^{-1}]\|$$

$$\leq L_{\nabla^2 f_3} L_{f_2} \|\nabla_x z^\top [\nabla^2_{zz} f_3]^{-1} - \nabla_x z^\top \mathbb{E}[[\nabla^2_{zz} f^\xi_3]^{-1}]\|$$

$$\quad + L_{\nabla^2 f_3} L_{f_2} \|\nabla_x z^\top \mathbb{E}[[\nabla^2_{zz} f^\xi_3]^{-1}] - \mathbb{E}[\nabla_x z^{\xi\top}] \mathbb{E}[[\nabla^2_{zz} f^\xi_3]^{-1}]\|$$

$$\leq L_{\nabla^2 f_3} L_{f_2} \|\nabla_x z^\top\|\|[\nabla^2_{zz} f_3]^{-1} - \mathbb{E}[[\nabla^2_{zz} f^\xi_3]^{-1}]\| + L_{\nabla^2 f_3} L_{f_2} \|\mathbb{E}[[\nabla^2_{zz} f^\xi_3]^{-1}]\|\|\nabla_x z^\top - \mathbb{E}[\nabla_x z^{\xi\top}]\|$$

$$\leq L_{\nabla^2 f_3} L_{f_2} \frac{L_{\nabla f_3}}{\mu_z} W_{zz}\theta + L_{\nabla^2 f_3} L_{f_2} b_{zz} U_x \theta \;=\; L_{\nabla^2 f_3} L_{f_2} \left( \frac{W_{zz} L_{\nabla f_3}}{\mu_z} + b_{zz} U_x \right) \theta, \qquad (D.14)$$

where the second-to-last inequality follows from the consistency of norms, and the last inequality follows from applying the derived bound equation D.5, equation equation E.14, and Assumptions 3.5 and 3.6.

Now, looking at equation D.11, applying Assumptions 3.1 and 3.4, adding and subtracting the term $[\nabla^2_{zz} f_3]^{-1} \nabla^2_{zz} f_2 \mathbb{E}[\nabla_x z^{\xi\top}]$, applying the triangle inequality, and using the consistency of matrix norms, we have

$$\|\nabla^2_{yz} f_3\|\|[\nabla^2_{zz} f_3]^{-1} \nabla^2_{zz} f_2 \nabla_x z^\top - \mathbb{E}[[\nabla^2_{zz} f^\xi_3]^{-1} \nabla^2_{zz} f^\xi_2 \nabla_x z^{\xi\top}]\|$$

$$\leq L_{\nabla f_3} \|[\nabla^2_{zz} f_3]^{-1} \nabla^2_{zz} f_2 \nabla_x z^\top - [\nabla^2_{zz} f_3]^{-1} \nabla^2_{zz} f_2 \mathbb{E}[\nabla_x z^{\xi\top}]\|$$

$$\quad + L_{\nabla f_3} \|[\nabla^2_{zz} f_3]^{-1} \nabla^2_{zz} f_2 \mathbb{E}[\nabla_x z^{\xi\top}] - \mathbb{E}[[\nabla^2_{zz} f^\xi_3]^{-1}] \nabla^2_{zz} f_2 \mathbb{E}[\nabla_x z^{\xi\top}]\|$$

$$\leq L_{\nabla f_3} \|[\nabla^2_{zz} f_3]^{-1}\|\|\nabla^2_{zz} f_2\|\|\nabla_x z^\top - \mathbb{E}[\nabla_x z^{\xi\top}]\|$$

$$\quad + L_{\nabla f_3} \|\nabla^2_{zz} f_2\|\|\mathbb{E}[\nabla_x z^{\xi\top}]\|\|[\nabla^2_{zz} f_3]^{-1} - \mathbb{E}[[\nabla^2_{zz} f^\xi_3]^{-1}]\|.$$

Now, applying Assumptions 3.1, 3.2, and 3.6, along with the derived bound equation D.5, we have

$$\|\nabla^2_{yz} f_3\|\|[\nabla^2_{zz} f_3]^{-1} \nabla^2_{zz} f_2 \nabla_x z^\top - \mathbb{E}[[\nabla^2_{zz} f^\xi_3]^{-1} \nabla^2_{zz} f^\xi_2 \nabla_x z^{\xi\top}]\|$$

$$\leq \frac{L_{\nabla f_3} L_{\nabla f_2} U_x}{\mu_z}\theta + L_{\nabla f_3} L_{\nabla f_2} b_{zz} L_{\nabla f_3} W_{zz}\theta \;=\; L_{\nabla f_3} L_{\nabla f_2} \left( \frac{U_x}{\mu_z} + b_{zz} L_{\nabla f_3} W_{zz} \right) \theta, \quad (D.15)$$

where the last inequality follows from $\|\mathbb{E}[\nabla_x z^{\xi\top}]\| = \| - \mathbb{E}[[\nabla^2_{zz} f^\xi_3]^{-1}] \nabla^2_{zx} f_3 \| \leq b_{zz} L_{\nabla f_3}$ (from Assumptions 3.1, 3.4, and 3.5).

Now, looking at equation D.8, applying Assumptions 3.1 and 3.4, adding and subtracting the term $[\nabla^2_{zz} f_3]^{-1} \nabla^3_{zzx} f_3 \mathbb{E}[[\nabla^2_{zz} f^\xi_3]^{-1}]$, applying the triangle inequality, and using the consistency of matrix norms, we have

$$\|\nabla^2_{yz} f_3\|\|\nabla_z f_2\|\|[\nabla^2_{zz} f_3]^{-1} \nabla^3_{zzx} f_3 [\nabla^2_{zz} f_3]^{-1} - \mathbb{E}[[\nabla^2_{zz} f^\xi_3]^{-1} \nabla^3_{zzx} f^\xi_3 [\nabla^2_{zz} f^\xi_3]^{-1}]\|$$

$$\leq L_{\nabla f_3} L_{f_2} \|[\nabla^2_{zz} f_3]^{-1} \nabla^3_{zzx} f_3 [\nabla^2_{zz} f_3]^{-1} - [\nabla^2_{zz} f_3]^{-1} \nabla^3_{zzx} f_3 \mathbb{E}[[\nabla^2_{zz} f^\xi_3]^{-1}]\|$$

$$\quad + L_{\nabla f_3} L_{f_2} \|[\nabla^2_{zz} f_3]^{-1} \nabla^3_{zzx} f_3 \mathbb{E}[[\nabla^2_{zz} f^\xi_3]^{-1}] - \mathbb{E}[[\nabla^2_{zz} f^\xi_3]^{-1}] \nabla^3_{zzx} f_3 \mathbb{E}[[\nabla^2_{zz} f^\xi_3]^{-1}]\|$$

$$\leq L_{\nabla f_3} L_{f_2} \|[\nabla^2_{zz} f_3]^{-1}\|\|\nabla^3_{zzx} f_3\|\|[\nabla^2_{zz} f_3]^{-1} - \mathbb{E}[[\nabla^2_{zz} f^\xi_3]^{-1}]\|$$

$$\quad + L_{\nabla f_3} L_{f_2} \|\nabla^3_{zzx} f_3\|\|\mathbb{E}[[\nabla^2_{zz} f^\xi_3]^{-1}]\|\|[\nabla^2_{zz} f_3]^{-1} - \mathbb{E}[[\nabla^2_{zz} f^\xi_3]^{-1}]\|$$

$$\leq L_{\nabla f_3} L_{f_2} \frac{1}{\mu_z} L_{\nabla^2 f_3} W_{zz}\theta + L_{\nabla f_3} L_{f_2} L_{\nabla^2 f_3} b_{zz} W_{zz}\theta \;=\; L_{\nabla f_3} L_{f_2} L_{\nabla^2 f_3} W_{zz} \left( \frac{1}{\mu_z} + b_{zz} \right) \theta,$$

$$(D.16)$$

where the last inequality follows from applying Assumptions 3.1, 3.2, 3.5, and 3.6.

Now, looking at equation D.9, applying Assumptions 3.1 and 3.4, adding and subtracting the term $[\nabla^2_{zz} f_3]^{-1} \nabla^3_{zzz} f_3 \nabla_x z^\top \mathbb{E}[[\nabla^2_{zz} f^\xi_3]^{-1}]$, applying the triangle inequality, and using the consistency

of matrix norms, we have

$$\|\nabla_{yz}^2 f_3\|\|\nabla_z f_2\|\|[\nabla_{zz}^2 f_3]^{-1}\nabla_{zzz}^3 f_3\nabla_x z^\top[\nabla_{zz}^2 f_3]^{-1} - \mathbb{E}[[\nabla_{zz}^2 f_3^\xi]^{-1}\nabla_{zzz}^3 f_3^\xi\nabla_x z^{\xi\top}[\nabla_{zz}^2 f_3^\xi]^{-1}]\|$$

$$\leq L_{\nabla f_3} L_{f_2}\|[\nabla_{zz}^2 f_3]^{-1}\nabla_{zzz}^3 f_3\nabla_x z^\top[\nabla_{zz}^2 f_3]^{-1} - [\nabla_{zz}^2 f_3]^{-1}\nabla_{zzz}^3 f_3\nabla_x z^\top\mathbb{E}[[\nabla_{zz}^2 f_3^\xi]^{-1}]\|$$

$$\quad + L_{\nabla f_3} L_{f_2}\|[\nabla_{zz}^2 f_3]^{-1}\nabla_{zzz}^3 f_3\nabla_x z^\top\mathbb{E}[[\nabla_{zz}^2 f_3^\xi]^{-1}] - \mathbb{E}[[\nabla_{zz}^2 f_3^\xi]^{-1}]\nabla_{zzz}^3 f_3\mathbb{E}[\nabla_x z^{\xi\top}]\mathbb{E}[[\nabla_{zz}^2 f_3^\xi]^{-1}]\|$$

$$\leq L_{\nabla f_3} L_{f_2}\|[\nabla_{zz}^2 f_3]^{-1}\|\|\nabla_{zzz}^3 f_3\|\|\nabla_x z^\top\|\|[\nabla_{zz}^2 f_3]^{-1} - \mathbb{E}[[\nabla_{zz}^2 f_3^\xi]^{-1}]\|$$

$$\quad + L_{\nabla f_3} L_{f_2}\|\mathbb{E}[[\nabla_{zz}^2 f_3^\xi]^{-1}]\|\|[\nabla_{zz}^2 f_3]^{-1}\nabla_{zzz}^3 f_3\nabla_x z^\top - \mathbb{E}[[\nabla_{zz}^2 f_3^\xi]^{-1}]\nabla_{zzz}^3 f_3\mathbb{E}[\nabla_x z^{\xi\top}]\|$$

$$\leq \frac{L_{\nabla f_3}^2 L_{f_2} L_{\nabla^2 f_3}}{\mu_z^2}W_{zz}\theta + L_{\nabla f_3} L_{f_2} b_{zz}\|[\nabla_{zz}^2 f_3]^{-1}\nabla_{zzz}^3 f_3\nabla_x z^\top - \mathbb{E}[[\nabla_{zz}^2 f_3^\xi]^{-1}]\nabla_{zzz}^3 f_3\mathbb{E}[\nabla_x z^{\xi\top}]\|,$$

where the last inequality follows from applying Assumptions 3.1, 3.2, 3.5, and 3.6, along with equation equation E.14. Now, using nearly identical arguments to those that were used in deriving the bound on equation D.11, we have

$$\|\nabla_{yz}^2 f_3\|\|\nabla_z f_2\|\|[\nabla_{zz}^2 f_3]^{-1}\nabla_{zzz}^3 f_3\nabla_x z^\top[\nabla_{zz}^2 f_3]^{-1} - \mathbb{E}[[\nabla_{zz}^2 f_3^\xi]^{-1}\nabla_{zzz}^3 f_3^\xi\nabla_x z^{\xi\top}[\nabla_{zz}^2 f_3^\xi]^{-1}]\|$$

$$\leq \frac{L_{\nabla f_3}^2 L_{f_2} L_{\nabla^2 f_3}}{\mu_z^2}W_{zz}\theta + L_{\nabla f_3} L_{f_2} b_{zz}\left(L_{\nabla^2 f_3}\left(\frac{U_x}{\mu_z} + b_{zz}L_{\nabla f_3}W_{zz}\right)\theta\right)$$

$$= L_{\nabla f_3} L_{f_2} L_{\nabla^2 f_3}\left(\frac{L_{\nabla f_3}}{\mu_z^2}W_{zz} + b_{zz}\left(\frac{U_x}{\mu_z} + b_{zz}L_{\nabla f_3}W_{zz}\right)\right)\theta. \tag{D.17}$$

Finally, substituting the newly-derived upper-bounds equation D.12–equation D.17 in for the terms equation D.6–equation D.11, we have the desired upper bound equation D.3 as

$$\|\nabla_{xy}^2\bar f - \mathbb{E}[\nabla_{xy}^2\bar f^\xi]\| \leq L_{\nabla^2 f_3}L_{f_2}W_{zz}\theta + L_{\nabla^2 f_3}L_{f_2}\left(\frac{W_{zz}L_{\nabla f_3}}{\mu_z} + b_{zz}U_x\right)\theta + L_{\nabla f_3}L_{\nabla f_2}W_{zz}\theta$$

$$\quad\quad + L_{\nabla f_3}L_{f_2}L_{\nabla^2 f_3}\left(\frac{L_{\nabla f_3}}{\mu_z^2}W_{zz} + b_{zz}\left(\frac{U_x}{\mu_z} + b_{zz}L_{\nabla f_3}W_{zz}\right)\right)\theta$$

$$\quad\quad + L_{\nabla f_3}L_{\nabla f_2}\left(\frac{U_x}{\mu_z} + b_{zz}L_{\nabla f_3}W_{zz}\right)\theta + L_{\nabla f_3}L_{f_2}L_{\nabla^2 f_3}W_{zz}\left(\frac{1}{\mu_z} + b_{zz}\right)\theta$$

$$\quad\quad = U_{xy}\theta,$$

where

$$U_{xy} := L_{\nabla^2 f_3}L_{f_2}\left(W_{zz} + b_{zz}U_x + L_{\nabla f_3}W_{zz}b_{zz} + 2\frac{W_{zz}L_{\nabla f_3}}{\mu_z}\right)$$

$$\quad + L_{\nabla f_3}\left(L_{f_2}L_{\nabla^2 f_3}\left(\frac{L_{\nabla f_3}}{\mu_z^2}W_{zz} + b_{zz}\left(\frac{U_x}{\mu_z} + b_{zz}L_{\nabla f_3}W_{zz}\right)\right) + L_{\nabla f_2}\left(\frac{U_x}{\mu_z} + b_{zz}L_{\nabla f_3}W_{zz} + W_{zz}\right)\right). \tag{D.18}$$

The proof of the biasedness errors for the estimator $\nabla_{xy}^2\bar f(x,y,z;\xi)$ in equation equation D.4 can be established following nearly identical arguments. $\quad\square$

**Lemma D.2 (Bounds on variance of $\nabla^2\bar f$)** *Under Assumptions 3.1, 3.4, and 3.5, the variances of the stochastic matrices $\nabla_{xy}^2\bar f^\xi$ and $\nabla_{yy}^2\bar f^\xi$ are bounded, i.e., there exist positive constants $V_{xy}$ and $V_{yy}$ such that*

$$\mathbb{E}[\|\nabla_{xy}^2\bar f(x,y,z;\xi) - \mathbb{E}\left[\nabla_{xy}^2\bar f(x,y,z;\xi)|\mathcal F_\xi\right]\|^2|\mathcal F_\xi] \leq V_{xy},$$

$$\mathbb{E}[\|\nabla_{yy}^2\bar f(x,y,z;\xi) - \mathbb{E}[\nabla_{yy}^2\bar f(x,y,z;\xi)|\mathcal F_\xi]\|^2|\mathcal F_\xi] \leq V_{yy}.$$

**Proof.** For this proof, we will omit the point $(x,y,z)$ that the terms are evaluated at; we will simply use a $\xi$-superscript as short-hand to indicate any random terms. We can obtain the bound on the variance of $\nabla_{xy}^2\bar f^\xi$ by first referencing equation A.1 and applying the fact that $\|\sum_{i=1}^N a_i\|^2 \leq$

$N \sum_{i=1}^{N} \|a_i\|^2$ (for some $a \in \mathbb{R}^N$) to the two initial difference terms as well as all of the resulting terms (leading to a total of 12 terms), Assumption 3.4, and the consistency of matrix norms, to obtain

$$\mathbb{E}[\|\nabla_{xy}^2 \bar{f}^\xi - \mathbb{E}[\nabla_{xy}^2 \bar{f}^\xi]\|^2]$$

$$\leq 12\mathbb{E}[\|\nabla_{yzx}^3 f_3^\xi\|^2]\mathbb{E}[\|[\nabla_{zz}^2 f_3^\xi]^{-1}\|^2]\mathbb{E}[\|\nabla_z f_2^\xi\|^2] \; + \; 12\|\nabla_{yzx}^3 f_3\|^2\|\mathbb{E}[[\nabla_{zz}^2 f_3^\xi]^{-1}]\|^2\|\nabla_z f_2\|^2$$

$$+ 12\mathbb{E}[\|\nabla_{yzz}^3 f_3^\xi\|^2]\mathbb{E}[\|\nabla_x z^{\xi\top}\|^2]\mathbb{E}[\|[\nabla_{zz}^2 f_3^\xi]^{-1}\|^2]\mathbb{E}[\|\nabla_z f_2^\xi\|^2]$$

$$+ 12\|\nabla_{yzz}^3 f_3\|^2\|\mathbb{E}[\nabla_x z^{\xi\top}]\|^2\|\mathbb{E}[[\nabla_{zz}^2 f_3^\xi]^{-1}]\|^2\|\nabla_z f_2\|^2$$

$$+ 12\mathbb{E}[\|\nabla_{yz}^2 f_3^\xi\|^2]\mathbb{E}[\|[\nabla_{zz}^2 f_3^\xi]^{-1}\|^2]\mathbb{E}[\|\nabla_{zzx}^3 f_3^\xi\|^2]\mathbb{E}[\|[\nabla_{zz}^2 f_3^\xi]^{-1}\|^2]\mathbb{E}[\|\nabla_z f_2^\xi\|^2]$$

$$+ 12\|\nabla_{yz}^2 f_3\|^2\|\mathbb{E}[[\nabla_{zz}^2 f_3^\xi]^{-1}]\|^2\|\nabla_{zzx}^3 f_3\|^2\|\mathbb{E}[[\nabla_{zz}^2 f_3^\xi]^{-1}]\|^2\|\nabla_z f_2\|^2$$

$$+ 12\mathbb{E}[\|\nabla_{yz}^2 f_3^\xi\|^2]\mathbb{E}[\|[\nabla_{zz}^2 f_3^\xi]^{-1}\|^2]\mathbb{E}[\|\nabla_{zzz}^3 f_3^\xi\|^2]\mathbb{E}[\|\nabla_x z^{\xi\top}\|^2]\mathbb{E}[\|[\nabla_{zz}^2 f_3^\xi]^{-1}\|^2]\mathbb{E}[\|\nabla_z f_2^\xi\|^2]$$

$$+ 12\|\nabla_{yz}^2 f_3\|^2\|\mathbb{E}[[\nabla_{zz}^2 f_3^\xi]^{-1}]\|^2\|\nabla_{zzz}^3 f_3\|^2\|\mathbb{E}[\nabla_x z^{\xi\top}]\|^2\|\mathbb{E}[[\nabla_{zz}^2 f_3^\xi]^{-1}]\|^2\|\nabla_z f_2\|^2$$

$$+ 12\mathbb{E}[\|\nabla_{yz}^2 f_3^\xi\|^2]\mathbb{E}[\|[\nabla_{zz}^2 f_3^\xi]^{-1}\|^2]\mathbb{E}[\|\nabla_{zx}^2 f_2^\xi\|^2] \; + \; 12\|\nabla_{yz}^2 f_3\|^2\|\mathbb{E}[[\nabla_{zz}^2 f_3^\xi]^{-1}]\|^2\|\nabla_{zx}^2 f_2\|^2$$

$$+ 12\mathbb{E}[\|\nabla_{yz}^2 f_3^\xi\|^2]\mathbb{E}[\|[\nabla_{zz}^2 f_3^\xi]^{-1}\|^2]\mathbb{E}[\|\nabla_{zz}^2 f_2^\xi\|^2]\mathbb{E}[\|\nabla_x z^{\xi\top}\|^2]$$

$$+ 12\|\nabla_{yz}^2 f_3\|^2\|\mathbb{E}[[\nabla_{zz}^2 f_3^\xi]^{-1}]\|^2\|\nabla_{zz}^2 f_2\|^2\|\mathbb{E}[\nabla_x z^{\xi\top}]\|^2.$$

Now, using the result that $\|\mathbb{E}[\nabla_x z^{\xi\top}]\|^2 \leq \|\mathbb{E}[[\nabla_{zz}^2 f_3^\xi]^{-1}]\|^2\|\nabla_{zx}^2 f_3\|^2 \leq b_{zz}^2 L_{\nabla f_3}^2$ (from Assumptions 3.1, 3.4, and 3.5 along with the consistency of matrix norms), the result that $\mathbb{E}[\|\nabla_x z^{\xi\top}\|^2] \leq \mathbb{E}[\|[\nabla_{zz}^2 f_3^\xi]^{-1}\|^2]\mathbb{E}[\|\nabla_{zx}^2 f_3\xi\|^2] \leq b_{zz}^2 \mathbb{E}[\|\nabla_{zx}^2 f_3\xi\|^2]$ (from Assumptions 3.4 and 3.5 along with the consistency of matrix norms), and applying Assumptions 3.1 and 3.5, we have

$$\mathbb{E}[\|\nabla_{xy}^2 \bar{f}^\xi - \mathbb{E}[\nabla_{xy}^2 \bar{f}^\xi]\|^2]$$

$$\leq 12\mathbb{E}[\|\nabla_{yzx}^3 f_3^\xi\|^2]b_{zz}^2\mathbb{E}[\|\nabla_z f_2^\xi\|^2] + 12L_{\nabla^2 f_3}^2 b_{zz}^2 L_{f_2}^2 + 12\mathbb{E}[\|\nabla_{yzz}^3 f_3^\xi\|^2]b_{zz}^2\mathbb{E}[\|\nabla_{zx}^2 f_3^\xi\|^2]b_{zz}^2\mathbb{E}[\|\nabla_z f_2^\xi\|^2]$$

$$+ 12L_{\nabla^2 f_3}^2 b_{zz}^2 L_{\nabla f_3}^2 b_{zz}^2 L_{f_2}^2 + 12\mathbb{E}[\|\nabla_{yz}^2 f_3^\xi\|^2]b_{zz}^2\mathbb{E}[\|\nabla_{zzx}^3 f_3^\xi\|^2]b_{zz}^2\mathbb{E}[\|\nabla_z f_2^\xi\|^2]$$

$$+ 12L_{\nabla f_3}^2 b_{zz}^2 L_{\nabla^2 f_3}^2 b_{zz}^2 L_{f_2}^2 + 12\mathbb{E}[\|\nabla_{yz}^2 f_3^\xi\|^2]b_{zz}^2\mathbb{E}[\|\nabla_{zzz}^3 f_3^\xi\|^2]b_{zz}^2\mathbb{E}[\|\nabla_{zx}^2 f_3^\xi\|^2]b_{zz}^2\mathbb{E}[\|\nabla_z f_2^\xi\|^2]$$

$$+ 12L_{\nabla f_3}^2 b_{zz}^2 L_{\nabla^2 f_3}^2 b_{zz}^2 L_{\nabla f_3}^2 b_{zz}^2 L_{f_2}^2 + 12\mathbb{E}[\|\nabla_{yz}^2 f_3^\xi\|^2]b_{zz}^2\mathbb{E}[\|\nabla_{zx}^2 f_2^\xi\|^2] + 12L_{\nabla f_3}^2 b_{zz}^2 L_{\nabla f_2}^2$$

$$+ 12\mathbb{E}[\|\nabla_{yz}^2 f_3^\xi\|^2]b_{zz}^2\mathbb{E}[\|\nabla_{zz}^2 f_2^\xi\|^2]b_{zz}^2\mathbb{E}[\|\nabla_{zx}^2 f_3^\xi\|^2] + 12L_{\nabla f_3}^2 b_{zz}^2 L_{\nabla f_2}^2 b_{zz}^2 L_{\nabla f_3}^2.$$

Finally, with all of the remaining expectation terms, we can apply the definition of variance (i.e., $\mathbb{E}[X^2] = \text{Var}[X] + \mathbb{E}[X]^2$) followed by Assumption 3.4 to upper-bound the variance term along with Assumptions 3.1 and 3.4 to upper-bound the $\mathbb{E}[X]^2$ term. These bounds are given as:

$$\mathbb{E}[\|\nabla_{yzx}^3 f_3^\xi\|^2] \leq \sigma_{\nabla^3 f_3}^2 + L_{\nabla^2 f_3}^2, \quad \mathbb{E}[\|\nabla_{yzz}^3 f_3^\xi\|^2] \leq \sigma_{\nabla^3 f_3}^2 + L_{\nabla^2 f_3}^2,$$

$$\mathbb{E}[\|\nabla_{zzx}^3 f_3^\xi\|^2] \leq \sigma_{\nabla^3 f_3}^2 + L_{\nabla^2 f_3}^2, \quad \mathbb{E}[\|\nabla_{zzz}^3 f_3^\xi\|^2] \leq \sigma_{\nabla^3 f_3}^2 + L_{\nabla^2 f_3}^2,$$

$$\mathbb{E}[\|\nabla_{zx}^2 f_3^\xi\|^2] \; \leq \sigma_{\nabla^2 f_3}^2 + L_{\nabla f_3}^2, \quad \mathbb{E}[\|\nabla_{yz}^2 f_3^\xi\|^2] \; \leq \sigma_{\nabla^2 f_3}^2 + L_{\nabla f_3}^2,$$

$$\mathbb{E}[\|\nabla_{zz}^2 f_3^\xi\|^2] \; \leq \sigma_{\nabla^2 f_3}^2 + L_{\nabla f_3}^2, \quad \mathbb{E}[\|\nabla_z f_2^\xi\|^2] \; \leq \sigma_{\nabla f_2}^2 + L_{f_2}^2.$$

Applying these bounds, we will obtain

$$\mathbb{E}[\|\nabla_{xy}^2 \bar{f}^\xi - \mathbb{E}[\nabla_{xy}^2 \bar{f}^\xi]\|^2]$$

$$\leq 12(\sigma_{\nabla^3 f_3}^2 + L_{\nabla^2 f_3}^2)b_{zz}^2(\sigma_{\nabla f_2}^2 + L_{f_2}^2) + 12L_{\nabla^2 f_3}^2 b_{zz}^2 L_{f_2}^2$$

$$+ 12(\sigma_{\nabla^3 f_3}^2 + L_{\nabla^2 f_3}^2)b_{zz}^2(\sigma_{\nabla^2 f_3}^2 + L_{\nabla f_3}^2)b_{zz}^2(\sigma_{\nabla f_2}^2 + L_{f_2}^2)$$

$$+ 12L_{\nabla^2 f_3}^2 b_{zz}^2 L_{\nabla f_3}^2 b_{zz}^2 L_{f_2}^2 + 12(\sigma_{\nabla^2 f_3}^2 + L_{\nabla f_3}^2)b_{zz}^2(\sigma_{\nabla^3 f_3}^2 + L_{\nabla^2 f_3}^2)b_{zz}^2(\sigma_{\nabla f_2}^2 + L_{f_2}^2)$$

$$+ 12L_{\nabla f_3}^2 b_{zz}^2 L_{\nabla^2 f_3}^2 b_{zz}^2 L_{f_2}^2 + 12(\sigma_{\nabla^2 f_3}^2 + L_{\nabla f_3}^2)b_{zz}^2(\sigma_{\nabla^3 f_3}^2 + L_{\nabla^2 f_3}^2)b_{zz}^2(\sigma_{\nabla^2 f_3}^2 + L_{\nabla f_3}^2)b_{zz}^2(\sigma_{\nabla f_2}^2 + L_{f_2}^2)$$

$$+ 12L_{\nabla f_3}^2 b_{zz}^2 L_{\nabla^2 f_3}^2 b_{zz}^2 L_{\nabla f_3}^2 b_{zz}^2 L_{f_2}^2 + 12(\sigma_{\nabla^2 f_3}^2 + L_{\nabla f_3}^2)b_{zz}^2(\sigma_{\nabla^2 f_3}^2 + L_{\nabla f_3}^2) + 12L_{\nabla f_3}^2 b_{zz}^2 L_{\nabla f_2}^2$$

$$+ 12(\sigma_{\nabla^2 f_3}^2 + L_{\nabla f_3}^2)b_{zz}^2(\sigma_{\nabla^2 f_3}^2 + L_{\nabla f_3}^2)b_{zz}^2(\sigma_{\nabla^2 f_3}^2 + L_{\nabla f_3}^2) + 12L_{\nabla f_3}^2 b_{zz}^2 L_{\nabla f_2}^2 b_{zz}^2 L_{\nabla f_3}^2$$

$$:= V_{xy}. \tag{D.19}$$

This completes the proof for the variance bound on $\nabla^2_{xy}\bar{f}^\xi$.

The proof of the variance bound on $\nabla^2_{yy}\bar{f}^\xi$ follows nearly identical arguments. $\qquad\square$

**Lemma D.3 (Bounds on bias and variance of UL direction)** *Recalling the definition of $\tilde{g}^i_{f_1}$ in equation equation 2.5, define $\bar{g}^i_{f_1} = \mathbb{E}[\tilde{g}^i_{f_1}|\mathcal{F}_i]$. Then, under Assumptions 3.1, 3.2, 3.4, 3.5, and 3.6, there exist positive constants $\omega$ and $\tau$ such that*

$$\|\nabla f(x^i, y^{i+1}, z^{i+1}) - \bar{g}^i_{f_1}\| \;\leq\; \omega\theta_i \qquad\text{and}\qquad \mathbb{E}[\|\tilde{g}^i_{f_1} - \bar{g}^i_{f_1}\|^2|\mathcal{F}_i] \;\leq\; \tau.$$

**Proof.** For this proof, we will omit the point $(x^i, y^{i+1}, z^{i+1})$ that the terms are evaluated at; we will simply use a $\xi^i$-superscript as short-hand to indicate any random terms. Similarly, we will use the notation $(\cdot)^{\xi^i}$ to denote that every term in the parenthesis is a random variable. To prove the upper-bound on the biasedness of $\tilde{g}^i_{f_1}$, we can begin by referring to equation 2.2 and applying Assumption 3.4, yielding

$$\bar{g}^i_{f_1} = \mathbb{E}[\tilde{g}^i_{f_1}] = \mathbb{E}[\nabla f(x^i, y^{i+1}, z^{i+1}; \xi^i)]$$
$$= \nabla_x f_1 - \nabla^2_{xz} f_3 \mathbb{E}[[\nabla^2_{zz} f_3^{\xi^i}]^{-1}]\nabla_z f_1 - \mathbb{E}[\nabla^2_{xy}\bar{f}^{\xi^i}]\mathbb{E}[[\nabla^2_{yy}\bar{f}^{\xi^i}]^{-1}]\nabla_y f_1$$
$$+ \mathbb{E}[\nabla^2_{xy}\bar{f}^{\xi^i}]\mathbb{E}[[\nabla^2_{yy}\bar{f}^{\xi^i}]^{-1}]\nabla^2_{yz} f_3 \mathbb{E}[[\nabla^2_{zz} f_3^{\xi^i}]^{-1}]\nabla_z f_1.$$

Now, to derive a bound on the biasedness $\|\nabla f(x^i, y^{i+1}, z^{i+1}) - \bar{g}^i_{f_1}\|$, we can begin by utilizing the triangle inequality, the consistency of matrix norms, and Assumption 3.1 and 3.6, yielding

$$\|\nabla f(x^i, y^{i+1}, z^{i+1}) - \bar{g}^i_{f_1}\|$$
$$\leq L_{\nabla f_3} L_{f_1} W_{zz}\theta_i + \underbrace{L_{f_1}\|\mathbb{E}[\nabla^2_{xy}\bar{f}^{\xi^i}]\mathbb{E}[[\nabla^2_{yy}\bar{f}^{\xi^i}]^{-1}] - \nabla^2_{xy}\bar{f}[\nabla^2_{yy}\bar{f}]^{-1}\|}_{T_1^{(1)}}$$
$$+ \underbrace{L_{f_1}\|\nabla^2_{xy}\bar{f}[\nabla^2_{yy}\bar{f}]^{-1}\nabla^2_{yz} f_3[\nabla^2_{zz} f_3]^{-1} - \mathbb{E}[\nabla^2_{xy}\bar{f}^{\xi^i}]\mathbb{E}[[\nabla^2_{yy}\bar{f}^{\xi^i}]^{-1}]\nabla^2_{yz} f_3\mathbb{E}[[\nabla^2_{zz} f_3^{\xi^i}]^{-1}]\|}_{T_2^{(1)}},$$

$$(D.20)$$

(**Analysis of $T_1^{(1)}$**): Now, to upper-bound $T_1^{(1)}$ in equation D.20, we begin by adding and subtracting the term $\nabla^2_{xy}\bar{f}\mathbb{E}[[\nabla^2_{yy}\bar{f}^{\xi^i}]^{-1}]$, applying the triangle inequality, and utilizing the consistency of matrix norms to obtain

$$L_{f_1} T_1^{(1)} \leq L_{f_1}\|\mathbb{E}[[\nabla^2_{yy}\bar{f}^{\xi^i}]^{-1}]\|\|\mathbb{E}[\nabla^2_{xy}\bar{f}^{\xi^i}] - \nabla^2_{xy}\bar{f}\| + L_{f_1}\|\nabla^2_{xy}\bar{f}\|\|\mathbb{E}[[\nabla^2_{yy}\bar{f}^{\xi^i}]^{-1}] - [\nabla^2_{yy}\bar{f}]^{-1}\|$$
$$\leq (b_{yy}U_{xy} + T_{xy}W_{yy})L_{f_1}\theta_i, \tag{D.21}$$

where the last inequality follows by applying Assumptions 3.1, 3.5, and 3.6 along with Lemma D.1, and where $\|\nabla^2_{xy}\bar{f}\| \leq T_{xy}$, which follows from the following reasoning (applying the triangle inequality, the consistency of matrix norms, along with Assumptions 3.1 and 3.2, and equation equation E.14):

$$\|\nabla^2_{xy}\bar{f}\|$$
$$\leq \|\nabla^2_{yx} f_2\| + \|\nabla^2_{yz} f_2\nabla_x z^\top\| + \|\nabla^3_{yzx} f_3\nabla^2_{zz} f_3^{-1}\nabla_z f_2\| + \|\nabla^3_{yzz} f_3\nabla_x z^\top\nabla^2_{zz} f_3^{-1}\nabla_z f_2\|$$
$$+ \|\nabla^2_{yz} f_3[\nabla^2_{zz} f_3]^{-1}\nabla^3_{zzx} f_3[\nabla^2_{zz} f_3]^{-1}\nabla_z f_2\| + \|\nabla^2_{yz} f_3[\nabla^2_{zz} f_3]^{-1}\nabla^3_{zzz} f_3\nabla_x z^\top[\nabla^2_{zz} f_3]^{-1}\nabla_z f_2\|$$
$$+ \|\nabla^2_{yz} f_3[\nabla^2_{zz} f_3]^{-1}\nabla^2_{zx} f_2\| + \|\nabla^2_{yz} f_3[\nabla^2_{zz} f_3]^{-1}\nabla^2_{zz} f_2\nabla_x z^\top\|$$
$$\leq \left(L_{\nabla f_2} + \frac{L_{\nabla^2 f_3} L_{f_2}}{\mu_z}\right)\left(1 + \frac{2L_{\nabla f_3}}{\mu_z} + \frac{L^2_{\nabla f_3}}{\mu_z^2}\right) := T_{xy}. \tag{D.22}$$

(**Analysis of $T_2^{(1)}$**): Now, to upper-bound $T_2^{(1)}$ in equation D.20, we begin by adding and subtracting the term $\nabla^2_{xy}\bar{f}[\nabla^2_{yy}\bar{f}]^{-1}\nabla^2_{yz} f_3\mathbb{E}[[\nabla^2_{zz} f_3^{\xi^i}]^{-1}]$, applying the triangle inequality, along with the

consistency of matrix norms to obtain

$$
L_{f_1} T_2^{(1)} \leq L_{f_1} \|\nabla_{xy}^2 \bar{f}\| \|[\nabla_{yy}^2 \bar{f}]^{-1}\| \|\nabla_{yz}^2 f_3\| \|[\nabla_{zz}^2 f_3]^{-1} - \mathbb{E}[[\nabla_{zz}^2 f_3^{\xi^i}]^{-1}]\|
$$
$$
+ L_{f_1} \|\nabla_{yz}^2 f_3\| \|\mathbb{E}[[\nabla_{zz}^2 f_3^{\xi^i}]^{-1}]\| \|\nabla_{xy}^2 \bar{f}[\nabla_{yy}^2 \bar{f}]^{-1} - \mathbb{E}[\nabla_{xy}^2 \bar{f}^{\xi^i}]\mathbb{E}[[\nabla_{yy}^2 \bar{f}^{\xi^i}]^{-1}]\|
$$
$$
\leq L_{f_1} L_{\nabla f_3} \left( \frac{T_{xy} W_{zz}}{\mu_y} + b_{zz} \left( b_{yy} U_{xy} + T_{xy} W_{yy} \right) \right) \theta_i, \tag{D.23}
$$

where the last inequality follows from applying Assumptions 3.1, 3.3, 3.5, and 3.6, the bound $\|\nabla_{xy}^2 \bar{f}\| \leq T_{xy}$ we derived in equation D.22, and the bound we derived on the term $T_1^{(1)}$ in equation D.21.

Finally, substituting the bounds equation D.21 and equation D.23 on the terms $T_1^{(1)}$ and $T_2^{(1)}$, respectively, back into equation D.20, we obtain the desired bound on the biasedness as

$$
\|\nabla f(x^i, y^{i+1}, z^{i+1}) - \bar{g}_{f_1}^i\|
$$
$$
\leq L_{f_1} \left( L_{\nabla f_3} W_{zz} + b_{yy} U_{xy} + T_{xy} W_{yy} + L_{\nabla f_3} \left( \frac{T_{xy} W_{zz}}{\mu_y} + b_{zz} \left( b_{yy} U_{xy} + T_{xy} W_{yy} \right) \right) \right) \theta_i := \omega \theta_i. \tag{D.24}
$$

Now, to bound the variance of $\tilde{g}_{f_1}^i$, we can begin by using the fact that $\|a + b + c + d\|^2 \leq 4 \left( \|a\|^2 + \|b\|^2 + \|c\|^2 + \|d\|^2 \right)$, with $a$, $b$, $c$, and $d$ real-valued vectors, to obtain (it bears mentioning that for ease of notation, we will use $(\cdot)^{\xi^i}$ to denote that all terms in the parenthesis are random variables)

$$
\mathbb{E}[\|\tilde{g}_{f_1}^i - \bar{g}_{f_1}^i\|^2] = \mathbb{E}[\|\tilde{g}_{f_1}^i - \mathbb{E}[\tilde{g}_{f_1}^i | \mathcal{F}_i]\|^2]
$$
$$
\leq \underbrace{4\mathbb{E}[\|\nabla_x f_1^{\xi^i} - \mathbb{E}[\nabla_x f_1^{\xi^i}]\|^2]}_{T_1^{(2)}} + \underbrace{4\mathbb{E}[\|\mathbb{E}[(\nabla_{xz}^2 f_3[\nabla_{zz}^2 f_3]^{-1}\nabla_z f_1)^{\xi^i}] - (\nabla_{xz}^2 f_3[\nabla_{zz}^2 f_3]^{-1}\nabla_z f_1)^{\xi^i}\|^2]}_{T_2^{(2)}}
$$
$$
+ \underbrace{4\mathbb{E}[\|\mathbb{E}[(\nabla_{xy}^2 \bar{f}[\nabla_{yy}^2 \bar{f}]^{-1}\nabla_y f_1)^{\xi^i}] - (\nabla_{xy}^2 \bar{f}[\nabla_{yy}^2 \bar{f}]^{-1}\nabla_y f_1)^{\xi^i}\|^2]}_{T_3^{(2)}}
$$
$$
+ \underbrace{4\mathbb{E}[\|(\nabla_{xy}^2 \bar{f}[\nabla_{yy}^2 \bar{f}]^{-1}\nabla_{yz}^2 f_3[\nabla_{zz}^2 f_3]^{-1}\nabla_z f_1)^{\xi^i} - \mathbb{E}[(\nabla_{xy}^2 \bar{f}[\nabla_{yy}^2 \bar{f}]^{-1}\nabla_{yz}^2 f_3[\nabla_{zz}^2 f_3]^{-1}\nabla_z f_1)^{\xi^i}]\|^2]}_{T_4^{(2)}}.
$$
$$\tag{D.25}$$

(**Analysis of $T_1^{(2)}$**): Notice that the term $T_1^{(2)}$ in equation D.25 can be bounded by Assumption 3.4

$$
4\mathbb{E}[\|\nabla_x f_1^{\xi^i} - \mathbb{E}[\nabla_x f_1^{\xi^i}]\|^2] \leq 4\sigma_{\nabla f_1}^2 := \tau_1. \tag{D.26}
$$

(**Analysis of $T_2^{(2)}$**): Now dealing with the contents of the term $T_2^{(2)}$, we can apply Assumption 3.4 and re-factorize to obtain

$$
\mathbb{E}[(\nabla_{xz}^2 f_3[\nabla_{zz}^2 f_3]^{-1}\nabla_z f_1)^{\xi^i}] - (\nabla_{xz}^2 f_3[\nabla_{zz}^2 f_3]^{-1}\nabla_z f_1)^{\xi^i}
$$
$$
= \nabla_{xz}^2 f_3 \mathbb{E}[[\nabla_{zz}^2 f_3^{\xi^i}]^{-1}]\nabla_z f_1 - \nabla_{xz}^2 f_3^{\xi^i}[\nabla_{zz}^2 f_3^{\xi^i}]^{-1}\nabla_z f_1^{\xi^i}
$$
$$
= (\nabla_{xz}^2 f_3 - \nabla_{xz}^2 f_3^{\xi^i})\mathbb{E}[[\nabla_{zz}^2 f_3^{\xi^i}]^{-1}]\nabla_z f_1
$$
$$
+ \nabla_{xz}^2 f_3^{\xi^i}(\mathbb{E}[[\nabla_{zz}^2 f_3^{\xi^i}]^{-1}] - [\nabla_{zz}^2 f_3^{\xi^i}]^{-1})\nabla_z f_1
$$
$$
+ \nabla_{xz}^2 f_3^{\xi^i}[\nabla_{zz}^2 f_3^{\xi^i}]^{-1}(\nabla_z f_1 - \nabla_z f_1^{\xi^i}).
$$

By using this, the fact that $\|a + b + c\|^2 \leq 3 \left( \|a\|^2 + \|b\|^2 + \|c\|^2 \right)$, with $a$, $b$, and $c$ real-valued vectors, along with the consistency of matrix norms, and Assumptions 3.1, 3.5, and 3.4, we can see

that the term $T_2^{(2)}$ is upper-bounded by

$$4\mathbb{E}[\|\mathbb{E}[(\nabla_{xz}^2 f_3[\nabla_{zz}^2 f_3]^{-1}\nabla_z f_1)^{\xi^i}] - (\nabla_{xz}^2 f_3[\nabla_{zz}^2 f_3]^{-1}\nabla_z f_1)^{\xi^i}\|^2]$$

$$\leq 12\sigma_{\nabla^2 f_3}^2 b_{zz}^2 L_{f_1}^2 + 12\mathbb{E}[\|\nabla_{xz}^2 f_3^{\xi^i}\|^2]\mathbb{E}[\|\mathbb{E}[[\nabla_{zz}^2 f_3^{\xi^i}]^{-1}] - [\nabla_{zz}^2 f_3^{\xi^i}]^{-1}\|^2]L_{f_1}^2$$

$$+ 12\mathbb{E}[\|\nabla_{xz}^2 f_3^{\xi^i}\|^2]b_{zz}^2\sigma_{\nabla f_1}^2. \tag{D.27}$$

Consider the term $\mathbb{E}[\|\nabla_{xz}^2 f_3^{\xi^i}\|^2]$ in equation D.27. Using the definition of variance (i.e., $\mathbb{E}[X^2] = \text{Var}[X] + \mathbb{E}[X]^2$) along with Assumptions 3.4 and 3.1, we have

$$\mathbb{E}[\|\nabla_{xz}^2 f_3^{\xi^i}\|^2] = \mathbb{E}[\|\nabla_{xz}^2 f_3^{\xi^i} - \mathbb{E}[\nabla_{xz}^2 f_3^{\xi^i}]\|^2] + \mathbb{E}[\|\mathbb{E}[\nabla_{xz}^2 f_3^{\xi^i}]\|^2] \leq \sigma_{\nabla^2 f_3}^2 + L_{\nabla f_3}^2. \tag{D.28}$$

Consider the term $\mathbb{E}[\|[\nabla_{zz}^2 f_3^{\xi^i}]^{-1} - \mathbb{E}[[\nabla_{zz}^2 f_3^{\xi^i}]^{-1}]\|^2]$ in equation D.27. Using the fact that $\|a + b\|^2 \leq 2\left(\|a\|^2 + \|b\|^2\right)$, with $a$ and $b$ real-valued vectors, and applying Assumption 3.5, we have

$$\mathbb{E}[\|[\nabla_{zz}^2 f_3^{\xi^i}]^{-1} - \mathbb{E}[[\nabla_{zz}^2 f_3^{\xi^i}]^{-1}]\|^2] \leq 2\mathbb{E}[\|[\nabla_{zz}^2 f_3^{\xi^i}]^{-1}\|^2] + 2\mathbb{E}[\|\mathbb{E}[[\nabla_{zz}^2 f_3^{\xi^i}]^{-1}]\|^2] \leq 4b_{zz}^2. \tag{D.29}$$

Now substituting the bounds equation D.28 and equation D.29 back into equation D.27, we obtain the bound on the term $T_2^{(2)}$ as

$$4\mathbb{E}[\|\mathbb{E}[(\nabla_{xz}^2 f_3[\nabla_{zz}^2 f_3]^{-1}\nabla_z f_1)^{\xi^i}] - (\nabla_{xz}^2 f_3[\nabla_{zz}^2 f_3]^{-1}\nabla_z f_1)^{\xi^i}\|^2]$$

$$\leq 12\sigma_{\nabla^2 f_3}^2 b_{zz}^2 L_{f_1}^2 + 48(\sigma_{\nabla^2 f_3}^2 + L_{\nabla f_3}^2)b_{zz}^2 L_{f_1}^2 + 12(\sigma_{\nabla^2 f_3}^2 + L_{\nabla f_3}^2)b_{zz}^2\sigma_{\nabla f_1}^2 := \tau_2. \tag{D.30}$$

(**Analysis of $T_3^{(2)}$**): Applying similar reasoning that was used in bounding the term $T_2^{(2)}$, along with utilizing Lemma D.2 and Assumptions 3.1, 3.5, and 3.4, we have

$$4\mathbb{E}[\|\mathbb{E}[(\nabla_{xy}^2 \bar{f}[\nabla_{yy}^2 \bar{f}]^{-1}\nabla_y f_1)^{\xi^i}] - (\nabla_{xy}^2 \bar{f}[\nabla_{yy}^2 \bar{f}]^{-1}\nabla_y f_1)^{\xi^i}\|^2]$$

$$\leq 12V_{xy}b_{yy}^2 L_{f_1}^2 + 12\mathbb{E}[\|\nabla_{xy}^2 \bar{f}^{\xi^i}\|^2]\mathbb{E}[\|\mathbb{E}[[\nabla_{yy}^2 \bar{f}^{\xi^i}]^{-1}] - [\nabla_{yy}^2 \bar{f}^{\xi^i}]^{-1}\|^2]L_{f_1}^2$$

$$+ 12\mathbb{E}[\|\nabla_{xy}^2 \bar{f}^{\xi^i}\|^2]b_{yy}^2\sigma_{\nabla f_1}^2. \tag{D.31}$$

Consider the term $\mathbb{E}[\|[\nabla_{yy}^2 \bar{f}^{\xi^i}]^{-1} - \mathbb{E}[[\nabla_{yy}^2 \bar{f}^{\xi^i}]^{-1}]\|^2]$ in equation D.31. Applying the same reasoning that was used to derive equation D.29, we have

$$\mathbb{E}[\|[\nabla_{yy}^2 \bar{f}^{\xi^i}]^{-1} - \mathbb{E}[[\nabla_{yy}^2 \bar{f}^{\xi^i}]^{-1}]\|^2] \leq 4b_{yy}^2. \tag{D.32}$$

Consider the term $\mathbb{E}[\|\nabla_{xy}^2 \bar{f}^{\xi^i}\|^2]$ in equation D.31. Using the definition of variance (i.e., $\mathbb{E}[X^2] = \text{Var}[X] + \mathbb{E}[X]^2$) and applying Lemma D.2, we have

$$\mathbb{E}[\|\nabla_{xy}^2 \bar{f}^{\xi^i}\|^2] \leq V_{xy} + \|\mathbb{E}[\nabla_{xy}^2 \bar{f}^{\xi^i}]\|^2. \tag{D.33}$$

Now, consider the $\|\mathbb{E}[\nabla_{xy}^2 \bar{f}^{\xi^i}]\|^2$ term in equation D.33. Noticing that $\mathbb{E}[\nabla_x z^{\xi^i\top}] = \mathbb{E}[[\nabla_{zz}^2 f_3^{\xi^i}]^{-1}]\nabla_{zx}^2 f_3$ (from Assumption 3.4), we can apply the triangle inequality along with the consistency of matrix norms and Assumptions 3.1, 3.4, and 3.5 to obtain

$$\|\mathbb{E}[\nabla_{xy}^2 \bar{f}^{\xi^i}]\| \leq L_{\nabla f_2} + b_{zz}(L_{\nabla f_2}^2 + L_{f_2}L_{\nabla^2 f_3}(1 + 2b_{zz}L_{\nabla f_3} + b_{zz}^2 L_{\nabla f_3}^2) + L_{\nabla f_3}L_{\nabla f_2}(1 + b_{zz}L_{\nabla f_3}))$$

$$:= \tilde{T}_{xy}. \tag{D.34}$$

Finally, squaring both sides of this inequality, we have

$$\|\mathbb{E}[\nabla_{xy}^2 \bar{f}^{\xi^i}]\|^2 \leq \tilde{T}_{xy}^2. \tag{D.35}$$

Thus, substituting equation D.35 back into equation D.33, we have $\mathbb{E}[\|\nabla_{xy}^2 \bar{f}^{\xi^i}\|^2] \leq V_{xy} + \tilde{T}_{xy}^2$. Finally, substituting this and bound equation D.32 back into equation D.31, we obtain the bound on the term $T_3^{(2)}$ as

$$4\mathbb{E}[\|\mathbb{E}[(\nabla_{xy}^2 \bar{f}[\nabla_{yy}^2 \bar{f}]^{-1}\nabla_y f_1)^{\xi^i}] - (\nabla_{xy}^2 \bar{f}[\nabla_{yy}^2 \bar{f}]^{-1}\nabla_y f_1)^{\xi^i}\|^2]$$

$$\leq 12V_{xy}b_{yy}^2 L_{f_1}^2 + 48(V_{xy} + \tilde{T}_{xy}^2)b_{yy}^2 L_{f_1}^2 + 12(V_{xy} + \tilde{T}_{xy}^2)b_{yy}^2\sigma_{\nabla f_1}^2 := \tau_3. \tag{D.36}$$

(**Analysis of $T_4^{(2)}$**): Now, dealing with the contents of the norm in term $T_4^{(2)}$, we can apply Assumption 3.4 and re-factorize to obtain

$$\left(\nabla_{xy}^2 \bar{f} [\nabla_{yy}^2 \bar{f}]^{-1} \nabla_{yz}^2 f_3 [\nabla_{zz}^2 f_3]^{-1} \nabla_z f_1\right)^{\xi^i} - \mathbb{E}[\left(\nabla_{xy}^2 \bar{f} [\nabla_{yy}^2 \bar{f}]^{-1} \nabla_{yz}^2 f_3 [\nabla_{zz}^2 f_3]^{-1} \nabla_z f_1\right)^{\xi^i}]$$

$$= (\nabla_{xy}^2 \bar{f}^{\xi^i} - \mathbb{E}[\nabla_{xy}^2 \bar{f}^{\xi^i}])[\nabla_{yy}^2 \bar{f}^{\xi^i}]^{-1} \nabla_{yz}^2 f_3^{\xi^i} [\nabla_{zz}^2 f_3^{\xi^i}]^{-1} \nabla_z f_1^{\xi^i}$$

$$+ \mathbb{E}[\nabla_{xy}^2 \bar{f}^{\xi^i}] \underbrace{\left([\nabla_{yy}^2 \bar{f}^{\xi^i}]^{-1} \nabla_{yz}^2 f_3^{\xi^i} [\nabla_{zz}^2 f_3^{\xi^i}]^{-1} - \mathbb{E}[[\nabla_{yy}^2 \bar{f}^{\xi^i}]^{-1}] \nabla_{yz}^2 f_3 \mathbb{E}[[\nabla_{zz}^2 f_3^{\xi^i}]^{-1}]\right)}_{\hat{T}_4^{(2)}} \nabla_z f_1^{\xi^i}$$

$$+ \mathbb{E}[\nabla_{xy}^2 \bar{f}^{\xi^i}] \mathbb{E}[[\nabla_{yy}^2 \bar{f}^{\xi^i}]^{-1}] \nabla_{yz}^2 f_3 \mathbb{E}[[\nabla_{zz}^2 f_3^{\xi^i}]^{-1}](\nabla_z f_1^{\xi^i} - \nabla_z f_1). \tag{D.37}$$

We can further re-factorize the term $\hat{T}_4^{(2)}$ in equation D.37 to obtain

$$\hat{T}_4^{(2)} = ([\nabla_{yy}^2 \bar{f}^{\xi^i}]^{-1} - \mathbb{E}[[\nabla_{yy}^2 \bar{f}^{\xi^i}]^{-1}]) \nabla_{yz}^2 f_3^{\xi^i} [\nabla_{zz}^2 f_3^{\xi^i}]^{-1} + \mathbb{E}[[\nabla_{yy}^2 \bar{f}^{\xi^i}]^{-1}](\nabla_{yz}^2 f_3^{\xi^i} - \nabla_{yz}^2 f_3)[\nabla_{zz}^2 f_3^{\xi^i}]^{-1}$$

$$+ \mathbb{E}[[\nabla_{yy}^2 \bar{f}^{\xi^i}]^{-1}] \nabla_{yz}^2 f_3 ([\nabla_{zz}^2 f_3^{\xi^i}]^{-1} - \mathbb{E}[[\nabla_{zz}^2 f_3^{\xi^i}]^{-1}]). \tag{D.38}$$

Substituting equation D.38 for $\hat{T}_4^{(2)}$ in equation D.37, we have

$$\left(\nabla_{xy}^2 \bar{f} [\nabla_{yy}^2 \bar{f}]^{-1} \nabla_{yz}^2 f_3 [\nabla_{zz}^2 f_3]^{-1} \nabla_z f_1\right)^{\xi^i} - \mathbb{E}[\left(\nabla_{xy}^2 \bar{f} [\nabla_{yy}^2 \bar{f}]^{-1} \nabla_{yz}^2 f_3 [\nabla_{zz}^2 f_3]^{-1} \nabla_z f_1\right)^{\xi^i}]$$

$$= (\nabla_{xy}^2 \bar{f}^{\xi^i} - \mathbb{E}[\nabla_{xy}^2 \bar{f}^{\xi^i}])[\nabla_{yy}^2 \bar{f}^{\xi^i}]^{-1} \nabla_{yz}^2 f_3^{\xi^i} [\nabla_{zz}^2 f_3^{\xi^i}]^{-1} \nabla_z f_1^{\xi^i}$$

$$+ \mathbb{E}[\nabla_{xy}^2 \bar{f}^{\xi^i}]([\nabla_{yy}^2 \bar{f}^{\xi^i}]^{-1} - \mathbb{E}[[\nabla_{yy}^2 \bar{f}^{\xi^i}]^{-1}]) \nabla_{yz}^2 f_3^{\xi^i} [\nabla_{zz}^2 f_3^{\xi^i}]^{-1} \nabla_z f_1^{\xi^i}$$

$$+ \mathbb{E}[\nabla_{xy}^2 \bar{f}^{\xi^i}] \mathbb{E}[[\nabla_{yy}^2 \bar{f}^{\xi^i}]^{-1}](\nabla_{yz}^2 f_3^{\xi^i} - \nabla_{yz}^2 f_3)[\nabla_{zz}^2 f_3^{\xi^i}]^{-1} \nabla_z f_1^{\xi^i}$$

$$+ \mathbb{E}[\nabla_{xy}^2 \bar{f}^{\xi^i}] \mathbb{E}[[\nabla_{yy}^2 \bar{f}^{\xi^i}]^{-1}] \nabla_{yz}^2 f_3 ([\nabla_{zz}^2 f_3^{\xi^i}]^{-1} - \mathbb{E}[[\nabla_{zz}^2 f_3^{\xi^i}]^{-1}]) \nabla_z f_1^{\xi^i}$$

$$+ \mathbb{E}[\nabla_{xy}^2 \bar{f}^{\xi^i}] \mathbb{E}[[\nabla_{yy}^2 \bar{f}^{\xi^i}]^{-1}] \nabla_{yz}^2 f_3 \mathbb{E}[[\nabla_{zz}^2 f_3^{\xi^i}]^{-1}](\nabla_z f_1^{\xi^i} - \nabla_z f_1). \tag{D.39}$$

Finally, substituting equation D.39 back into the norm for $T_4^{(2)}$ in equation D.25 and using the fact that $\|a+b+c+d+e\|^2 \le 5\left(\|a\|^2 + \|b\|^2 + \|c\|^2 + \|d\|^2 + \|e\|^2\right)$, with $a$, $b$, $c$, $d$, and $e$ real-valued vectors, along with the consistency of matrix norms, and applying Assumptions 3.1, 3.4, 3.5, along with Lemma D.2 and bounds equation D.29, equation D.32, and equation D.35, to obtain

$$T_4^{(2)} \le 20 b_{yy}^2 b_{zz}^2 V_{xy} \mathbb{E}[\|\nabla_{yz}^2 f_3^{\xi^i}\|^2] \mathbb{E}[\|\nabla_z f_1^{\xi^i}\|^2] + 80 b_{zz}^2 \tilde{T}_{xy}^2 b_{yy}^2 \mathbb{E}[\|\nabla_{yz}^2 f_3^{\xi^i}\|^2] \mathbb{E}[\|\nabla_z f_1^{\xi^i}\|^2]$$

$$+ 20 b_{yy}^2 \sigma_{\nabla^2 f_3}^2 b_{zz}^2 \tilde{T}_{xy}^2 \mathbb{E}[\|\nabla_z f_1^{\xi^i}\|^2] + 80 b_{yy}^2 L_{\nabla f_3}^2 b_{zz}^2 \tilde{T}_{xy}^2 \mathbb{E}[\|\nabla_z f_1^{\xi^i}\|^2] + 20 b_{yy}^2 L_{\nabla f_3}^2 b_{zz}^2 \sigma_{\nabla f_1}^2 \tilde{T}_{xy}^2. \tag{D.40}$$

Consider the terms $\mathbb{E}[\|\nabla_{yz}^2 f_3^{\xi^i}\|^2]$ and $\mathbb{E}[\|\nabla_z f_1^{\xi^i}\|^2]$ in equation D.40. Applying nearly identical reasoning that was used to derive equation D.28, we have

$$\mathbb{E}[\|\nabla_{yz}^2 f_3^{\xi^i}\|^2] \le \sigma_{\nabla^2 f_3}^2 + L_{\nabla f_3}^2, \tag{D.41}$$

$$\mathbb{E}[\|\nabla_z f_1^{\xi^i}\|^2] \le \sigma_{\nabla f_1}^2 + L_{f_1}^2. \tag{D.42}$$

Now substituting the bounds equation D.41 and equation D.42 back into equation D.40, we obtain the bound on the term $T_4^{(2)}$ as

$$4\mathbb{E}[\|(\nabla_{xy}^2 \bar{f} [\nabla_{yy}^2 \bar{f}]^{-1} \nabla_{yz}^2 f_3 [\nabla_{zz}^2 f_3]^{-1} \nabla_z f_1)^{\xi^i} - \mathbb{E}[(\nabla_{xy}^2 \bar{f} [\nabla_{yy}^2 \bar{f}]^{-1} \nabla_{yz}^2 f_3 [\nabla_{zz}^2 f_3]^{-1} \nabla_z f_1)^{\xi^i}]\|^2]$$

$$\le 20 b_{yy}^2 b_{zz}^2 V_{xy}(\sigma_{\nabla^2 f_3}^2 + L_{\nabla f_3}^2)(\sigma_{\nabla f_1}^2 + L_{f_1}^2) + 80 b_{zz}^2 \tilde{T}_{xy}^2 b_{yy}^2 (\sigma_{\nabla^2 f_3}^2 + L_{\nabla f_3}^2)(\sigma_{\nabla f_1}^2 + L_{f_1}^2)$$

$$+ 20 b_{yy}^2 \sigma_{\nabla^2 f_3}^2 b_{zz}^2 \tilde{T}_{xy}^2 (\sigma_{\nabla f_1}^2 + L_{f_1}^2) + 80 b_{yy}^2 L_{\nabla f_3}^2 \tilde{T}_{xy}^2 b_{zz}^2 (\sigma_{\nabla f_1}^2 + L_{f_1}^2) + 20 b_{yy}^2 L_{\nabla f_3}^2 b_{zz}^2 \sigma_{\nabla f_1}^2 \tilde{T}_{xy}^2$$

$$:= \tau_4. \tag{D.43}$$

The proof is completed by substituting the derived bounds for $T_1^{(2)}$, $T_2^{(2)}$, $T_3^{(2)}$, and $T_4^{(2)}$ (bounds equation D.26, equation D.30, equation D.36, and equation D.43, respectively) back

into equation D.25, yielding the desired variance bound (including the omitted $\sigma$-algebra $\mathcal{F}_i$ that the expectation is conditioned on):

$$\mathbb{E}[\|\tilde{g}^i_{f_1} - \bar{g}^i_{f_1}\|^2 | \mathcal{F}_i] \leq \tau, \quad \text{where} \quad \tau := \tau_1 + \tau_2 + \tau_3 + \tau_4. \tag{D.44}$$

$\square$

**Lemma D.4 (Boundedness of UL direction)** *Under Assumptions 3.1, 3.2, 3.4, 3.5, and 3.6, there exists a positive constant $\zeta$ such that*

$$\mathbb{E}[\|\tilde{g}^i_{f_1}\|^2 | \mathcal{F}_i] \leq \zeta.$$

**Proof.** For this proof, we may omit the point $(x^i, y^{i+1}, z^{i+1})$ that the terms are evaluated at; we will simply use a $\xi^i$-superscript as short-hand to indicate any random terms. From the definition of variance along with using Lemma D.3, we have

$$\mathbb{E}[\|\tilde{g}^i_{f_1}\|^2 | \mathcal{F}_i] = \|\bar{g}^i_{f_1}\|^2 + \mathbb{E}[\|\tilde{g}^i_{f_1} - \bar{g}^i_{f_1}\|^2 | \mathcal{F}_i] \leq \|\bar{g}^i_{f_1}\|^2 + \tau. \tag{D.45}$$

Now, considering the $\|\bar{g}^i_{f_1}\|$ term in equation D.45, we can apply the triangle inequality, Assumption 3.4, along with the consistency of matrix norms, to obtain

$$\|\bar{g}^i_{f_1}\| \leq \|\nabla_x f_1\| + \|\nabla^2_{xz} f_3 \mathbb{E}[[\nabla^2_{zz} f_3^{\xi^i}]^{-1} | \mathcal{F}_i] \nabla_z f_1\| + \|\mathbb{E}[\nabla^2_{xy} \bar{f}^{\xi^i} | \mathcal{F}_i] \mathbb{E}[[\nabla^2_{yy} \bar{f}^{\xi^i}]^{-1} | \mathcal{F}_i] \nabla_y f_1\|$$

$$+ \|\mathbb{E}[\nabla^2_{xy} \bar{f}^{\xi^i} | \mathcal{F}_i] \mathbb{E}[[\nabla^2_{yy} \bar{f}^{\xi^i}]^{-1} | \mathcal{F}_i] \nabla^2_{yz} f_3 \mathbb{E}[[\nabla^2_{zz} f_3^{\xi^i}]^{-1} | \mathcal{F}_i] \nabla_z f_1\|$$

$$\leq L_{f_1} + L_{\nabla f_3} L_{f_1} \|\mathbb{E}[[\nabla^2_{zz} f_3^{\xi^i}]^{-1} | \mathcal{F}_i]\| + L_{f_1} \|\mathbb{E}[\nabla^2_{xy} \bar{f}^{\xi^i} | \mathcal{F}_i]\| \|\mathbb{E}[[\nabla^2_{yy} \bar{f}^{\xi^i}]^{-1} | \mathcal{F}_i]\|$$

$$+ L_{\nabla f_3} L_{f_1} \|\mathbb{E}[\nabla^2_{xy} \bar{f}^{\xi^i} | \mathcal{F}_i]\| \|\mathbb{E}[[\nabla^2_{yy} \bar{f}^{\xi^i}]^{-1} | \mathcal{F}_i]\| \|\mathbb{E}[[\nabla^2_{zz} f_3^{\xi^i}]^{-1} | \mathcal{F}_i]\|$$

$$\leq L_{f_1} + L_{\nabla f_3} L_{f_1} b_{zz} + L_{f_1} \tilde{T}_{xy} b_{yy} + L_{\nabla f_3} L_{f_1} \tilde{T}_{xy} b_{yy} b_{zz},$$

where the second inequality follows from applying Assumption 3.1 and the last inequality follows from applying Assumption 3.5 along with the derived bound equation D.34 from Lemma D.3. Further, squaring both sides, we have the bound $\|\bar{g}^i_{f_1}\|^2 \leq (L_{f_1} + L_{\nabla f_3} L_{f_1} b_{zz} + L_{f_1} \tilde{T}_{xy} b_{yy} + L_{\nabla f_3} L_{f_1} \tilde{T}_{xy} b_{yy} b_{zz})^2$. Substituting this back into equation D.45, we obtain the bound

$$\mathbb{E}[\|\tilde{g}^i_{f_1}\|^2 | \mathcal{F}_i] \leq \zeta, \quad \text{where} \quad \zeta := (L_{f_1} + L_{\nabla f_3} L_{f_1} b_{zz} + L_{f_1} \tilde{T}_{xy} b_{yy} + L_{\nabla f_3} L_{f_1} \tilde{T}_{xy} b_{yy} b_{zz})^2 + \tau. \tag{D.46}$$

$\square$

**Lemma D.5 (Bounds on bias and variance of ML direction)** *Recalling the definition of $\tilde{g}^{i,j}_{f_2}$ in equation equation 2.4, define $\bar{g}^{i,j}_{f_2} = \mathbb{E}[\tilde{g}^{i,j}_{f_2} | \mathcal{F}_{i,j}]$. Then, under Assumptions 3.1, 3.4, 3.5, and 3.6, there exist positive constants $\hat{\omega}$ and $\hat{\tau}$ such that*

$$\|\nabla_y \bar{f}(x^i, y^{i,j}, z^{i,j+1}) - \bar{g}^{i,j}_{f_2}\| \leq \hat{\omega}\theta_i \quad \text{and} \quad \mathbb{E}[\|\tilde{g}^{i,j}_{f_2} - \bar{g}^{i,j}_{f_2}\|^2 | \mathcal{F}_{i,j}] \leq \hat{\tau}.$$

**Proof.** For this proof, we may omit the point $(x^i, y^{i,j}, z^{i,j+1})$ that the terms are evaluated at; we will simply use a $\xi^{i,j}$-superscript as short-hand to indicate any random terms. From Assumption 3.4, we have

$$\|\nabla_y \bar{f} - \bar{g}^{i,j}_{f_2}\| = \|\nabla_y f_2 - \nabla^2_{yz} f_3 [\nabla^2_{zz} f_3]^{-1} \nabla_z f_2 - (\nabla_y f_2 - \nabla^2_{yz} f_3 \mathbb{E}[[\nabla^2_{zz} f_3^{\xi^{i,j}}]^{-1} | \mathcal{F}_{i,j}] \nabla_z f_2)\|$$

$$= \|\nabla^2_{yz} f_3 (\mathbb{E}[[\nabla^2_{zz} f_3^{\xi^{i,j}}]^{-1} | \mathcal{F}_{i,j}] - [\nabla^2_{zz} f_3]^{-1}) \nabla_z f_2\|$$

$$\leq L_{\nabla f_3} L_{f_2} \|\mathbb{E}[[\nabla^2_{zz} f_3^{\xi^{i,j}}]^{-1} | \mathcal{F}_{i,j}] - [\nabla^2_{zz} f_3]^{-1}\|,$$

where the inequality follows from Assumption 3.1 along with the consistency of matrix norms. Utilizing Assumption 3.6, we obtain the desired first result of

$$\|\bar{g}^{i,j}_{f_2} - \nabla_y \bar{f}\| \leq \hat{\omega}\theta_i, \quad \text{where} \quad \hat{\omega} := L_{\nabla f_3} L_{f_2} W_{zz}. \tag{D.47}$$

Now, to estimate the variance of $\tilde{g}_{f_2}^{i,j}$, we can apply Assumption 3.4 and the fact that $\|a+b\|^2 \leq 2\|a\|^2 + 2\|b\|^2$, with $a$ and $b$ real-valued vectors, yielding

$$\mathbb{E}[\|\tilde{g}_{f_2}^{i,j} - \bar{g}_{f_2}^{i,j}\|^2|\mathcal{F}_{i,j}] = \mathbb{E}[\|\tilde{g}_{f_2}^{i,j} - \mathbb{E}[\tilde{g}_{f_2}^{i,j}|\mathcal{F}_{i,j}]\|^2|\mathcal{F}_{i,j}]$$

$$= \mathbb{E}[\|\nabla_y f_2^{\xi^{i,j}} - \nabla_y f_2 + \nabla_{yz}^2 f_3 \mathbb{E}[[\nabla_{zz}^2 f_3^{\xi^{i,j}}]^{-1}|\mathcal{F}_{i,j}]\nabla_z f_2 - \nabla_{yz}^2 f_3^{\xi^{i,j}} [\nabla_{zz}^2 f_3^{\xi^{i,j}}]^{-1}\nabla_z f_2^{\xi^{i,j}}\|^2|\mathcal{F}_{i,j}]$$

$$\leq 2\sigma_{\nabla f_2}^2 + 2\mathbb{E}[\|\nabla_{yz}^2 f_3 \mathbb{E}[[\nabla_{zz}^2 f_3^{\xi^{i,j}}]^{-1}|\mathcal{F}_{i,j}]\nabla_z f_2 - \nabla_{yz}^2 f_3^{\xi^{i,j}} [\nabla_{zz}^2 f_3^{\xi^{i,j}}]^{-1}\nabla_z f_2^{\xi^{i,j}}\|^2|\mathcal{F}_{i,j}],$$

(D.48)

Now, dealing with the contents of the norm in the right-most term of equation D.48, we have

$$\nabla_{yz}^2 f_3 \mathbb{E}[[\nabla_{zz}^2 f_3^{\xi^{i,j}}]^{-1}|\mathcal{F}_{i,j}]\nabla_z f_2 - \nabla_{yz}^2 f_3^{\xi^{i,j}} [\nabla_{zz}^2 f_3^{\xi^{i,j}}]^{-1}\nabla_z f_2^{\xi^{i,j}}$$

$$= (\nabla_{yz}^2 f_3 - \nabla_{yz}^2 f_3^{\xi^{i,j}})\mathbb{E}[[\nabla_{zz}^2 f_3^{\xi^{i,j}}]^{-1}|\mathcal{F}_{i,j}]\nabla_z f_2 \; + \; \nabla_{yz}^2 f_3^{\xi^{i,j}} (\mathbb{E}[[\nabla_{zz}^2 f_3^{\xi^{i,j}}]^{-1}|\mathcal{F}_{i,j}] - [\nabla_{zz}^2 f_3^{\xi^{i,j}}]^{-1})\nabla_z f_2$$

$$+ \nabla_{yz}^2 f_3^{\xi^{i,j}} [\nabla_{zz}^2 f_3^{\xi^{i,j}}]^{-1}(\nabla_z f_2 - \nabla_z f_2^{\xi^{i,j}}).$$

Using this, the fact that $\|a+b+c\|^2 \leq 3\left(\|a\|^2 + \|b\|^2 + \|c\|^2\right)$, with $a$, $b$, and $c$ real-valued vectors, along with the consistency of matrix norms, and applying Assumptions 3.1, 3.5, 3.4, and 3.6, we can see that the norm term in equation D.48 can be bounded as

$$\mathbb{E}[\|\nabla_{yz}^2 f_3 \mathbb{E}[[\nabla_{zz}^2 f_3^{\xi^{i,j}}]^{-1}|\mathcal{F}_{i,j}]\nabla_z f_2 - \nabla_{yz}^2 f_3^{\xi^{i,j}} [\nabla_{zz}^2 f_3^{\xi^{i,j}}]^{-1}\nabla_z f_2^{\xi^{i,j}}\|^2|\mathcal{F}_{i,j}]$$

$$\leq 3\sigma_{\nabla^2 f_3}^2 b_{zz}^2 L_{f_2}^2 + 3\mathbb{E}[\|\nabla_{yz}^2 f_3^{\xi^{i,j}}\|^2|\mathcal{F}_{i,j}]W_{zz}^2 \theta_i^2 L_{f_2}^2 + 3\mathbb{E}[\|\nabla_{yz}^2 f_3^{\xi^{i,j}}\|^2|\mathcal{F}_{i,j}]\mathbb{E}[\|[\nabla_{zz}^2 f_3^{\xi^{i,j}}]^{-1}\|^2|\mathcal{F}_{i,j}]\sigma_{\nabla f_2}^2$$

$$\leq 3\sigma_{\nabla^2 f_3}^2 b_{zz}^2 L_{f_2}^2 + 3(\sigma_{\nabla^2 f_3}^2 + L_{\nabla f_3}^2)W_{zz}^2 \theta_i^2 L_{f_2}^2 + 3(\sigma_{\nabla^2 f_3}^2 + L_{\nabla f_3}^2)(W_{zz}^2 \theta_i^2 + b_{zz}^2)\sigma_{\nabla f_2}^2,$$

where the last inequality follows from using $\mathbb{E}[\|\nabla_{yz}^2 f_3^{\xi^{i,j}}\|^2|\mathcal{F}_{i,j}] = \mathrm{Var}[\nabla_{yz}^2 f_3^{\xi^{i,j}}|\mathcal{F}_{i,j}] + \|\mathbb{E}[\nabla_{yz}^2 f_3^{\xi^{i,j}}|\mathcal{F}_{i,j}]\|^2 \leq \sigma_{\nabla^2 f_3}^2 + L_{\nabla f_3}^2$ (by the definition of variance along with Assumptions 3.1 and 3.4) and by using $\mathbb{E}[\|[\nabla_{zz}^2 f_3^{\xi^{i,j}}]^{-1}\|^2|\mathcal{F}_{i,j}] = \mathrm{Var}[[\nabla_{zz}^2 f_3^{\xi^{i,j}}]^{-1}|\mathcal{F}_{i,j}] + \|\mathbb{E}[[\nabla_{zz}^2 f_3^{\xi^{i,j}}]^{-1}|\mathcal{F}_{i,j}]\|^2 \leq W_{zz}^2 \theta_i^2 + b_{zz}^2$ (by the definition of variance along with Assumptions 3.5 and 3.6). Plugging this expression back into equation D.48 and using the fact that $0 \leq \theta_i^2 \leq 1$, we obtain the desired result

$$\mathbb{E}[\|\tilde{g}_{f_2}^{i,j} - \bar{g}_{f_2}^{i,j}\|^2|\mathcal{F}_{i,j}]$$

$$\leq 2\sigma_{\nabla f_2}^2 + 6\sigma_{\nabla^2 f_3}^2 b_{zz}^2 L_{f_2}^2 + 6(\sigma_{\nabla^2 f_3}^2 + L_{\nabla f_3}^2)W_{zz}^2 L_{f_2}^2 + 6(\sigma_{\nabla^2 f_3}^2 + L_{\nabla f_3}^2)(W_{zz}^2 + b_{zz}^2)\sigma_{\nabla f_2}^2$$

$$:= \hat{\tau}.$$

(D.49)

$\square$

**Lemma D.6 (Boundedness of ML direction)** *Under Assumptions 3.1, 3.2, 3.4, 3.5, and 3.6, there exists the positive constant $\Upsilon$ such that*

$$\mathbb{E}[\|\tilde{g}_{f_2}^{i,j}\|^2|\mathcal{F}_{i,j}] \; \leq \; \Upsilon \qquad and \qquad \|\bar{g}_{f_2}^{i,j}\|^2 \; \leq \; \Upsilon.$$

**Proof.** From the definition of variance, we have $\mathbb{E}[\|\tilde{g}_{f_2}^{i,j}\|^2|\mathcal{F}_{i,j}] = \|\bar{g}_{f_2}^{i,j}\|^2 + \mathbb{E}[\|\tilde{g}_{f_2}^{i,j} - \bar{g}_{f_2}^{i,j}\|^2|\mathcal{F}_{i,j}]$. Now, adding and subtracting $\nabla_y \bar{f}(x^i, y^{i,j}, z^{i,j+1})$ to the first term, followed by utilizing the fact that $\|a+b\|^2 \leq 2\|a\|^2 + 2\|b\|^2$, with $a$ and $b$ real-valued vectors, along with Lemma D.5, we have

$$\mathbb{E}[\|\tilde{g}_{f_2}^{i,j}\|^2|\mathcal{F}_{i,j}] \leq 2\|\bar{g}_{f_2}^{i,j} - \nabla_y \bar{f}(x^i, y^{i,j}, z^{i,j+1})\|^2 + 2\|\nabla_y \bar{f}(x^i, y^{i,j}, z^{i,j+1})\|^2 + \hat{\tau}. \quad \text{(D.50)}$$

Referencing equation A.4 and equation A.9, the $\|\nabla_y \bar{f}(x^i, y^{i,j}, z^{i,j+1})\|$ term can be bounded by applying the triangle inequality, the consistency of matrix norms, and Assumptions 3.1 and 3.2, yielding

$$\|\nabla_y \bar{f}(x^i, y^{i,j}, z^{i,j+1})\| \leq L_{f_2} + \frac{L_{\nabla f_3} L_{f_2}}{\mu_z} \quad \implies \quad \|\nabla_y \bar{f}(x^i, y^{i,j}, z^{i,j+1})\|^2 \leq \left(L_{f_2} + \frac{L_{\nabla f_3} L_{f_2}}{\mu_z}\right)^2.$$

Substituting this back into equation D.50, utilizing Lemma D.5, and letting $W := L_{f_2} + \frac{L_{\nabla f_3} L_{f_2}}{\mu_z}$, yields

$$\mathbb{E}[\|\tilde{g}_{f_2}^{i,j}\|^2 | \mathcal{F}_{i,j}] = 2\hat{\omega}^2 \theta_i^2 + \hat{\phi}, \quad \text{where} \quad \hat{\phi} := 2W^2 + \hat{\tau}. \tag{D.51}$$

Finally, using the fact that $0 < \theta_i^2 \leq 1$, it follows that

$$\mathbb{E}[\|\tilde{g}_{f_2}^{i,j}\|^2 | \mathcal{F}_{i,j}] \leq \Upsilon, \quad \text{where} \quad \Upsilon := 2\hat{\omega}^2 + \hat{\phi}. \tag{D.52}$$

The second result follows from the definition of variance and applying bound equation D.52, yielding

$$\|\bar{g}_{f_2}^{i,j}\|^2 = \mathbb{E}[\|\tilde{g}_{f_2}^{i,j}\|^2 | \mathcal{F}_{i,j}] - \mathbb{E}[\|\tilde{g}_{f_2}^{i,j} - \bar{g}_{f_2}^{i,j}\|^2 | \mathcal{F}_{i,j}] \leq \mathbb{E}[\|\tilde{g}_{f_2}^{i,j}\|^2 | \mathcal{F}_{i,j}] \leq \Upsilon.$$

$\square$

# E  LIPSCHITZ CONTINUITY PROPERTIES

This appendix contains all of the statements of derived Lipschitz continuity properties of the functions, gradients, Hessians, and Jacobians involved in the trilevel adjoint gradient equation 2.2. All of their corresponding proofs are provided in Appendix B.5 of the PhD thesis Kent (2025).

**Proposition E.1** *Under Assumptions 3.1–3.2, there exist positive constants $L_z$, $L_{z_{xy}}$, and $L_{z_y}$, such that the following Lipschitz continuity properties hold:*

$$\|z(x_1) - z(x_2)\| \leq L_z \|x_1 - x_2\|, \tag{E.1}$$
$$\|z(x_1, y_1) - z(x_2, y_2)\| \leq L_{z_{xy}} \|(x_1, y_1) - (x_2, y_2)\|, \tag{E.2}$$
$$\|z(x, y_1) - z(x, y_2)\| \leq L_{z_y} \|y_1 - y_2\|. \tag{E.3}$$

**Proposition E.2** *Under Assumptions 3.1–3.3, there exist positive constants $L_y$, $L_{\nabla z}$, $L_{\bar{F}}$, $L_{\bar{F}_y}$, $L_{\bar{F}_z}$, $L_{\nabla_{yx}^2 \bar{f}}$, $L_{\nabla_{yy}^2 \bar{f}}$, $L_F$, $L_{F_{yz}}$, and $L_{\nabla y}$, such that the following Lipschitz properties hold:*

$$\|y(x_1) - y(x_2)\| \leq L_y \|x_1 - x_2\|, \tag{E.4}$$
$$\|\nabla z(x_1) - \nabla z(x_2)\| \leq L_{\nabla z} \|x_1 - x_2\|, \tag{E.5}$$
$$\|\nabla_y \bar{f}(x_1) - \nabla_y \bar{f}(x_2)\| \leq L_{\bar{F}} \|x_1 - x_2\|, \tag{E.6}$$
$$\|\nabla_y \bar{f}(x, y_1) - \nabla_y \bar{f}(x, y_2)\| \leq L_{\bar{F}_y} \|y_1 - y_2\|, \tag{E.7}$$
$$\|\nabla_y \bar{f}(x, y, z_1) - \nabla_y \bar{f}(x, y, z_2)\| \leq L_{\bar{F}_z} \|z_1 - z_2\|, \tag{E.8}$$
$$\|\nabla_{yx}^2 \bar{f}(x_1, y(x_1)) - \nabla_{yx}^2 \bar{f}(x_2, y(x_2))\| \leq L_{\nabla_{yx}^2 \bar{f}} \|x_1 - x_2\|, \tag{E.9}$$
$$\|\nabla_{yy}^2 \bar{f}(x_1, y(x_1)) - \nabla_{yy}^2 \bar{f}(x_2, y(x_2))\| \leq L_{\nabla_{yy}^2 \bar{f}} \|x_1 - x_2\|, \tag{E.10}$$
$$\|\nabla f(x_1) - \nabla f(x_2)\| \leq L_F \|x_1 - x_2\|, \tag{E.11}$$
$$\|\nabla f(x, y_1, z_1) - \nabla f(x, y_2, z_2)\| \leq L_{F_{yz}} \|(y_1, z_1) - (y_2, z_2)\|, \tag{E.12}$$
$$\|\nabla y(x_1) - \nabla y(x_2)\| \leq L_{\nabla y} \|x_1 - x_2\|. \tag{E.13}$$

A useful intermediary result of Proposition E.1 is the following:

$$\|\nabla_x z(x, y(x))\| \leq \frac{L_{\nabla f_3}}{\mu_z} \text{ and } \|\nabla_y z(x, y(x))\| \leq \frac{L_{\nabla f_3}}{\mu_z}, \tag{E.14}$$

where $\mu_z$ is the constant of the strong convexity of $f_3$ (Assumption 3.2).

# F  NUMERICAL EXPERIMENTAL SETUP

## F.1  COMPUTING THE TSG ADJOINT GRADIENT INEXACTLY

Let us rewrite the adjoint gradient equation 2.2 in $x$ as follows:

$$\nabla f = a - AB^{-1}b, \tag{F.1}$$

where $a = \nabla_x f_1 - \nabla_{xz}^2 f_3 \nabla_{zz}^2 f_3^{-1} \nabla_z f_1$, $A = \nabla_{xy}^2 \bar{f}$, $B = \nabla_{yy}^2 \bar{f}$, and $b = \nabla_y f_1 - \nabla_{yz}^2 f_3 \nabla_{zz}^2 f_3^{-1} \nabla_z f_1$. Note that this is the same structure arising in the adjoint gradient of a BLO problem. Two approaches have been proposed in the BLO literature to deal with $B^{-1}$. One option is to compute the adjoint gradient by first solving the linear system given by the adjoint equation $B\lambda = b$ for the adjoint variables $\lambda$, and then calculating $a - A\lambda$. The second option is to truncate the Neumann series given by $B^{-1} = \sum_{h=0}^{\infty}(I - B)^h$, which requires the assumption of $\|B\|_2 < 1$ to guarantee the convergence of the series. Note that the same two approaches can be used to deal with $\nabla_{zz}^2 f_3^{-1}$ in $a$ and $b$ in equation F.1, as well as in the expression for $\nabla_y \bar{f}$, given in equation F.2 below. The expression for the adjoint gradient $\nabla_y \bar{f}$ follows from equation A.9 in Appendix A, together with equation A.4:

$$\nabla_y \bar{f}(x,y) = \nabla_y f_2 - \nabla_{yz}^2 f_3 \nabla_{zz}^2 f_3^{-1} \nabla_z f_2, \tag{F.2}$$

where all gradients and Hessians on the right-hand side are evaluated at $(x, y, z(x, y))$.

## F.2   TSG-N-FD

Our first proposed method, TSG-N-FD, solves the adjoint systems in equation 2.2 and equation F.2 by using an iterative method where each Hessian-vector product is approximated with an FD scheme. In particular, let us rewrite equation 2.2 and equation F.2 by highlighting the adjoint systems as follows:

$$\nabla f = (\nabla_x f_1 - \nabla_{xz}^2 f_3 \underbrace{\nabla_{zz}^2 f_3^{-1} \nabla_z f_1}_{\lambda_z}) - \nabla_{xy}^2 \bar{f} \underbrace{\nabla_{yy}^2 \bar{f}^{-1}(\nabla_y f_1 - \nabla_{yz}^2 f_3 \underbrace{\nabla_{zz}^2 f_3^{-1} \nabla_z f_1}_{\lambda_z})}_{\lambda_y}, \tag{F.3}$$

$$\nabla_y \bar{f} = \nabla_y f_2 - \nabla_{yz}^2 f_3 \underbrace{\nabla_{zz}^2 f_3^{-1} \nabla_z f_2}_{\bar{\lambda}_z}. \tag{F.4}$$

Specifically, the adjoint systems in equation F.3 are $\nabla_{zz}^2 f_3 \lambda_z = \nabla_z f_1$ and $\nabla_{yy}^2 \bar{f} \lambda_y = \nabla_y f_1 - \nabla_{yz}^2 f_3 \lambda_z$. The adjoint system in equation F.4 is $\nabla_{zz}^2 f_3 \bar{\lambda}_z = \nabla_z f_2$.

First, we focus on equation F.3. In TSG-N-FD, the adjoint system $\nabla_{zz}^2 f_3 \lambda_z = \nabla_z f_1$ is solved for the adjoint variables $\lambda_z$ by using the linear CG method, with $\nabla_{zz}^2 f_3 \lambda_z$ being approximated as follows:

$$\nabla_{zz}^2 f_3(x^i, y^{i,j}, z^{i,j,k}; \xi^{i,j,k})\lambda_z \approx \frac{\nabla_z f_3(x^i, y^{i,j}, z_+^{i,j,k}; \xi^{i,j,k}) - \nabla_z f_3(x^i, y^{i,j}, z_-^{i,j,k}; \xi^{i,j,k})}{2\varepsilon}, \tag{F.5}$$

where $z_{\pm}^{i,j,k} = z^{i,j,k} \pm \varepsilon \lambda_z$, with $\varepsilon > 0$. Then, the adjoint equation $\nabla_{yy}^2 \bar{f} \lambda_y = \nabla_y f_1 - \nabla_{yz}^2 f_3 \lambda_z$ is solved for the adjoint variables $\lambda_y$ by using the linear CG method again, with $\nabla_{yz}^2 f_3 \lambda_z$ being approximated via an FD scheme similar to equation F.5, and $\nabla_{yy}^2 \bar{f} \lambda_y$ being approximated as follows:

$$\nabla_{yy}^2 \bar{f}(x^i, y^{i,j}, z^{i,j+1}; \xi^{i,j})\lambda_y \approx \frac{\nabla_y \bar{f}(x^i, y_+^{i,j}, z^{i,j+1}; \xi^{i,j}) - \nabla_y \bar{f}(x^i, y_-^{i,j}, z^{i,j+1}; \xi^{i,j})}{2\varepsilon}, \tag{F.6}$$

where $y_{\pm}^{i,j} = y^{i,j} \pm \varepsilon \lambda_y$, with $\varepsilon > 0$. Then, the adjoint gradient is calculated from

$$\nabla f \approx (\nabla_x f_1 - \nabla_{xz}^2 f_3 \lambda_z) - \nabla_{xy}^2 \bar{f} \lambda_y, \tag{F.7}$$

where $\nabla_{xz}^2 f_3 \lambda_z$ and $\nabla_{xy}^2 \bar{f} \lambda_y$ are approximated via FD schemes similar to equation F.5 and equation F.6, respectively.

Let us now focus on equation F.4. The adjoint system $\nabla_{zz}^2 f_3 \bar{\lambda}_z = \nabla_z f_2$ is solved for the adjoint variables $\bar{\lambda}_z$ by using the linear CG method, with $\nabla_{zz}^2 f_3 \bar{\lambda}_z$ being approximated as in equation F.5. Then, the adjoint gradient is calculated from

$$\nabla_y \bar{f} \approx \nabla_y f_2 - \nabla_{yz}^2 f_3 \bar{\lambda}_z, \tag{F.8}$$

where $\nabla_{yz}^2 f_3 \bar{\lambda}_z$ is approximated via an FD scheme similar to equation F.5.

The schema of BSG-N-FD is included in Algorithm 4. The "N" in the algorithm name refers to the Newton-type system defined by the adjoint equation, while the "FD" refers to the finite-difference approximations we use. We set the FD parameter value to $\varepsilon = 0.1$.

---

**Algorithm 4** TSG-N-FD

---

TSG-N-FD is obtained from Algorithm 3 with the following modifications:

 In Step 1, replace Step 2 of Algorithm 2 with the following:
  **Step 2.** Compute an approximation $\tilde{g}_{f_2}^{i,j}$, using equation F.8.

 In Step 3, replace the content with the following:
  **Step 3.** Compute an approximation $\tilde{g}_{f_1}^{i}$, using equation F.7.

---

## F.3   TSG-AD

Our second proposed method, TSG-AD, is based on the truncated Neumann series approach. We will illustrate such an approach by applying it to the two terms from the adjoint gradient equation 2.2 that require it, i.e., $\nabla_{xz}^2 f_3 \nabla_{zz}^2 f_3^{-1} \nabla_z f_1$ and $\nabla_{xy}^2 \bar{f} \nabla_{yy}^2 \bar{f}^{-1} b$, where $b = \nabla_y f_1 - \nabla_{yz}^2 f_3 \nabla_{zz}^2 f_3^{-1} \nabla_z f_1$. A similar approach can be applied to handle the term $\nabla_{yz}^2 f_3 \nabla_{zz}^2 f_3^{-1} \nabla_z f_2$ in equation F.2.

Let us start with $\nabla_{xz}^2 f_3 \nabla_{zz}^2 f_3^{-1} \nabla_z f_1$ from equation 2.2. Approximating $\nabla_{zz}^2 f_3^{-1}$ using a Neumann series (i.e., $B^{-1} = \sum_{h=0}^{\infty} (I - B)^h$, where $B$ plays the role of $\nabla_{zz}^2 f_3$) requires $\|\nabla_{zz}^2 f_3\|_2 < 1$, which is a strong assumption in practice. However, recall that $f_3$ is thrice continuously differentiable and $\nabla_z f_3$ is Lipschitz continuous in $z$ with some constant $C_0 > 0$ by Assumption 3.1, implying that $\|\nabla_{zz}^2 f_3\| < C_0$ (Beck, 2017, Theorem 5.12). Therefore, following a common approach in the BLO literature Ji et al. (2020), we apply the truncated Neumann series to approximate $[(1/C_0)\nabla_{zz}^2 f_3]^{-1}$.

Given an accuracy level $Q > 0$, we can write the truncated Neumann series as $B^{-1} \approx \sum_{h=0}^{Q} (I - B)^h = \sum_{h=0}^{Q} \prod_{\ell=Q-h+1}^{Q} (I - B)$, where we define $\prod_{\ell=Q+1}^{Q}(\cdot) = I$ for simplicity. Therefore, we can approximate $\nabla_{zz}^2 f_3^{-1} \nabla_z f_1$ as follows

$$
\nabla_{zz}^2 f_3^{-1} \nabla_z f_1 \; \approx \; (1/C_0) \left( \sum_{h=0}^{Q} \prod_{\ell=Q-h+1}^{Q} (I - (1/C_0)\nabla_{zz}^2 f_3(x^i, y^{i,j}, z^{i,j,k}; \xi_\ell^{i,j,k})) \right) \nabla_z f_1,
$$
(F.9)

with $\xi_\ell^{i,j,k}$ representing the $\ell$-th sample (or batch of samples) from the sequence of random variables $\{\xi^{i,j,k}\}$. The expression on the right-hand side of equation F.9 can be efficiently computed using the AD procedure detailed in Algorithm 5. Then, given $v_z$ returned by Algorithm 5, we can compute the desired term as follows

$$
\nabla_{xz}^2 f_3 \nabla_{zz}^2 f_3^{-1} \nabla_z f_1 \; \approx \; \frac{d}{dx} (\nabla_z f_3(x^i, y^{i,j}, z^{i,j,k}; \xi^{i,j,k})^\top v_z),
$$
(F.10)

where differentiation with respect to $x$ is performed using AD (note that $\nabla_z f_3$ is a function of $x$ and $v_z$ is fixed).

---

**Algorithm 5** Automatic differentiation procedure to compute $\nabla^2_{zz} f_3^{-1} \nabla_z f_1$

---

**Input:** $(x^i, y^{i,j}, z^{i,j,k})$.

**For** $\ell = 1, 2, \ldots, Q$ **do**
$\quad G_\ell(z^{i,j,k}) = z^{i,j,k} - (1/C_0)\nabla_z f_3(x^i, y^{i,j}, z^{i,j,k}; \xi_\ell^{i,j,k})$.
**End**
Set $r_0 = \nabla_z f_1(x^i, y^{i,j}, z^{i,j,k}; \xi^{i,j,k})$.
**For** $h = 0, 1, \ldots, Q - 1$ **do**
$\quad$ Calculate $r_{h+1} = \frac{d}{dz}(G_{h+1}(z^{i,j,k})^\top r_h) = (I - (1/C_0)\nabla^2_{zz} f_3(x^i, y^{i,j}, z^{i,j,k}; \xi_{h+1}^{i,j,k})) r_h$,
where differentiation with respect to $z$ is performed using AD (note that $G_{h+1}$ is a function of $z$
and $r_h$ is fixed).
**End**

**Output:** $v_z = (1/C_0)\sum_{h=0}^{Q} r_h$.

---

Let us now focus on $\nabla^2_{xy}\bar{f}\nabla^2_{yy}\bar{f}^{-1}b$ from equation 2.2. Recall that $f_2$ is twice continuously differentiable and $\nabla_y\bar{f}$ is Lipschitz continuous in $y$ with some constant $C_1 > 0$ as a consequence of equation E.7 in Proposition E.2 of Appendix E (such a proposition implies that $C_1$ is equal to $L_{\bar{F}_y}$, but we prefer to use $C_1$ for generality). Similar to equation F.9, we apply the truncated Neumann series to $[(1/C_1)\nabla^2_{yy}\bar{f}]^{-1}$, which allows us to approximate $\nabla^2_{yy}\bar{f}^{-1}b$ as follows:

$$\nabla^2_{yy}\bar{f}^{-1}b \approx (1/C_1)\left(\sum_{h=0}^{Q}\prod_{\ell=Q-h+1}^{Q}(I - (1/C_1)\nabla^2_{yy}\bar{f}(x^i, y^{i,j}, z^{i,j+1}; \xi_\ell^{i,j}))\right) b, \qquad \text{(F.11)}$$

where $\xi_\ell^{i,j}$ represents the $\ell$-th sample (or batch of samples) from the sequence of random variables $\{\xi^{i,j}\}$. The expression on the right-hand side of equation F.9 can be efficiently computed using the AD procedure detailed in Algorithm 6. Then, given $v_y$ returned by Algorithm 6, we can compute the desired term as follows

$$\nabla^2_{xy}\bar{f}\nabla^2_{yy}\bar{f}^{-1}b \approx \frac{d}{dx}(\nabla_y\bar{f}(x^i, y^{i,j}, z^{i,j+1}; \xi^{i,j})^\top v_y), \qquad \text{(F.12)}$$

where differentiation with respect to $x$ is performed using AD (note that $\nabla_y\bar{f}$ is a function of $x$ and $v_y$ is fixed).

---

**Algorithm 6** Automatic differentiation procedure to compute $\nabla^2_{yy}\bar{f}^{-1}b$

---

**Input:** $(x^i, y^{i,j}, z^{i,j+1})$.

**For** $\ell = 1, 2, \ldots, Q$ **do**
$\quad G_\ell(y^{i,j}) = y^{i,j} - (1/C_1)\nabla_y\bar{f}(x^i, y^{i,j}, z^{i,j+1}; \xi_\ell^{i,j})$.
**End**
Set $r_0 = b$.
**For** $h = 0, 1, \ldots, Q - 1$ **do**
$\quad$ Calculate $r_{h+1} = \frac{d}{dy}(G_{h+1}(y^{i,j})^\top r_h) = (I - (1/C_1)\nabla^2_{yy}\bar{f}(x^i, y^{i,j}, z^{i,j+1}; \xi_{h+1}^{i,j})) r_h$,
where differentiation with respect to $y$ is performed using AD (note that $G_{h+1}$ is a function of $y$
and $r_h$ is fixed).
**End**

**Output:** $v_y = (1/C_1)\sum_{h=0}^{Q} r_h$.

---

The schema of TSG-AD is included in Algorithm 7.

---

**Algorithm 7** TSG-AD

TSG-AD is obtained from Algorithm 3 with the following modifications:

In Step 1, replace Step 2 of Algorithm 2 with the following:
**Step 2.** Compute an approximation $\tilde{g}_{f_2}^{i,j}$ by applying to $\nabla_{yz}^2 f_3 \nabla_{zz}^2 f_3^{-1} \nabla_z f_2$ the same approach that was used to compute $\nabla_{xz}^2 f_3 \nabla_{zz}^2 f_3^{-1} \nabla_z f_1$ in equation F.10.

In Step 3, replace the content with the following:
**Step 3.** Compute an approximation $\tilde{g}_{f_1}^i$, using equation F.10 and equation F.12.

---

### F.4 SYNTHETIC TRILEVEL PROBLEMS

Given $h_x \in \mathbb{R}^n$, $h_y \in \mathbb{R}^m$, and $h_z \in \mathbb{R}^t$, the UL and ML objective functions for both the quadratic and quartic synthetic trilevel problems considered in the experiments are respectively given by

$$f_1(x,y,z) = h_x^\top x + h_y^\top y + h_z^\top z + 0.5\, x^\top H_{xx} x + x^\top H_{xy} y + x^\top H_{xz} z, \qquad (F.13)$$

$$f_2(x,y,z) = 0.5\, y^\top H_{yy} y - y^\top H_{yx} x - y^\top H_{yz} z, \qquad (F.14)$$

where $H_{xx} \in \mathbb{R}^{n \times n}$ and $H_{yy} \in \mathbb{R}^{m \times m}$ are symmetric positive definite matrices, and $H_{xy} \in \mathbb{R}^{n \times m}$, $H_{xz} \in \mathbb{R}^{n \times t}$, $H_{yx} = H_{xy}^\top$, and $H_{yz} \in \mathbb{R}^{m \times t}$ are arbitrary matrices. The LL objective functions of the two problems are respectively defined as follows

$$f_3(x,y,z) = 0.5\, z^\top H_{zz} z - z^\top H_{zx} x - z^\top H_{zy} y, \qquad (F.15)$$

$$f_3(x,y,z) = 0.5\|z^\top H_{zz} z - z^\top H_{zx} x - z^\top H_{zy} y\|^2, \qquad (F.16)$$

where $H_{zz} \in \mathbb{R}^{t \times t}$ is a symmetric positive definite matrix, and $H_{zx} = H_{xz}^\top$ and $H_{zy} = H_{yz}^\top$ are arbitrary matrices.

In all the numerical experiments, we considered the same dimension at all levels (i.e., $n = m = t = 50$) for the quadratic problem, and varying dimensions (i.e., $n = m = 5$ and $t = 1$) for the quartic problem. In equation F.13, the components of the vectors $h_x$, $h_y$, and $h_z$ were randomly generated from a uniform distribution between 0 and 10 for the quadratic problem, and between 0 and 0.1 for the quartic problem. We set all matrices in equation F.13–equation F.16 equal to identity matrices, except for $H_{yy}$ in equation F.14, which was set to four times the identity matrix.

When using equation F.15, our choices for the matrices in equation F.13–equation F.15 ensure that $f_3$, $\bar{f}$, and $f$ have unique solutions.[†] When using equation F.16, the resulting LL problem has two optimal solutions: $z(x,y) = 0$ and $z(x,y) = H_{zx} x + H_{zy} y$. Our choice for the initial points $x^0$, $y^{0,0}$, and $z^{0,0,0}$ ensures that the methods considered in the experiments converge to the LL optimal solution $z(x,y) = H_{zx} x + H_{zy} y$. Specifically, the components of the initial points were randomly generated from a uniform distribution over the interval [0, 20] when using equation F.15, and over the intervals [-0.4, 0], [-0.2, 0], and [-0.6, 0] (for the UL, ML, and LL variables, respectively) when using equation F.16.

All algorithms (i.e., TSG-H, TSG-N-FD, and TSG-AD) were compared using a decaying step size at each level. Specifically, we used $\alpha_i = \bar{\alpha}/i$, $\beta_j = \bar{\beta}/j$, and $\gamma_k = \bar{\gamma}/k$, where $\bar{\alpha}$, $\bar{\beta}$, and $\bar{\gamma}$ are positive scalars carefully chosen to ensure good performance for each algorithm (without conducting extensive, time-consuming grid searches at all levels, as our goal is not to compare our algorithms against others). The values of $\bar{\alpha}$, $\bar{\beta}$, and $\bar{\gamma}$ are provided in Table 2.

### F.4.1 ADDITIONAL FIGURES AND DISCUSSION FOR THE SYNTHETIC TRILEVEL PROBLEMS

In the deterministic case, Figures 8 and 9 break down the behavior of TSG-H, TSG-N-FD, and TSG-AD at the UL, ML, and LL levels. Specifically, such figures plot the sequence of $f(x^i)$ values (upper plot), $\bar{f}(x^i, y^{i,j})$ values (middle plot), and $f_3(x^i, y^{i,j}, z^{i,j,k})$ values (lower plot). They

---

[†]We have $z(x,y) = H_{zz}^{-1}(H_{zx}x + H_{zy}y)$, $y(x) = (H_{yy} - 2H_{yz}H_{zz}^{-1}H_{zy})^{-1}(H_{yx} + H_{yz}H_{zz}^{-1}H_{zx})$, $\nabla_y \bar{f}(x,y) = H_{yy}y - H_{yx}x - H_{yz}H_{zz}^{-1}(H_{zx}x + 2H_{zy}y)$, and $\nabla_{yy}^2 \bar{f}(x,y) = H_{yy} - 2H_{yz}H_{zz}^{-1}H_{zy}$. We omit the expressions of $\nabla f(x)$ of $\nabla^2 f(x)$ for brevity.

Table 2: Details of the stepsizes ($\alpha_i = \bar{\alpha}/i$, $\beta_j = \bar{\beta}/j$, $\gamma_k = \bar{\gamma}/k$) used across algorithms for the synthetic quadratic and quartic trilevel problems

| Problem | Algorithm | Case | $\bar{\alpha}$ | $\bar{\beta}$ | $\bar{\gamma}$ |
|---|---|---|---|---|---|
| | TSG-H | Deterministic | 0.3 | 0.2 | 0.1 |
| | TSG-N-FD | Deterministic | 0.01 | 0.1 | 0.05 |
| Quadratic | TSG-AD | Deterministic | 0.01 | 0.1 | 0.1 |
| | TSG-H | Stochastic | 0.1 | 0.1 | 0.1 |
| | TSG-N-FD | Stochastic | 0.01 | 0.1 | 0.1 |
| | TSG-AD | Stochastic | 0.01 | 0.1 | 0.1 |
| | TSG-H | Deterministic | 0.3 | 0.2 | 0.1 |
| | TSG-N-FD | Deterministic | 0.3 | 0.2 | 0.0001 |
| Quartic | TSG-AD | Deterministic | 0.3 | 0.2 | 0.0001 |
| | TSG-H | Stochastic | 0.3 | 0.2 | 0.1 |
| | TSG-N-FD | Stochastic | 0.01 | 0.01 | 0.001 |
| | TSG-AD | Stochastic | 0.3 | 0.2 | 0.0001 |

also include the values $f(x_*)$ (only for the quadratic problem, where it can be computed analytically), with $x_*$ denoting the optimal solution of the trilevel problem, as well as $\bar{f}(x^i, y(x^i))$ and $f_3(x^i, y^{i,j}, z(x^i, y^{i,j}))$. The goal is for the sequences of $f$, $\bar{f}$, and $f_3$ values to converge to their respective dashed lines. In the middle- and lower-level plots, the horizontal axis represents cumulative ML and LL iterations, respectively.

As evident from Figure 8, for the quadratic problem, the sequences of function values at the UL and ML problems converge when the function values at the ML and LL problems, respectively, also converge. As evident from Figure 9, for the quartic problem, the sequences of function values at all levels converge after a few iterations.

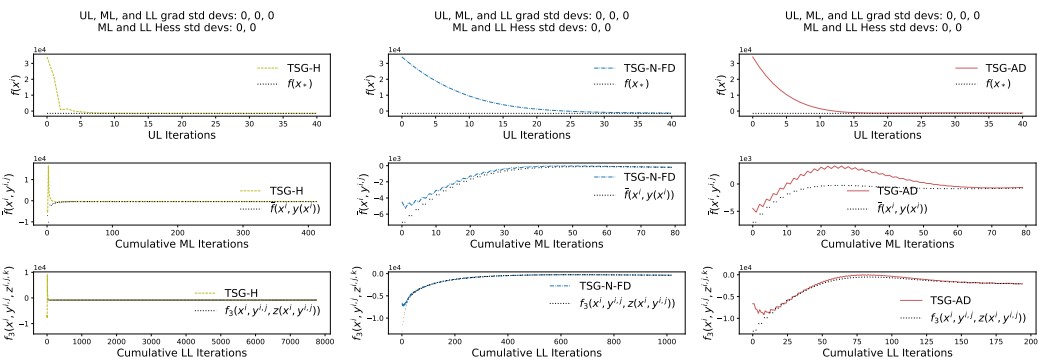

Figure 8: Breakdown of the algorithms, quadratic problem, deterministic case.

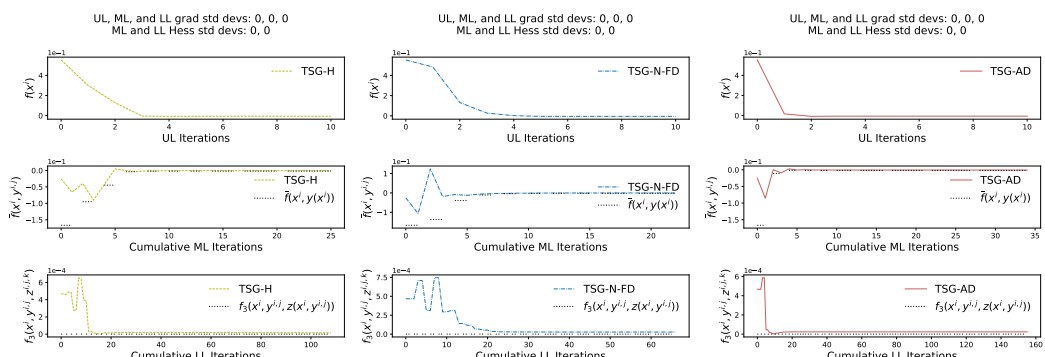

Figure 9: Breakdown of the algorithms, quartic problem, deterministic case.

### F.5 TRILEVEL HYPERPARAMETER ADVERSARIAL TUNING

Let us denote the whole learning dataset used in the experiments by $\mathcal{D} = \{(u_j, v_j), j \in \{1, \ldots, N\}\}$, which consists of $N$ pairs given by a feature vector $u_j$ and the corresponding true label $v_j$. We denote the datasets used for training and validation as $D_{\mathcal{D}}$ and $D_{\text{val}}$, which respectively consist of $N_{\mathcal{D}}$ and $N_{\text{val}}$ pairs extracted from the original dataset $\mathcal{D}$ (with additional pairs set aside for testing). Let $\phi(u_j; \theta)$ be the prediction function, where $\theta$ is a vector of parameters. The adversarial training problem can be written according to the following minimax formulation (see, e.g., Madry et al. (2017)):

$$\min_{\theta} \frac{1}{N_{\mathcal{D}}} \sum_{(u,v) \in D_{\mathcal{D}}} \max_{\|\delta_u\| \leq \epsilon} \ell(\phi(u + \delta_u; \theta), v), \tag{F.17}$$

where $\delta_u$ is a perturbation vector associated with each sample $u$ in the training set, and $\epsilon$ is a positive threshold. Introducing $\delta = (\delta_u \mid (u, v) \in D_{\mathcal{D}})$, we propose the following TLO problem for hyperparameter adversarial tuning, inspired by Sato et al. (2021):

$$\min_{\lambda \in \mathbb{R}, \, \theta \in \mathbb{R}^m, \, \delta \in \mathbb{R}^t} \quad \frac{1}{N_{\text{val}}} \sum_{(u,v) \in D_{\text{val}}} \ell(\phi(u; \theta), v)$$

$$\text{s.t. } \theta, \delta \in \underset{\theta \in \mathbb{R}^m, \delta \in \mathbb{R}^t}{\arg\min} \quad \frac{1}{N_{\mathcal{D}}} \sum_{(u,v) \in D_{\mathcal{D}}} \ell(\phi(u + \delta_u; \theta), v) + \Phi(\theta; \lambda) \tag{F.18}$$

$$\text{s.t. } \delta \in \underset{\delta \in \mathbb{R}^t}{\arg\max} \quad \frac{1}{N_{\mathcal{D}}} \sum_{(u,v) \in D_{\mathcal{D}}} \ell(\phi(u + \delta_u; \theta), v) - \Psi(\delta),$$

where $\lambda$ is a penalty coefficient, and $\Phi(\theta; \lambda) = (e^\lambda \|\theta\|_{1\star})/m$ (with $\|\cdot\|_{1\star}$ being a smooth approximation of the $\ell_1$-norm (Saheya et al., 2019, Eq. (18) with $\mu = 0.25$)) and $\Psi(\delta) = (c\|\delta\|^2)/(mN_{\mathcal{D}})$ (with $c = 0.1$ being a penalty coefficient) are penalty terms that penalize large values of $\theta$ and $\delta$, respectively. To convert the LL problem into a minimization problem, we switch to $\arg\min$ by multiplying the objective function by $-1$. Following Sato et al. (2021), we use a linear prediction function and mean squared error (MSE) as the loss function in our experiments.

Regarding the datasets used in the experiments, the red and white wine quality datasets Cortez et al. (2009) contain 1,599 and 4,898 samples, respectively, each with 11 features, while the California housing dataset Pace & Barry (1997) contains 20,640 samples and 8 features. Each dataset is split into training, validation, and test sets in proportions of 70%, 15%, and 15%, respectively.

For TSG-N-FD and TSG-AD, we use the same configuration described in Section 4.2, including decaying stepsizes ($\alpha_i = \bar{\alpha}/i$, $\beta_j = \bar{\beta}/j$, and $\gamma_k = \bar{\gamma}/k$), where the positive scalars $\bar{\alpha}$, $\bar{\beta}$, and $\bar{\gamma}$ are selected via grid search over the set $\{0.1, 0.01, 0.001\}$. For the BSG-AD algorithms, which are derived from TSG-AD to solve the BLO problems obtained from equation F.18, we once again use decaying stepsizes selected via grid search over $\{0.1, 0.01, 0.001\}$. Specifically, the values of $\bar{\alpha}$, $\bar{\beta}$, and $\bar{\gamma}$ are provided in Table 3. In all experiments, the algorithms use a minibatch size of 64 for training, and the results presented in the figures are averaged over 10 runs.

Table 3: Details of the stepsizes ($\alpha_i = \bar{\alpha}/i$, $\beta_j = \bar{\beta}/j$, $\gamma_k = \bar{\gamma}/k$) used across algorithms, formulations, and datasets in the trilevel hyperparameter adversarial tuning experiments

| Algorithm | Formulation | Dataset | $\bar{\alpha}$ | $\bar{\beta}$ | $\bar{\gamma}$ |
|-----------|-------------|---------|------|------|------|
| TSG-N-FD | Sato et al. (2021) | Red Wine | 0.1 | 0.1 | 0.1 |
| TSG-AD | Sato et al. (2021) | Red Wine | 0.01 | 0.01 | 0.01 |
| TSG-AD | equation F.18 | Red & White Wine | 0.1 | 0.01 | 0.1 |
| TSG-AD | equation F.18 | California Housing | 0.01 | 0.001 | 0.01 |
| BSG-AD (without UL) | equation F.18 | Red & White Wine | – | 0.01 | 0.1 |
| BSG-AD (without UL) | equation F.18 | California Housing | – | 0.001 | 0.1 |
| BSG-AD (without LL) | equation F.18 | Red & White Wine | 0.1 | 0.01 | – |
| BSG-AD (without LL) | equation F.18 | California Housing | 0.1 | 0.001 | – |

### F.5.1 ADDITIONAL FIGURES AND DISCUSSION FOR TRILEVEL HYPERPARAMETER ADVERSARIAL TUNING

In Figure 10, we assess the TLO problem for hyperparameter adversarial tuning proposed in Sato et al. (2021), which can be obtained by swapping the ML and LL problems in equation F.18. The results on the red wine dataset demonstrate that both TSG-N-FD and TSG-AD exhibit essentially similar performance in terms of test MSE. However, the test MSE values are consistently worse or comparable to those obtained using the formulation in equation F.18 (see Figure 5), which is why we discontinued testing the formulation from Sato et al. (2021).

When using equation F.18, TSG-N-FD does not perform well and is therefore excluded from further analysis. This outcome is not surprising, as the results from the synthetic problems in Section 4.2 indicated that TSG-N-FD is more affected by noise in $\nabla f_3$ than TSG-AD. In equation F.18, the noise is further amplified by the fact that the size of $\delta$ corresponds to the number of rows times the number of columns of the entire dataset, making $\nabla f_3$ more susceptible to minibatch sampling.

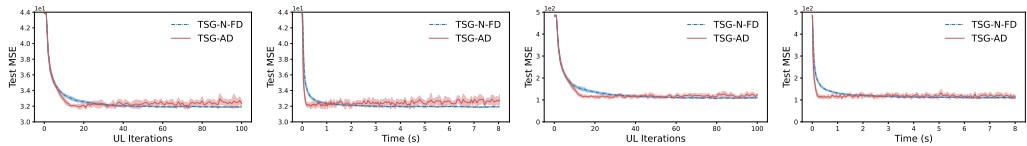

Figure 10: Trilevel adversarial learning formulation proposed in Sato et al. (2021), red wine quality dataset. The two left plots correspond to noise with standard deviation 0, and the two right plots to standard deviation 5.

