# OpenReview forum: "A stochastic gradient method for trilevel optimization"
_ICLR.cc/2026/Conference — ICLR 2026 Conference Withdrawn Submission_

### Official Review · Reviewer_KAxF · 2025-10-27

**Soundness:** 3
**Presentation:** 3
**Contribution:** 2
**Rating:** 4
**Confidence:** 4

**Summary:**

The paper studies trilevel optimizations problems of the form $\min_{x,y,z}f_1(x,y,z)$ with $z\in\arg\min_{z'}f_3(x,y,z')$ and $y,z\in\arg\min_{y',z'}\in f_2(x, y', z')$. The authors reformulate this problem into a single-level program and leverage implicit differentiation to design a stochastic solver to handle this problem. They establish a convergence rate in terms of gradient norm in $\mathcal{O}(J/\sqrt{I})$ where $I$ is the number of outer iterations and $J$ is the number of middle-level iterations. Numerical experiments on a synthetic problem and on adversarial hyperparameter tuning are provided to evaluate the proposed method.

**Strengths:**

- **S1**: The paper tackle trilevel optimization which has been little explored in the community.

**Weaknesses:**

- **W1**: The introduction of the problem could be improved for clarity. It would be helpful to explicitly state that the $f_i$ are expectations, as I guess by reading the paper. Without this clarification, the motivation for using a stochastic solver is unclear. Additionally, sample derivatives, like $\nabla_z f_3(x, y^{i,j}, z^{i,j,k}; \xi^{i,j,k})$ (line 146), are not defined. Stating immediately from the beginning that $f_3(x, y, z) = \mathbb{E}_\xi[f_3(x, y, z;\xi)]$ would be sufficient to give a meaning to $\nabla_z f_3(x, y^{i,j}, z^{i,j,k}; \xi^{i,j,k})$.

- **W2**: In Assumption 3.3, $\bar{f}$ is assumed to be strongly-convex. As mentioned by the author, this hold when $z(x, y)$ is an affine function of $(x, y)$ and $f_2$ is strongly convex with respect to $(x, z)$. However, I am not sure if there are practical cases where this still holds without having $z(x, y)$ affine w.r.t. $(x, y)$. In general, even if $f_2$ and $f_3$ are strongly convex, $\bar{f}$ can be non-convex.

- **W3**: It would be convenient to have the convergence result expressed in terms of sample complexity, that is the number of gradients, Hessian-vector products, and Jacobian-vector products required to reach a gradient norm lower than $\epsilon$ (as done in [1, 2] for instance). This gives a better understanding of the overall complexity of the algorithm. In particular, due to the lower bound on $K$, the cost of one outer iteration seems very high, and thus also the total complexity, which is a bit hidden when providing the result only in terms of convergence rate.

- **W4**: The theory does not take into account that the solution of the different linear systems that appear in $\nabla f$ are approximated, differing from the practical setting.

#### Minor/Typo
- **Equations (TLO) and (BLO)**: Avoid writing expressions of the form $z\in\arg\min_z f(z)$. It is more rigorous to distinguish the variable $z$ and the dummy variable in the argmin. by writing $z\in\arg\min_z' f(z')$.
- **line 357**: "approximating each Hessian-vector product by using automatic differentiation". AD enables to compute exact derivatives and HVPs [3, 4] thus "computing" might be more appropriate than "approximating".

- **line 1937**: "$\sum_{i=0}^{I-1}\alpha_i = I\alpha_i$" -> Don't use the symbol indexing the sum $i$ outside the sum (r.h.s.).

[1] [Ghadimi, S. and Lan, G. *Stochastic first- and zeroth-order methods for nonconvex stochastic programming*. SIAM Journal on Optimization, 2013.](https://arxiv.org/pdf/1309.5549)

[2] [Arbel, M. and  Mairal, J. *Amortized Implicit Differentiation for Stochastic Bilevel Optimization*. ICLR, 2022.](https://arxiv.org/pdf/2111.14580)

[3] [Pearlmutter, B. A. *Fast exact multiplication by the hessian*. Neural computation, 1994](https://www.bcl.hamilton.ie/~barak/papers/nc-hessian.pdf)

[4] [Dagréou, M., Ablin, P., Vaiter, S., and Moreau, T. *How to compute hessian-vector products?* ICLR Blogposts 2024.](https://iclr-blogposts.github.io/2024/blog/bench-hvp/)

**Questions:**

N/A

---

### Official Review · Reviewer_pmYw · 2025-10-30

**Soundness:** 1
**Presentation:** 3
**Contribution:** 1
**Rating:** 2
**Confidence:** 4

**Summary:**

The paper studies trilevel optimization, a natural extension of bilevel optimization.

**Strengths:**

As a theory-driven article, it clearly articulates the problem formulation, hypotheses, algorithm derivation, and convergence rate.

**Weaknesses:**

I believe the article has several major weaknesses:
1. The novelty and technical contributions of the paper are clearly insufficient, and I don't think it meets the conference standards. The results presented are a natural extension of bilevel optimization, and I do not see any significant challenges in this extension.

2. The problem formulation in the paper relies on several strong assumptions. For example, Assumption 3.3 is a very strong assumption, and since $\bar f$ is an implicit objective, it is often not possible to verify whether it is strongly convex in many practical situations. Furthermore, Assumptions 3.5 and 3.6 are also quite strong, and not all papers in the bilevel optimization field adopt these assumptions.

3.  The algorithm presented in the paper directly uses matrix inversion, such as in Eq. (2.4), which is very inefficient for large-scale optimization problems. A core difficulty in bilevel optimization is to use Hessian-vector products or gradients as oracles to avoid the aforementioned matrix inversion.

**Questions:**

Please see the weakness part.

---

### Official Review · Reviewer_pcKp · 2025-11-01

**Soundness:** 2
**Presentation:** 2
**Contribution:** 2
**Rating:** 4
**Confidence:** 3

**Summary:**

The paper proposes the first trilevel stochastic gradient framework (TSG) for unconstrained trilevel optimization, extending bilevel stochastic methods to an additional level. It provides convergence analysis under strong convexity of the middle and lower levels and presents experiments on synthetic and hyperparameter-adversarial tasks.

**Strengths:**

- The paper formalizes the trilevel adjoint gradient through a clean extension of bilevel implicit differentiation.

- It provides a comprehensive theoretical treatment with explicit assumptions (smoothness, strong convexity, unbiased gradient noise, bounded Hessians).

- Experiments include synthetic and real data settings, showing numerical feasibility.

**Weaknesses:**

- The formal definition of trilevel optimization (Eq. TLO) is incorrect or at least misleading: it allows
$y, z \in \arg\min f_2(x, y, z), \ z\in \arg\min f_3(x, y, z)$

which implies that the middle level and the lower level both optimize $z$.
This formulation conflicts with the intended hierarchical semantics. But later in line 110-113, we could know that the middle level should optimize only $y$.

- TSG is essentially a mechanical extension of stochastic bilevel frameworks (e.g., StocBiO and BSA).
The adjoint-gradient recursion simply adds one more nesting, reusing existing implicit-differentiation logic.
No algorithmic innovation is introduced.

- The analysis assumes differentiability up to third order and global Lipschitz continuity for all derivatives, along with strong convexity of both middle and lower problems. Assumption 3.5 and 3.6 introduce some very strong assumptions about "Bounded inverse Hessians" and "Bounded bias of inverted stochastic Hessians", but these assumptions are not verified.

- The experiments are conducted on small-scale settings with marginal performance differences, providing little empirical evidence of real advantage. There is no analysis of computational cost or memory cost.
Moreover, the paper fails to present any compelling machine learning application that genuinely motivates the use of trilevel optimization.

**Questions:**

Please check Weaknesses

---

### Official Review · Reviewer_b4WX · 2025-11-01

**Soundness:** 2
**Presentation:** 2
**Contribution:** 2
**Rating:** 2
**Confidence:** 3

**Summary:**

This paper proposes TSG, a stochastic gradient-based algorithm for trilevel optimization, claiming to be the first such method. It provides convergence guarantees under strong convexity for the middle- and lower-level subproblems and presents experiments on synthetic problems and a trilevel formulation for hyperparameter adversarial tuning.

**Strengths:**

1. The paper addresses an underexplored area by extending stochastic gradient methods to trilevel optimization.

2. The convergence analysis is detailed.
The experimental section includes both synthetic and application-based evaluations.

**Weaknesses:**

1. The theoretical results rely on strong convexity of both the middle- and lower-level problems, which greatly limits applicability to real-world scenarios. In particular, Assumption 3.3 is justified by the authors through the case where the LL problem is a QP; however, in that case, the entire trilevel structure can be reformulated as a linearly constrained bilevel problem, for which many well-established algorithms already exist. Many real-world multi-level problems (e.g., in adversarial training or hyperparameter optimization) do not satisfy such convexity assumptions.


2. The convergence analysis (Theorem B.5) is provided for the full, "exact" TSG method (what the paper calls TSG-H in the experiments). This method requires access to stochastic third-order derivative tensors, which is computationally prohibitive. The paper's practical algorithms (TSG-N-FD and TSG-AD) avoid this, but do so by introducing new sources of bias (e.g., from finite-difference approximation or truncated Neumann series) that are not modeled in the convergence theory. The claim that the theory "covers all forms of inexactness" is thus incorrect, as it omits the approximation error of the practical methods.

3. The synthetic experiments are not compelling. The hyperparameter adversarial tuning application, while relevant, is compared only against simplified bilevel baselines. There is no comparison with more competitive or state-of-the-art methods, and the performance gains are not clearly demonstrated.

4. The paper does not provide new insights into the structure or difficulty of trilevel problems beyond what is already known from bilevel optimization. The propagation of errors across levels is discussed, but the conclusions are intuitive and not deeply novel.

5. The paper is very long and technically dense, but fails to highlight clear conceptual advancements. Algorithmic design choices seem heuristic and not well-motivated.

**Questions:**

Why were no comparisons made with other trilevel or multi-level optimization methods ?

---

### Note · Authors · 2025-11-19

I have read and agree with the venue's withdrawal policy on behalf of myself and my co-authors.